# A bipartite bacterial virulence factor targets the complement system and neutrophil activation

Kurni Kurniyati[1,4], Nicholas D Clark [2,4], Hongxia Wang[1], Yijie Deng[1], Ching Wooen Sze[1], Michelle B Visser[3], Michael G Malkowski [2✉] & Chunhao Li [1✉]

## Abstract

**The complement system and neutrophils constitute the two main pillars of the host innate immune defense against infection by bacterial pathogens. Here, we identify T-Mac, a novel virulence factor of the periodontal pathogen *Treponema denticola* that allows bacteria to evade both defense systems. We show that T-Mac is expressed as a pre-protein that is cleaved into two functional units. The N-terminal fragment has two immunoglobulin-like domains and binds with high affinity to the major neutrophil chemokine receptors FPR1 and CXCR1, blocking *N*-formyl-Met-Leu-Phe- and IL-8-induced neutrophil chemotaxis and activation. The C-terminal fragment functions as a cysteine protease with a unique proteolytic activity and structure, which degrades several components of the complement system, such as C3 and C3b. Murine infection studies further reveal a critical T-Mac role in tissue damage and inflammation caused by bacterial infection. Collectively, these results disclose a novel innate immunity-evasion strategy, and open avenues for investigating the role of cysteine proteases and immunoglobulin-like domains of gram-positive and -negative bacterial pathogens.**

**Keywords** Immune Evasion; Complement System; Neutrophils; Cysteine Protease; Immunoglobulin-like Domain
**Subject Categories** Immunology; Microbiology, Virology & Host Pathogen Interaction

## Introduction

The innate immune system is the first line of defense against invading microbial pathogens (Diacovich and Gorvel, 2010; Paludan et al, 2021). In humans, neutrophils, also known as polymorphonuclear leukocytes (PMNs), are critical components of the innate immune system comprising 50–70% of all circulating leukocytes (Margraf et al, 2022; Mayadas et al, 2014). Neutrophils detect and respond to inciting stimuli (also known as

chemoattractants) released during tissue damage and/or infection, a phenomenon called chemotaxis, and are recruited to inflamed sites in great numbers within minutes to fight against invading microbes. Neutrophil chemoattractants include lipids (e.g., leukotriene B4), *N*-formylated peptides (e.g., fMLF), anaphylatoxins (e.g., C3a and C5a), and chemokines (e.g., IL-8), which exert their effects through interaction with specific G protein-coupled receptors (GPCRs) expressed on cell surfaces and induce cell polarization and subsequent migration towards inflamed sites (Mayadas et al, 2014; Petri and Sanz, 2018). For example, fMLF stimulates neutrophils chemotaxis through binding to the *N*-formyl peptide receptor 1 (FPR1); interleukin-8 (IL-8) does so mainly through binding to the CXC chemokine receptor 1 and 2 (CXCR1/2) (Holmes et al, 1991; Russo et al, 2014; Zhuang et al, 2022). Signaling through GPCRs can also prime neutrophil activation in response to other activating agents (e.g., IgG and opsonin) leading to full cellular activation. Activated neutrophils kill invaded pathogens through antimicrobial peptides, phagocytosis, generation of reactive oxygen species (ROS), and formation of neutrophil extracellular traps (NETs) (Kobayashi et al, 2018; Mayadas et al, 2014). In addition to immune defense, neutrophils can shape the immune landscape by communicating with macrophages, dendritic cells (DCs), and cells of the adaptive immune response through direct cell-cell contact or soluble mediators (Margraf et al, 2022; Mayadas et al, 2014).

The complement system constitutes the other arm of the host innate immune defense against microbial infections (Hajishengallis et al, 2017; Hovingh et al, 2016; Reis et al, 2019). It comprises a network of soluble and cell membrane proteins that work in a coordinated manner, which can be activated by antigen-antibody complexes (classical pathway, CP), carbohydrates (lectin pathway, LP), or by a variety of surfaces that are not protected by natural inhibitors (alternative pathway, AP). CP is initiated when C1q, the recognition protein of the C1 complex, binds to an antigen-antibody complex. AP is activated by continuous "turnover" of C3 into C3b which then binds to foreign pathogens in a process called opsonization. LP is triggered by binding of mannose-binding lectin to carbohydrate residues on bacterial cell surfaces. All three pathways converge in the formation of a C3 convertase which triggers an activation cascade culminating in the formation of the membrane attack complex (MAC) on bacterial cell surfaces. MAC forms cytotoxic pores that disrupt bacterial cell membranes and results in

[1]Philips Institute for Oral Health Research, School of Dentistry, Virginia Commonwealth University, Richmond, VA, USA. [2]Department of Structural Biology, Jacobs School of Medicine and Biomedical Sciences, University of Buffalo, the State University of New York, Buffalo, NY, USA. [3]Department of Oral Biology, School of Dentistry, University of Buffalo, the State University of New York, Buffalo, NY, USA. [4]These authors contributed equally: Kurni Kurniyati, Nicholas D Clark. ✉E-mail: mgm22@buffalo.edu; cli5@vcu.edu

cell lysis and death. In addition to the innate immunity, the complement system is also interwoven with the adaptive immunity, e.g., opsonization by the complement system can interact with various receptors on B cells (i.e., Complement receptor 1 and 2) and modulate their responses (Dunkelberger and Song, 2010; Mastellos et al, 2024). Likewise, the complement system can also interact with T cells, i.e., complement deficiency reduces priming of CD4 and CD8 T cells (Reis et al, 2019; West et al, 2018).

As an open entry of the gastrointestinal track, the oral cavity constitutes a unique environment that harbors hundreds of different microbes in the form of a microbial consortium (microbiota) while being protected by the innate and adaptive immunity. Consequently, a constant battle ensues between the oral microbiota and the host immune systems. This dynamic has been at the center of studying the pathogenesis of periodontitis, a chronic inflammation that affects nearly one half of American adult population (Hajishengallis and Chavakis, 2021; Hajishengallis et al, 2023; Lamont et al, 2018). Periodontitis is a risk factor for numerous systemic illnesses (de Pablo et al, 2009; Dominy et al, 2019; Lamont et al, 2018; Whitmore and Lamont, 2014), such as diabetes, cardiovascular disease, rheumatoid arthritis, cancer, and Alzheimer's disease. Among those identified periodontal pathogens, the oral bacterium *Treponema denticola* is motile and invasive, acting as a pioneer at the interface between the oral microbiota and the gingival epithelium. Therefore, it directly encounters robust host immune attacks launched by the complement system and neutrophils, a primary phagocyte in the oral cavity (Dashper et al, 2011; Ellen and Galimanas, 2005; Yousefi et al, 2020). As a successful periodontal pathogen, *T. denticola* can adapt to and survive in this hostile environment and even become a predominant species as the disease progresses. In doing so, *T. denticola* must evade the host immune defense exerted by the complement system and neutrophils; however, its underpinning molecular mechanism remains elusive. In this report, we first identified a putative virulence factor in the secretome of *T. denticola* and then elucidated its role and underlying mechanism in the innate immune evasion by using an approach of genetics, biochemistry, immunology, cell biology, structural biology, and animal models. The results reported here provide new mechanistic insights into understanding host-pathogen interactions and bacterial innate immune evasion.

# Results

## Identification of a Mac-like protein in the culture supernatant proteome of *T. denticola*

A previous study by Sela et al reported that the supernatant prepared from *T. denticola* cultures inhibited superoxide production from human neutrophils (Sela et al, 1988), suggesting that this bacterium might produce and secrete unknown modulators into its extracellular milieu. Yet, it was technically challenging to characterize the secretome of *T. denticola* mainly because its growth media (i.e., NOS and TYGVS) contain 10% rabbit serum (Ohta et al, 1986) which overshadows the secreted proteins in the growth media. To resolve this issue, we tested different growth media without rabbit serum and found that *T. denticola* could grow in a serum-free TYGV medium supplemented with L-cysteine (Fig. EV1A). Using this medium, we

prepared *T. denticola* spent culture supernatants (SCSs) by removing bacterial cells and outer membrane vesicles (OMVs) through multi-step centrifugations, filtration, and ultracentrifugation (see Experimental Procedures for the detail). The resulting SCSs were subjected to SDS-PAGE and 2-D gel electrophoresis and numerous protein bands were detected in the SCSs of *T. denticola* but not in the growth medium alone (Fig. EV1B). MALDI-TOF-MS analysis identified 11 proteins at high confidence (Fig. EV1C), including TDE0471 (neuraminidase) and TDE0405 (major surface protein, Msp)—two secreted virulence factors of *T. denticola* (Anand et al, 2013; Kurniyati et al, 2013).

Among those identified proteins, TDE0362 consists of 647 amino acids (aa) with a predicted molecular weight (MW) of 73.94 kDa. The gene encoding this protein is within a cluster of eight open reading frames (*orfs*) (Appendix Fig. S1A). Transcriptional analyses revealed that these eight genes are co-transcribed, forming an operon that is regulated by a sigma (Paget and Helmann, 2003)-like promoter (Appendix Fig. S1). TDE0362 contains four structural units, including a N-terminal signal peptide (SP, 1–22 aa), two bacterial Ig-like (Big) domains (23–204 aa), a linker region (205–329 aa), and a C-terminal Mac-like domain (330–647 aa) (Fig. 1A,B). Big domains have been identified in several bacterial pathogens but their roles remain largely unknown (Chatterjee et al, 2021). The C-terminal Mac-like domain of TDE0362 (hereafter named C0362) shares ~25% sequence identities with IdeS and Mac-1, two cysteine proteases from *Streptococcus pyogenes* and Group A *Streptococcus* (GAS), respectively (Agniswamy et al, 2006; Lei et al, 2001; Vincents et al, 2004). Similar to IdeS and Mac-1, C0362 contains a catalytic triad (Cys-412/His-561/Asn-585) of cysteine proteases (Fig. 1C), suggesting that C0362 is a cysteine protease. *Saccharomyces cerevisiae* has served as a powerful system to study the function of bacterial virulence factors (Lesser and Miller, 2001; Siggers and Lesser, 2008). Therefore, this system was applied to assess the cytotoxicity of TDE0362. For this experiment, the genes encoding the full-length TDE0362 (FL0362, 1-647 aa) and C0362 (330–647 aa) were cloned into pYES2/NTA plasmid with a galactose-inducible promoter. Upon the induction with galactose, both FL0362 and C0362 were found to be toxic to yeast cells, while the point mutants in the catalytic triad did not exhibit such toxicity (Fig. 1D). A similar toxic effect was observed when the genes encoding these proteins were constitutively expressed in yeasts using p426GPD plasmid (Fig. EV2A–C). The N-terminus of TDE0362 (23–204 aa, hereafter named N0362) was not toxic to yeast cells (Fig. EV2D). To substantiate this study, we transiently transfected HEK293T cells with constructs expressing N0362-GFP, C0362-GFP, C0362$^{C412A}$-GFP (a point mutant in which Cys-412 was replaced with alanine) or GFP alone (control) and found that N0362-GFP was mainly localized around the nuclear membrane or around the peripheral of the nucleus while C0362-GFP led to cell shrinkage, rounding up, and lysis, indicating that C0362 is toxic. Such a toxic effect was not observed in the cells transfected with either C0362$^{C412A}$-GFP or GFP alone (Fig. 1E). Collectively, these results indicate that TDE0362 is a putative virulence factor with a distinct domain composition and pathogenic effect.

## TDE0362 is secreted and cleaved by PrtP

Detection of TDE0362 in the SCSs of *T. denticola* suggests that it is secreted. To validate this observation, the SCSs from *T. denticola*

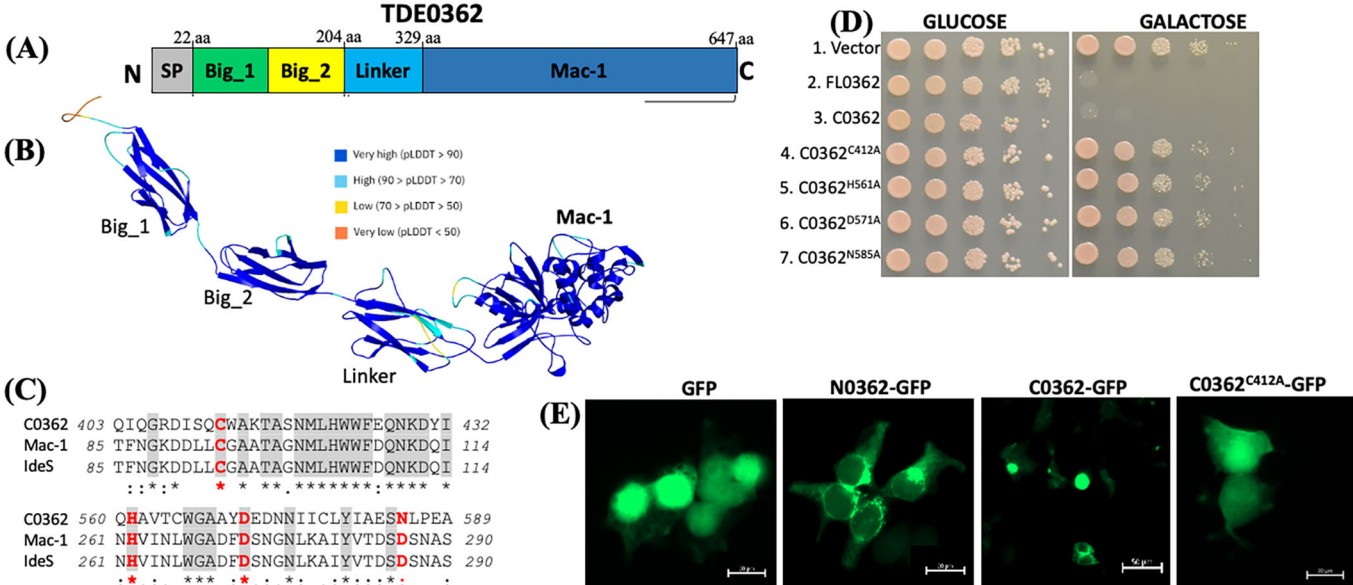

**Figure 1. TDE0362 is a homolog of Mac-1 that has a unique domain composition and pathogenic effect.**

(A) A diagram illustrating the domain composition of TDE0362 that consists of a signal peptide (SP, 1–22 aa), a N-terminal domain (N0362, 23–204 aa) with two bacteria Ig-like domains (Big_1 and Big_2), and a C-terminal domain (C0362) with a linker (205–329 aa) and a Mac-1 domain (330–647 aa). (B) Depicted is the AlphaFold model of full-length TDE0362 with secondary structure features shown in cartoon representation and colored according to pLDDT. Residues which are modeled with very high confidence (pLDDT >90) are shown as dark blue; confident (90> pLDDT >70) in light blue, low confidence (70> pLDDT >50) in yellow, and very low confidence (pLDDT < 50) in orange. (C) Sequence alignment shows that C0362 (403–589 aa; AAS10857.1) is a homolog of IdeS (85–290 aa; QSG91221.1) and Mac-1 (85–290 aa; AAZ51286.1) and harbors a catalytic triad of cysteine protease (Cys-412/His-561/Asn585). Only one part of aligned sequence was shown here. The four residues highlighted in red were mutated by site-directed mutagenesis. The alignments were conducted using Clustal Omega. (D) TDE0362 is toxic to yeast cells. Yeast strains expressing the full-length TDE0362 (FL0362, 1–647 aa), C0362 (330–647 aa) and its four point mutants under the control of the galactose-inducible promoter (Pgal1 promoter; pYES2/NTA2 plasmid) were serial-diluted and spotted onto plates containing either glucose or galactose. Plates were incubated at 30 °C for 48 h before image acquisition. (E) Expression of N0362, C0362, and C0362$^{C412A}$ mutant in HEK297T cells. For this experiment, HEK297T cells were transiently transfected with vectors expressing N0362-GFP, C0362-GFP, C C0362$^{C412A}$-GFP or GFP alone (control). Fluorescence images were collected 72 h after transfections. Scale bars: 50 μm. Source data are available online for this figure.

wild type (WT) and *Δ0362*, a deletion mutant of *TDE0362* (Appendix Fig. S2), were prepared and then subjected to immunoblots probed with polyAb, a polyclonal antibody raised using recombinant FL0362, and mAb-C3, a monoclonal antibody against C0362. Both antibodies detected one major band below 50 kDa and one minor band around ~60 kDa in the SCS of WT but none of them was detected in that of *Δ0362* (Fig. 2A,B). *T. denticola* produces a chymotrypsin-like protease (PrtP, also known as dentilisin) (Goetting-Minesky et al, 2021; Ishihara et al, 1996). A recent report suggested that PrtP is involved in the maturation of TDE0362 (Miyai-Murai et al, 2023). To corroborate this, we detected the SCSs of *ΔCCE*, a PrtP deficient mutant of *T. denticola* (Goetting-Minesky et al, 2021) and found that the band below 50 kDa was absent. Instead, four bands above ~60 kDa were detected in the SCSs of *ΔCCE* (Fig. 2A,B). It is noteworthy that one of these four bands smeared, an indication of lipoproteins. In line with this, a lipoprotein signal peptide is identified at the N-terminus of TDE0362 (Appendix Fig. S3), suggesting that TDE0362 is lipidated.

To further confirm that PrtP is responsible for the cleavage of TDE0362, we treated recombinant FL0362 with the cell lysate (CL) of WT and *ΔCCE* and found that WT-CL, but not *ΔCCE*-CL, cleaved FL0362 in a dose-dependent manner (Fig. 2C). Two major cleaved protein bands were detected, one is above 60 kDa (F1) and

the other one is below 50 kDa (F2), which are similar to what we detected in the SCSs of WT (Fig. 2A,B). F1 gradually disappeared as the dose of WT-CL increased (Fig. 2C), suggesting that this band is an intermediate product. To map their cleavage sites, F1 and F2 bands were excised and subjected to LC-MS/MS. The results showed that F1 was cleaved after Phe-132 and F2 was cleaved after Phe-235, both of which are in line with the catalytic activity of PrtP as a phenylalanine protease (Fenno et al, 2001; Goetting-Minesky et al, 2013). The cleavage at Phe-132 presumably generates a N-terminal product with a molecular mass of 14.2 kDa and a C-terminal product with a molecular mass of 59.5 kDa (F1). The cleavage at Phe-235 presumably yields a N-terminal product with a molecular mass of 25.7 kDa (F3) and a C-terminal product with a molecular mass of 48.2 kDa (F2). These two N-terminal cleaved products (F3 and F4/F5) were faint but could be visualized on SDS-PAGE and detected by immunoblots with polyAb (Fig. EV3).

In addition to the secreted form, we also determined if TDE0362 is exposed on the cell surface of *T. denticola* by using immunofluorescence assays (IFA) and detected strong fluorescent signals in the WT but not in the *Δ0362* mutant, which was restored in *CΔ0362*, an isogenic *cis*-complemented strain of *Δ0362* (Fig. 2D). The observed fluorescence signals were abolished by the treatment of proteinase K (PK), indicating that TDE0362, or its derivatives,

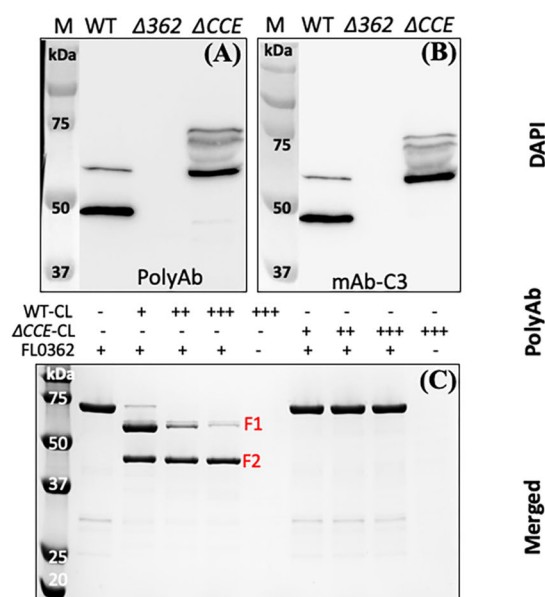

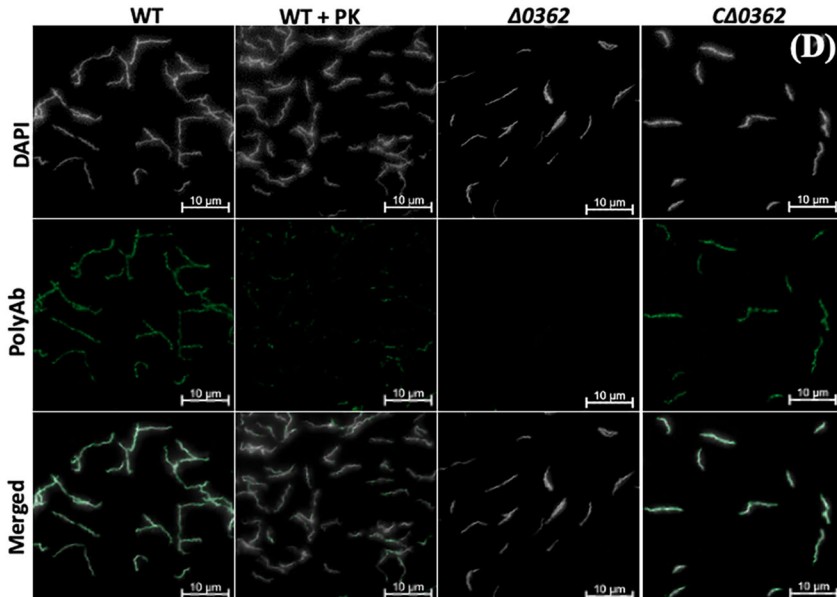

**Figure 2. TDE0362 is cleaved and secreted.**

(A, B) Detection of TDE0362 in the spent culture supernatants (SCSs) of *T. denticola* wild type (WT), *Δ0362*, a deletion mutant of TDE0362, and *ΔCCE*, a PrtP deletion mutant using immunoblots probed against polyAb, a polyclonal antibody against FL0362, and mAb-C3, a monoclonal antibody against C0362. (C) Proteolytic activity of PrtP against TDE0362. For this assay, recombinant full-length TDE0362 proteins (FL0362) were treated with different amounts of WT and *ΔCCE* whole cell lysates (CL) for overnight at 37 °C either aerobically or anaerobically. The resulted samples were subjected to 10% or 13% SDS-PAGE gel stained with Coomassie blue. F1 and F2 bands were cut and subjected to LC-MS/MS to map their cleavage sites via comparing their sequences to that of TDE0362 (see "Experimental Procedures" for details). (D) Detection of TDE0362 on *T. denticola* cell surfaces using immunofluorescence assays (IFA) with or without proteinase K treatment. For this study, *T. denticola* cells, including WT, *Δ0362*, and *CΔ0362*, an isogenic complemented strain of *Δ0362*, were first stained with polyAb and then probed against Alexa Fluor Plus 488-labled goat anti-rat IgG (H + L) secondary antibody. Micrographs were captured under fluorescence microscopy using DAPI and FITC emission filters, and the images were merged. Scale bar: 10 μm. Source data are available online for this figure.

resides on the surface of *T. denticola* cells. Taken together, we propose that after removal of the N-terminal signal peptide TDE0362 is exported, anchored to the cell surface likely through lipidation, and then is cleaved by PrtP, generating different cleaved forms, some of which are released into the culture supernatant and others are exposed on the cell surface.

## C0362 cleaves complement factors and protects bacteria from the complement killing

IdeS and Mac-1 recognize and cleave human immunoglobulin G (hIgG) (Lei et al, 2001; Sudol et al, 2022). Since C0362 is a homolog of IdeS and Mac-1, we speculated that it might have a similar proteolytic activity against hIgG. To test this, we expressed and purified His-tagged C0362 recombinant protein (rC0362, 205–647 aa) and a point mutant C412A, a conserved residue in the catalytic triad of cysteine proteases, and found that both recombinant proteins were unable to cleave hIgG (Fig. 3A). We then examined its proteolytic activity against other potential substrates, including human C3, C3b, C4, C5, factor H (FH), C4 binding protein (C4bp), and IgA, and found that rC0362, but not C412A, cleaved C3 (Fig. 3B), C3b (Fig. 3C), C4 (Fig. 3D), and FH (Fig. 3E). There was no proteolytic activity detected against C5 (Fig. 3F), C4bp, and IgA (Appendix Fig. S4). To map its cleavage site, the cleaved fragment of C3b (as indicated with asterisk, Fig. 3C) was subjected to N-terminal sequencing with Edman

degradation; the result showed that the cleavage in the α chain of C3b most likely occurred between Glu-207 and Gly-208. We also noticed that rC0362 quickly became inactive during protein purifications due to the oxidation of Cys-412. To prevent this issue, dithiothreitol (DTT, 2 mM) was added to the buffers for protein purification and dialysis, and proteolytic reactions were performed under anaerobic conditions.

Since C0362 cleaves C3, C3b, and C4, we reasoned that such a proteolytic activity might disrupt the complement system and thus protect *T. denticola* from the complement killing. To test this hypothesis, serum killing assays were carried out. Consistent with the previous reports (Kurniyati et al, 2013; McDowell et al, 2011), the wild-type strain (WT) was resistant to serum killing with an average survival rate of 82% following 90 min incubation in 25% normal human serum (NHS) (Fig. 3G). By contrast, the *Δ0362* mutant was susceptible to the serum killing with an average survival rate of 25%. The attenuated serum resistance was partially restored in the complemented strain *CΔ0362*. To substantiate this study, a similar experiment was performed in *E. coli* DH5α by pretreating NHS with *T. denticola* cell supernatants. The results revealed that the supernatants from WT and *CΔ0362* strains, but not the one from *Δ0362*, protected *E. coli* cells from the serum killing, e.g., the average survival rate of *E. coli* treated with WT-treated serum reached ~93% but the one treated with *Δ0362*-treated serum was only ~43% (Fig. 3H). Taken together, we conclude that C0362 functions as a cysteine protease that cleaves key complement

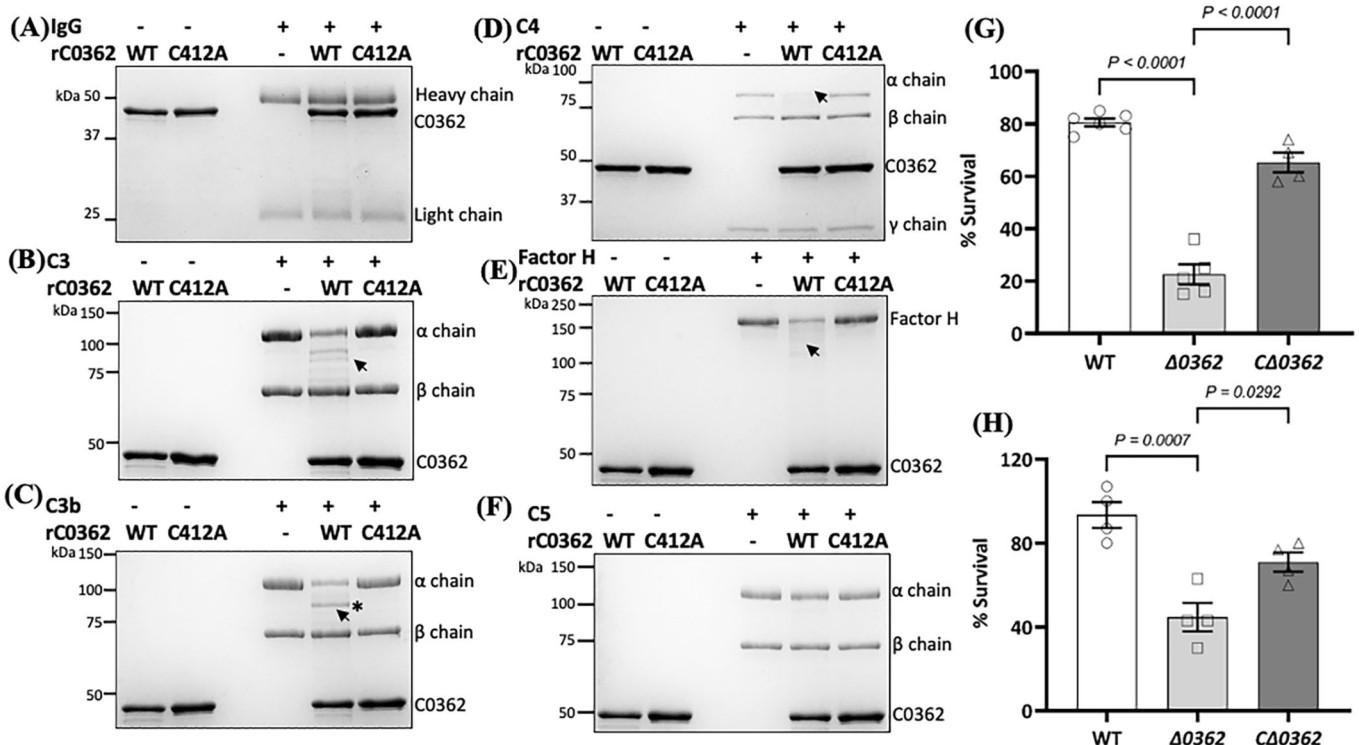

**Figure 3. C0362 degrades complement factors and disarms the bactericidal activity of human serum.**

(A–F) Measuring the proteolytic activity of recombinant C0362 protein (rC0362) against human IgG (A), C3 (B), C3b (C), C4 (D), Factor H (E) and C5 (F). For this study, different substrates, including IgG (1.16 μM), C3 (675.68 nM), C3b (852.27 nM), C4 (365.85 nM), FH (967.74 nM), and C5 (789.48 nM) were incubated with either rC0362 (1.90 μM) or its inactive form C412A point mutant (1.90 μM) in a reaction buffer (0.2 M Tris-HCl, pH 7.4, containing 0.1 M NaCl, 5 mM $CaCl_2$ and 2 mM DTT) for overnight at 37 °C anaerobically. After the incubation, Laemmli sample buffer was added to the samples, boiled for 5 min, and then subjected to SDS-PAGE followed by Coomassie blue staining. Arrows point to cleaved products; * the cleaved band in the α-chain of C3b was excised for N-terminal sequencing with Edman degradation. (G) Serum killing assays using *T. denticola* cells. For this study, *T. denticola* strains, including WT, Δ0362, and CΔ0362, were incubated with either 25% normal human serum (NHS) or heat-inactivated human serum (HIS) for 90 min at 37 °C. The number of living bacterial cells was counted using a Petroff Hausser counting chamber. The survival rates were calculated as follows: the total number of living cells in NHS divided by the total number of living cells in HIS. Five biological samples ($n = 5$) were included, $P < 0.0001$. (H) *E. coli* serum killing protection assays. Prior to serum killing assays, NHS or HIS was pretreated with the spent culture supernatants (SCSs) from WT, Δ0362, or CΔ0362 for 5 h at 37 °C anaerobically. The resulted serum was incubated with *E. coli* DH5α culture ($1 \times 10^3$ cells/ml) for 15 min at 37 °C. The final samples were plated on LB agar plates and incubated at 37 °C for 24 h before enumerating the surviving colonies. The survival rates were calculated as follows: the total number of living cells in NHS divided by the total number of living cells in HIS. The results are represented as the mean of survival rates ± standard error of the mean (SEM). Four biological samples ($n = 4$) were included. Statistical analysis was determined using one-way ANOVA followed by Tukey's multiple comparison at $P < 0.05$. Source data are available online for this figure.

factors such as C3 and thus renders bacteria protection against the complement system.

## C0362 is a cysteine protease with a unique structural feature

To elucidate the catalytic mechanism of TDE0362 as a cysteine protease, we attempted to crystallize the full-length (23–647 aa) and N0362 (23–204 aa) proteins but were unsuccessful, despite employing numerous constructs and crystallization methodologies. Nevertheless, we generated their structural models using Alpha-Fold2 which were further validated using size exclusion chromatography (SEC)-coupled small angle X-ray scattering (SEC-SAXS). The results revealed that the structure of full-length TDE0362 has an extended conformation nature with inherent mobility (Appendix Figs. S5–S7), which can explain, at least in part, the unsuccessful attempts to crystallize the full-length TDE0362 and N0362 proteins.

We successfully determined the crystal structure of C0362 (211–647 aa). Initial phases were determined utilizing SeMet SAD phasing methods on a dataset that diffracted to 2.19 Å resolution (Appendix Table S1). We then utilized molecular replacement and the model derived from SAD phasing to phase a wild-type dataset which diffracted to 1.77 Å. The overall structure of C0362 consists of two major domains, encompassing 437 residues (Fig. 4). The small N-terminal domain (N-120; residues Lys-211 through Pro-324) is comprised of two antiparallel β-sheets that form a β-sandwich, while the C-terminal domain (residues Val-325 through Gln-647) exhibits an α/β tertiary fold. N-120 is connected to the C-terminal domain by a 9-residue (residues Gln-318 through Val-326) linker which has no significant similarity to known sequences outside the *Treponema* genus in the NCBI database. N-120 shares limited structural homology in the PDB, as determined by moderate Z-scores (highest of 7.8) and moderate root mean square deviations (RMSD, ~3 Å) from the DALI server (Holm et al, 2023). Most of the similarity observed in the PDB is related to the

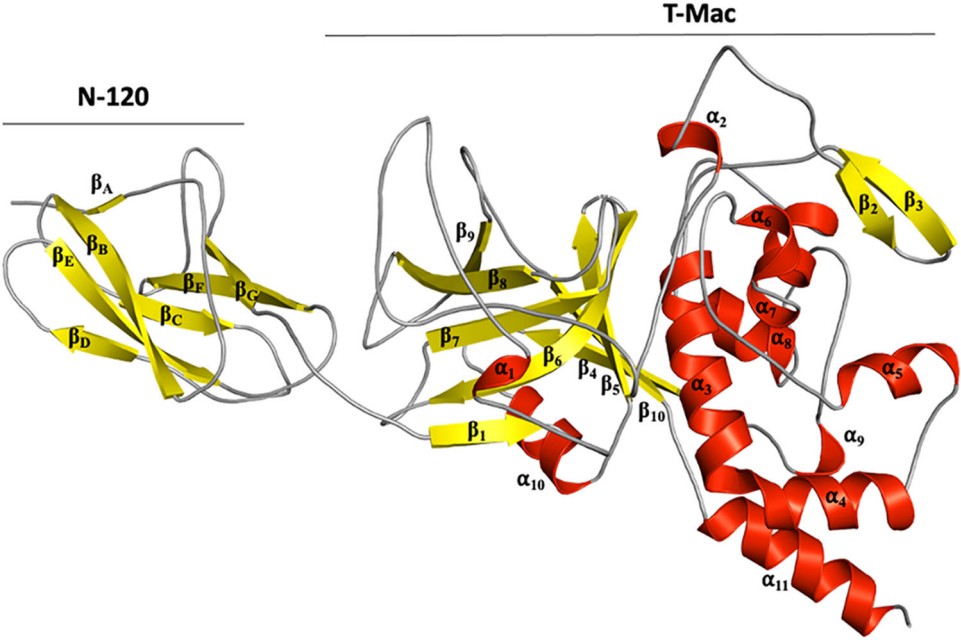

**Figure 4.  Overall Structure of C0362.**

Cartoon representation of C0362, with α-helical and β-strand secondary structural elements labeled and colored red and yellow, respectively. Secondary structural elements corresponding with the 120-domain (N-120) are labeled with letters, while those corresponding to the Mac-1 domain (T-Mac) are labeled with numbers. Unmodeled residues are shown as dashed lines.

Ig-like fold (Bao et al, 2014; Fioravanti et al, 2019; Larsbrink et al, 2014; Pakharukova et al, 2015; Verger et al, 2006).

The C-terminal Mac domain of C0362 (hereafter referred to as T-Mac) adopts the canonical papain superfamily fold, conserved across the CA clan of cysteine proteases, as defined by the MEROPS database (Rawlings et al, 2014). Closer inspection of T-Mac reveals a significant structural homology with Mac-1 in Group A *Streptococcus* (gasMac-1, PDB entry 2AU1) (Agniswamy et al, 2006) and IdeS in *S. pyogenes* (IdeS, PDB entry 1Y08) (Wenig et al, 2004). IdeS and gasMac-1 are prokaryotic representatives of the papain superfamily of cysteine proteases. As such, they share a conserved structural architecture consisting of a core that is split into two distinct lobes, with the active side cleft located at its interface. The N-terminal lobe in T-Mac (371–516 aa and 633–647 aa) is predominantly α-helical, while the C-terminal lobe (326–370 aa and 517–632 aa) consists of a twisted antiparallel β-sheet. Superposition of T-Mac with gasMac-1 and IdeS resulted in RMSDs of 1.71 Å and 1.70 Å, respectively. Despite the low sequence identity (~25%) between T-Mac, IdeS and gasMac-1, virtually all of the secondary structural elements are conserved and in topologically equivalent positions (Fig. 5A–C).

## Identification of T-Mac active site

The active site of T-Mac is located within a shallow cleft between the N- and C-terminal lobes near the surface of the enzyme (Fig. 5A). Two surface accessible loops, comprised of 478–483 aa and 550–560 aa, flank the shallow cleft. As observed in the IdeS and gasMac-1 crystal structures, these loops are conformationally flexible (Fig. 5B,C). The residues comprising the catalytic triad in

T-Mac – Cys-412, His-561, and Asn-585 are in identical spatial locations to that observed for the catalytic triad of IdeS and gasMac-1 (Ser/Cys-94, His-262, and Asp-284), when these structures are superimposed (Fig. 5D,E). The catalytic cysteine, Cys-412, is located at the N-terminal end of $\alpha_3$ at the base of the active site cleft. Closer inspection revealed additional electron density protruding from the $S\gamma$ atom of the Cys-412 side chain. Numerous modifications were built onto this side chain during model refinement. Adding a cysteine sulfenic acid residue (CSO) at position 412 (CSO-412), representing oxidation of the cysteine side chain, resulted in the best fit of the model to the electron density (Appendix Fig. S8). The side chain of CSO-412 exhibits a similar pose to that observed for Ser/Cys-94 in IdeS and gasMac-1, where the $S_\gamma$ atom forms a hydrogen bond with the $N_\zeta$ atom of Lys-379. Of note, Ser/Cys-94 makes contacts with a sulfate ion in the IdeS crystal structure (Wenig et al, 2004) and a partially disordered 2-mercaptoethanol molecule in the gasMac-1 crystal structure (Agniswamy et al, 2006).

His-561 is equivalent to His-262 in IdeS and gasMac-1, where the $N_{\varepsilon2}$ atom of the side chain forms a hydrogen bond with the $S_\gamma$ atom of Ser/Cys-94. In T-Mac, the side chain of His-561 exhibits two distinct conformations, given its relative positioning at the surface of the enzyme (Fig. 5D,E). One of the His-561 conformers lies in a similar position to that seen for His-262 in IdeS and gasMac-1, with the $N_{\varepsilon2}$ atom positioned 3.4 Å away from the $S_\gamma$ atom of CSO-412. The other His-561 side chain conformer points away from the active site. Asn-585 aids in the proper positioning and stabilization of the side chain of His-561. The side chain of Asn-585 also forms an ionic interaction with the $N_\zeta$ atom of Lys-379. While aspartic acid replaces asparagine at this position in both

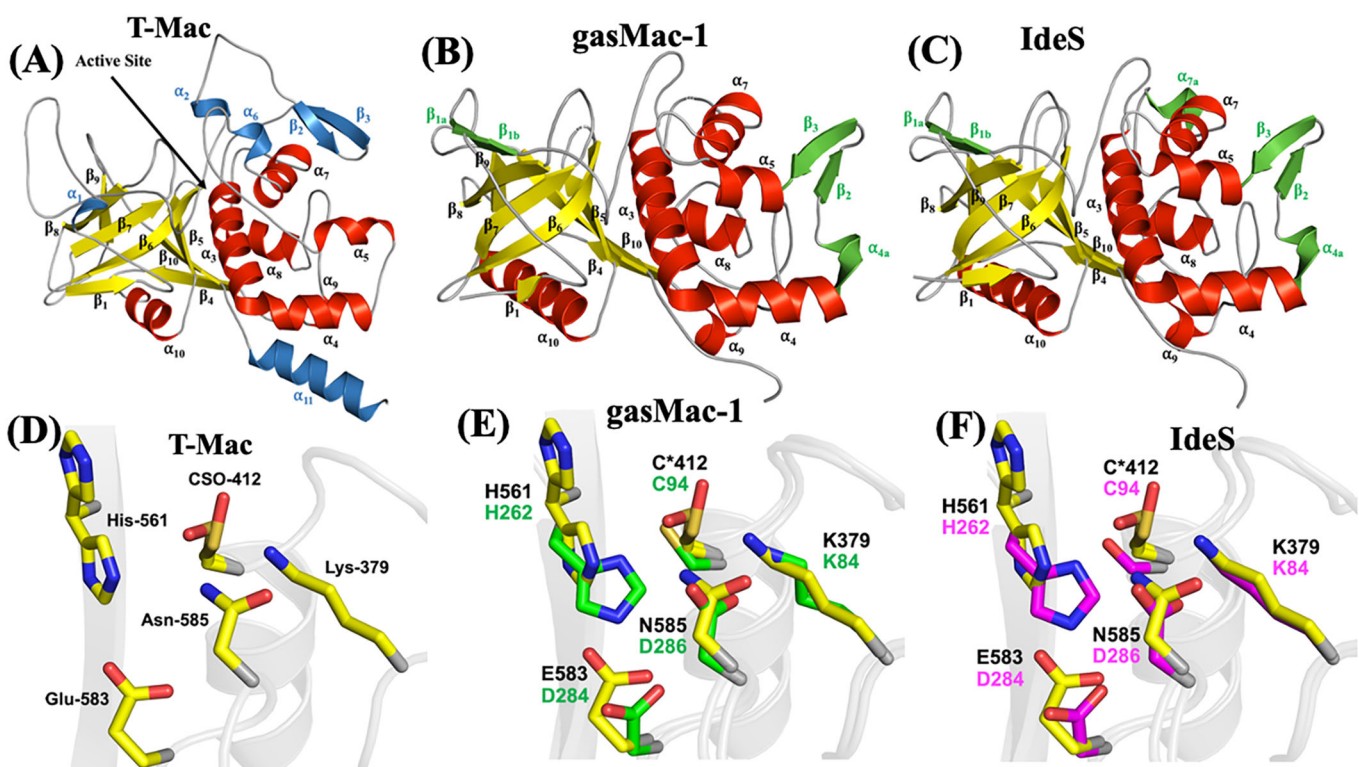

**Figure 5.  Structural comparison of T-Mac, gasMac-1, and IdeS.**

Cartoon depicting the secondary structural elements of the Mac-1 domain found in the crystal structures of (**A**) T-Mac (**B**) gasMac-1 (PDB entry 2AU1), and (**C**) IdeS (PDB entry 1Y08). α-helical and β-strand secondary structural elements that are common to all three structures are colored red and yellow, respectively, while secondary structural elements unique to T-Mac compared to the other two structures are colored green and blue, respectively. Secondary structural elements for IdeS and gasMac-1 are labeled according to their equivalent in T-Mac and denoted with a lettered number, if absent in C0362 (e.g., β₁ₐ corresponds to a β-strand is absent in T-Mac). (**D–F**) Close-up of the Mac-1 active site residues in T-Mac, gasMac-1, and IdeS. (**D**) Active site residues Cso-412, Lys-379, Glu-383, His-561, and Asn-585 of T-Mac are depicted. Cys-412 is oxidized and both Cys-412 and His-561 exhibit multiple rotamer conformations. The equivalent residues Lys-84, Cys-94, His-262, Asp-284, and Asp-286 are depicted for gasMac-1 (**E**) and IdeS (**F**). C0362 sidechains are shown as yellow sticks throughout while gasMac-1 sidechains are shown as green sticks and IdeS sidechains are shown as magenta sticks.

IdeS and gasMac-1, the aspartic acid side chain exhibits the same interactions with the equivalent histidine and lysine residues within their respective active sites (Fig. 5D,E). Lys-84, Asp-284, and Tyr-255 in IdeS and gasMac-1 have been shown to be important for maintaining the architecture of the active site, as mutation of these residues resulted in considerably decreased enzymatic activity (Agniswamy et al, 2006; Wenig et al, 2004). In T-Mac, no equivalent to Tyr-255 is present, while Lys-379 and Glu-583 are in equivalent positions to Lys-84 and Asp-284, suggesting these residues may also be important for maintaining the active site architecture of T-Mac. Collectively, these results demonstrate that the C-terminal domain of TDE0362 is a cysteine protease with some unique structural features that differ from its counterparts such as gas-Mac-1 and IdeS.

## N0362 binds to and inhibits neutrophils chemotaxis and killing

The AlphaFold2 model of N0362 (23–211 aa) contains two bacterial Ig-like (Big) domains (Fig. 1B; Appendix Figs. S5 and S6). The first Big domain (Big_1) is formed by Lys-41 through Thr-121 and folds into three distinct antiparallel β-sheets, which contains a central 4-stranded β-sheet that folds to form a concave surface. This concave surface is occluded by the second 3-stranded antiparallel β-sheet, forming a V-shaped pattern. The third 2-stranded antiparallel β-sheet is localized to the top of this V-shape, occluding the opening. Pro-122 through Val-127 are modeled as a strand and serve as a linker between the Big_1 and Big_2 domains. Thr-128 through Lys-211 comprise the Big_2 domain that is essentially identical to that described for Big_1, with three antiparallel β-sheets forming an occluded V-shaped fold.

Though Big-like domains have been identified in several bacterial pathogens (Chatterjee et al, 2021), little is known about their roles and underlying molecular mechanism. To explore the function of N0362, several experiments were conducted. Strikingly, transwell assays revealed that recombinant N0362 (23–204 aa) protein potently inhibited human neutrophils chemotaxis induced by fMLF, a chemoattractant of neutrophils (Boulay et al, 1990; Osei-Owusu et al, 2019), in a dose-dependent manner, e.g., 10 nM N0362 reduced neutrophil chemotaxis by nearly 80% (Fig. 6A). A similar inhibitory effect was observed in mouse neutrophils (Fig. 6B). Since N0362 inhibits neutrophils chemotaxis, we reasoned that such an inhibitory effect may protect *T. denticola* from neutrophils killing. To test this, neutrophils killing assays

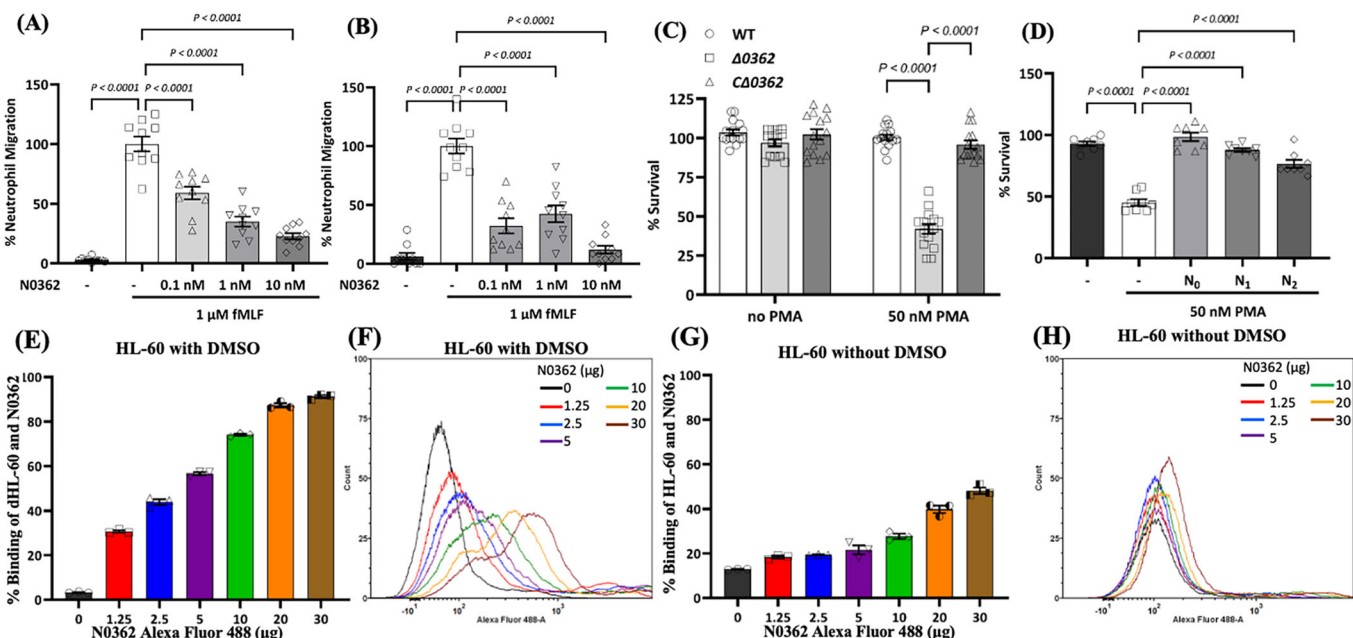

**Figure 6. N0362 inhibits neutrophil chemotaxis and killing.**

Transwell chemotaxis assays were performed using human neutrophils (**A**) and mouse neutrophils (**B**). For these assays, neutrophils were treated with different concentrations of N0362 recombinant proteins (0.1, 1, and 10 nM) for 30 min and stimulated with fMLF for 1 h in a transwell. Cells that migrated across the membrane were fixed, stained with crystal violet, and counted. Results were normalized and compared to the control with fMLF. Ten biological samples ($n = 10$) were included in two independent experiments, $P < 0.0001$. Error bars: standard error of the mean (SEM). (**C**) Neutrophil killing assays. For this assay, differentiated HL-60 cells (dHL-60) were stimulated with PMA for 30 min and then incubated with WT, $\Delta 0362$, and $C\Delta 0362$ cells at MOI of 1:2 for 1 h. Following incubation, dead spirochetes were stained with propidium iodide; the number of living spirochete cells was enumerated using a Petroff-Hausser counting chamber. Total 12 biological samples were included in three independent experiments, $P < 0.0001$. Error bars: standard error of the mean (SEM). (**D**) Neutrophil killing protection assays using N0362 proteins. For this assay, dHL-60 cells were incubated with N0362 for 30 min prior to activation with PMA. After activation, the cells were incubated with $\Delta 0362$ at MOI of 2 for 1 h. Following incubation, dead spirochetes were stained with propidium iodide; the number of living spirochete cells was enumerated using a Petroff-Hausser counting chamber. $N_0$, N0362 (23–204 aa); $N_1$, N0362 with the Big_1 domain (23–114 aa); and $N_2$, N0362 with the Big_2 domain (115–204 aa). The readouts are expressed as the mean ± SEM. The data were analyzed by one-way ANOVA followed by Tukey's multiple comparison. Total 12 biological samples were included in three independent experiments, $P < 0.0001$. Error bars: standard error of the mean (SEM). (**E–H**) N0362 binds to dHL-60. For this study, HL-60 cells or DMSO differentiated HL-60 cells (dHL-60) were incubated with different concentrations of Alexa Fluor 488-labeled N0362 proteins, ranging from 0 to 30 μg as indicated, washed, and then subjected to flow cytometry. The gating cells strategy was based on FSC-A/SSC-A profile, followed by selecting single cells population (FSC-A/FCH-H), and subsequently detecting fluorescently labeled cells (Alexa Fluor 488-A/PE-A). The binding affinities were presented as bar graphs (**E, G**) and histograms (**F, H**). Total six biological samples ($n = 6$) were included in two independent experiments. Error bars: standard error of the mean (SEM). Source data are available online for this figure.

were carried out using HL-60 cells which were differentiated into neutrophil-like cells with DMSO and then stimulated with phorbol myristate acetate (PMA), a potent activator of neutrophils (Lagast et al, 1984; Suzuki and Lehrer, 1980). The results showed that without PMA stimulation, three *T. denticola* strains (WT, $\Delta 0362$, and $C\Delta 0362$) had nearly 100% survival rates. With PMA stimulation, while WT had a near 100% survival rate after 1-h incubation, the survival rate of $\Delta 0362$ was decreased to 42%, which was restored to 96% in $C\Delta 0362$ (Fig. 6C), indicating that TDE0362 can protect *T. denticola* from neutrophils killing. To corroborate this observation, a protection assay was conducted using three different recombinant proteins, including the full-length N0362 (N0, 23–204 aa), the one with the Big_1 domain (N1, 23–114 aa), and the one with the Big_2 domain (N2, 115–204 aa). The result showed that the $\Delta 0362$ mutant was protected by the treatment and its average survival rate in PMA-stimulated HL60 cells increased to 98%, 88% and 77% by N0, N1, and N2, respectively (Fig. 6D). Impairment of neutrophil chemotaxis and killing suggests that N0362 might bind to neutrophils. To determine if this is the case, flow cytometry analysis was performed using Alexa Fluor 488-

labeled N0362. The results showed that N0362 bound to DMSO-differentiated HL60 (dHL-60) cells in a dose-dependent manner (Fig. 6E,F). By contrast, N0362 showed a very weak binding to non-differentiated HL-60 cells (Fig. 6G,H). Immunofluorescence staining and flow cytometry analyses also showed that N0362 bound to human neutrophils in a dose-dependent manner (Appendix Fig. S12). Collectively, these studies indicate that N0362 binds to and inhibits neutrophils chemotaxis and killing.

## N0362 binds to FPR1 and CXCR1 receptors

N0362 binds to human neutrophils (Fig. EV4) and dHL-60 cells (Fig. 6E,F); however, the receptor(s) to which it binds remains unknown. To address this question, co-IP assays were first conducted using dHL-60 cell lysates and His-tagged N0362 protein. The precipitated samples were subjected to immunoblots probed against monoclonal antibodies that recognize four chemokine receptors, including FPR1, CXCR1, CXCR2, and C5αR (CD88) (Holmes et al, 1991; Mayadas et al, 2014; Murphy and Tiffany, 1991; Thomas et al, 1990). The co-IP result showed that FPR1, CXCR1/CXCR2, but not

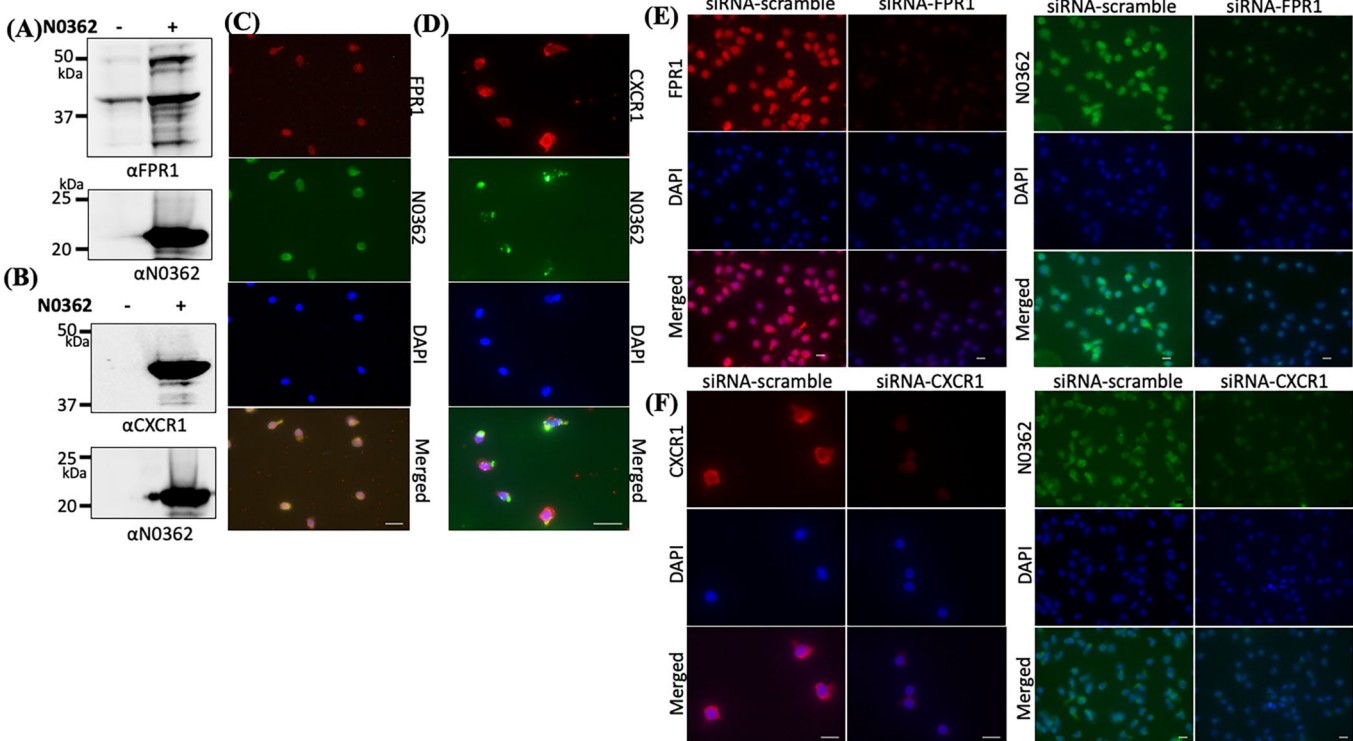

**Figure 7. N0362 binds to FPR1 and CXCR1 receptors.**

(A, B) co-IP assays. For this experiment, dHL-60 cell lysates were incubated with or without His-tagged N0362 protein (10 μg) for 3 h at 4 °C. Then, 5 μl Ni-NTA resin slurry was added to the mixture and further incubated at 4 °C for overnight. Following the incubation, the mixture was washed five times with PBS containing 0.05% Tween 20. The final pellets were suspended in Laemmli sample buffer, boiled for 5 min, and then briefly centrifuged to collect the supernatants, which were then subjected to immunoblotting analysis probed against different antibodies, including mAb-N7, a monoclonal antibody that recognizes N0362, anti-FPR1 (αFPR1), anti-CXCR1 (αCXCR1), anti-CXCR2 (αCXCR2), anti-CD11b (αCD11b), and anti-C5αR (αC5αR, Fig. EV5). (C, D) Immunofluorescence analysis of dHL-60 cells. The cells were incubated with Alexa Fluor 488-labeled N0362 proteins for 2 h, stained with αFPR1 or αCXCR1, followed by staining with Alexa Fluor 594. The micrographs were captured under fluorescence microscopy using FITC, TRITC, and DAPI emission filters, and the images were merged. The scale bars represent 20 μm. (E, F) Downregulation of FPR1 or CXCR1 expression using siRNA impairs the binding of N0362 to dHL-60 cells. For this study, dHL-60 cells were treated using siRNA specific to either (E) FPR1 or (F) CXCR1 for 96 h, followed by immunofluorescence analysis as above described. The scale bars represent 20 μm. Source data are available online for this figure.

C5αR, were precipitated along with N0362 (Figs. 7A,B and EV5), suggesting that N0362 binds to FPR1 and CXCR1/CXCR2, the receptors that are mainly recognized by fMLF and IL-8 (also known as CXCL8), respectively (Liu et al, 2020; Osei-Owusu et al, 2019; Russo et al, 2014; Zhuang et al, 2022). Of note, two major FPR1 bands were detected (Fig. 7A), which might be its heterogeneous glycosylated forms (Gao et al, 1993). To substantiate the co-IP result, immuno-fluorescence staining analysis was conducted, which revealed that fluorescence labeled N0362 colocalized with FPR1 and CXCR1 on dHL-60 cell surfaces (Fig. 7C,D). To further confirm the binding specificity, short interfering RNA (siRNA) specific to either FPR1 or CXCR1 were utilized to silence FPR1 or CXCR1 expression in dHL-60 cells, followed by immunofluorescence and immunoblotting analysis. The results showed that the binding of N0362 to dHL-60 cells was significantly reduced when the expression of two receptors was knocked down by either siRNA-FPR1 or siRNA-CXCR1. By contrast, scramble siRNA had no impact on the binding of N0362 to dHL-60 cells and the expression of FPR1 and CXCR1 (Figs. 7E,F and EV5D–F). Collectively, these results demonstrate that N0362 binds to neutrophils mainly through the FPR1 and CXCR1 receptors.

## N0362 binds to CXCR1 with high affinity and blocks IL-8 stimulated calcium mobilization

To further validate the binding between N0362 and the two receptors, enzyme-linked immunosorbent assays (ELISA) were conducted using GST-tagged FPR1 and CXCR1. The result showed that N0362 bound to GST-CXCR1 and GST-FPR1, but not GST alone, in a dose-dependent manner (Appendix Fig. S9). Following the ELISA, surface plasmon resonance (SPR) analysis was carried out to measure the binding affinity between N0362 and CXCR1 (Fig. 8A,B). For this experiment, GST-CXCR1 was immobilized to CM5 sensor chips, which were used for measuring the binding affinity of N0362 proteins (ranging from 0.05 μM to 3 μM) diluted in the running buffer. The results showed that N0362 bound to GST-CXCR1 at a high association rate constant ($k_a = 4.95 \times 10^3$ M$^{-1}$ s$^{-1}$) and a low dissociation rate constant ($k_d = 8.57 \times 10^{-4}$ s$^{-1}$) (Fig. 8A), an indicative of high binding affinity between the two proteins. The obtained SPR results were fitted to one to two binding model for the kinetic study (Fig. 8B). The result showed that N0362 bound to CXCR1 with an average

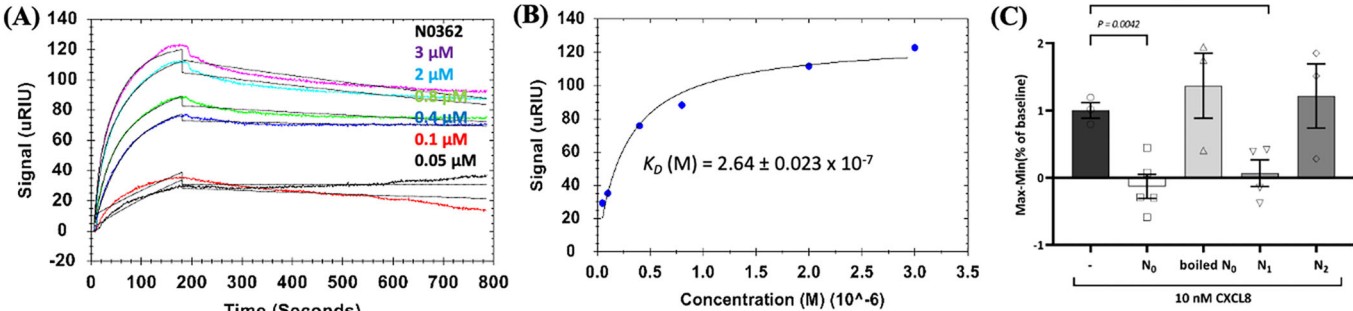

**Figure 8.  N0362 binds to CXCR1 with high affinity and blocks IL-8 stimulated calcium mobilization.**

(A, B) Determination of affinity constants between N0362 and CXCR1 using surface plasmon resonance (SPR). For this study, recombinant GST-CXCR1 proteins were coupled to CM5 sensors which were used for the binding experiment using six different concentrations of N0362 proteins (ranging from 0.05 μM to 3 μM) diluted in the running buffer. (A) Representative sensorgram of CXCR1-N0362 binding plots with kinetic fit (1:2 binding model). (B) Dose-dependent saturation binding of N0362 to CXCR1. The equilibrium disassociation constant $K_D$ ($k_d/k_a$) was calculated. (C) IL-8 reporter assays using ready-to-assay CXCR1 chemokine receptor cells (Chem-1 cells). For this experiment, Chem-1 cells were first treated with different recombinant N0362 proteins (5 μg) as labeled for 30 min and then stimulated with IL-8 (10 nM). The fluorescence intensity was monitored at Ex/Em = 340/510 nm and Ex/Em = 380/510 nm. The data are presented as the mean of relative fold change based on the fluorescence ratio of 340/380 nm ± standard error of the mean (SEM). The sample with only IL-8 is set as the baseline (onefold). $N_0$, N0362 (23–204 aa); $N_1$, N0362 (23–114 aa); and $N_2$, N0362 (115–204 aa). Five biological samples ($n = 5$) were included. Statistical analysis was determined using unpaired $t$ test. Source data are available online for this figure.

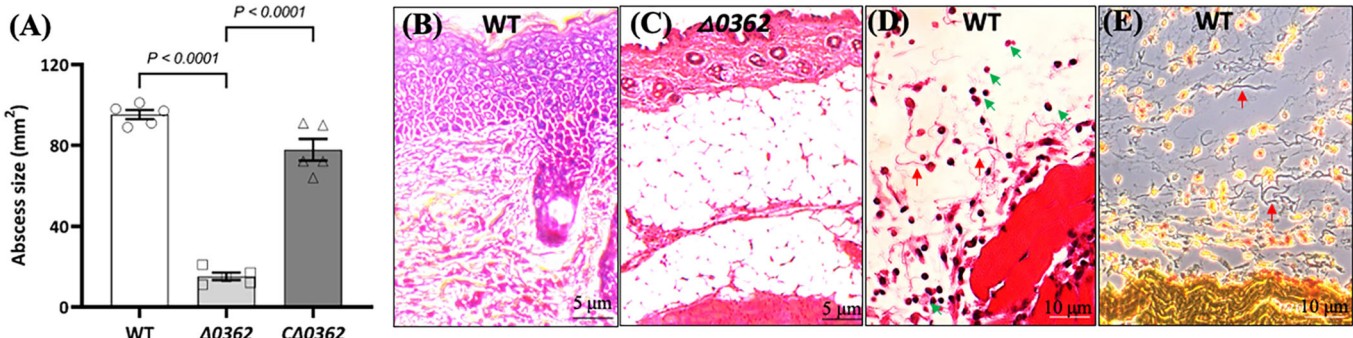

**Figure 9.  Assessing the role of TDE0362 in *T. denticola* virulence using a murine skin abscess model.**

This study was carried out by injecting $1 \times 10^9$ *T. denticola* cells, including WT, *Δ0362*, and *CΔ0362*, subcutaneously into each mouse. BALB/c mice (5 mice per bacterial strain) were used in this study. Ten days post-infection, the sizes of the abscesses were measured (A). The average sizes of the observed abscesses were calculated. The results are expressed as the mean ± SEM. $P < 0.0001$. The data were analyzed by one-way ANOVA followed by Tukey's multiple comparison. (B–E) Histological analysis of infected tissues using hematoxylin and eosin (H&E) staining (B–D) and Warthin Starry staining (E). Green arrows highlight inflammatory infiltrates; red arrows highlight *T. denticola* cells. Source data are available online for this figure.

equilibrium dissociation constant of $K_D = 264 ± 2.30$ nM. These results indicate that N0362 binds to CXCR1 with high affinity.

Although CXCR1 can be activated by many CXC chemokines, it is primarily activated by IL-8, which is essential for neutrophil chemotaxis (Mayadas et al, 2014; Russo et al, 2014). IL-8 binds to CXCR1, a G protein-coupled receptor (GPCR), triggering a transient increase in the concentration of intracellular calcium (Burg et al, 2015). Since N0362 binds to CXCR1 with high affinity, we speculated that it may function as an antagonist of IL-8 and prevent it from binding to CXCR1, thereby blocking neutrophils chemotaxis. To test this, a CXCR1-expressing cell line (chem-1 cells) was used to measure IL-8-induced calcium mobilization with/without N0362. For this experiment, chem-1 cells were pretreated with three different N0362 recombinant proteins as

labeled at a concentration of 5 μM for 10 min and then stimulated with IL-8 (10 nM). The results showed that N0 and N1, but not boiled N0 and N2, blocked the calcium mobilization induced by IL-8 (Fig. 8C). These results indicate that N0362 acts as an antagonist of IL-8 and blocks its binding to CXCR1 most likely through the N-terminal Big_1 domain.

## TDE0362 contributes to the virulence of *T. denticola*

The importance of TDE0362 in the virulence of *T. denticola* was further assessed in vivo using a previously reported mouse skin infection model (Bian et al, 2013; Kurniyati et al, 2013). For this study, each BALB/c mouse was subcutaneously injected with $1 \times 10^{10}$ *T. denticola* (WT, *Δ0362*, and *CΔ0362*) cells at its dorsal site. Ten days after the injections, mice were

sacrificed; the induced skin abscesses were measured and recorded as mm$^2$ in sizes. While demarcated subcutaneous abscesses were observed in all infected mice, the skin lesions induced by $\Delta 0362$ (15 mm$^2$, $n = 5$ mice) were significantly smaller than those induced by WT (95 mm$^2$, $n = 5$ mice) and $C\Delta 0362$ (78 mm$^2$, $n = 5$ mice) (Fig. 9A). Histological studies using H&E staining revealed that WT, but not the $\Delta 0362$ mutant, induced severe inflammation which was evident by a large amount of intrasinusoidal inflammatory infiltrates in the infected skin tissues (Fig. 9B,C). In addition, both H&E and Warthin Starry staining detected abundant intact helical-shaped bacterial cells (Fig. 9D,E), an indicative of living spirochetes, in the skin tissues infected by WT but not in those infected by the mutant. This study underscores the importance of TDE0362 in the virulence of *T. denticola*.

## Discussion

In this report, we functionally and structurally characterized TDE0362, a new virulence factor of *T. denticola* with unique domain composition, processing pattern, structure, pathogenic trait, and underlying molecular mechanism. TDE0362 contains two major functional units. Our initial hypothesis is that it is expressed as a zymogen and activated by cleaving its N-terminal unit (i.e., N0362) which acts as a pro-domain that inhibits its proteolytic activity. However, this hypothesis was rejected by the following evidence: (1) The full-length TDE0362 is active and shows toxicity to yeast cells (Fig. 1D), indicating that the N-terminal domain does not function as a pro-domain; (2) AlphaFold structural modeling and SEC-SAXS analysis demonstrate that the full-length TDE0362 forms a structure with an extended conformation and there is no interaction between N0362 and the C-terminal catalytic domain of TDE0362 (Appendix Figs. S5 and S6); and (3) N0362 forms a functional unit that blocks neutrophils chemotaxis via binding to FPR1 and CXCR1/2 receptors (Figs. 7, 8).

We sought to delineate the processing mechanism of TDE0362 and found that its cleavage and processing is complex and dynamic. Particularly, its cleavage pattern varies with growth phases (Appendix Fig. S10). Nevertheless, we were able to detect at least two cleaved products (F1 and F2) in the spent culture supernatants of *T. denticola* (Fig. 2). By using genetics and in vitro proteolytic assays alongside mass spectrometry, we found that F1 and F2 are the two cleavage products of PrtP. The cleavage occurs at Phe-132/Glu-133 and Phe-235/Glu236 (Fig. EV3) which is in line with the catalytic activity of PrtP as a phenylalanine specific protease (Ishihara et al, 1996; Rosen et al, 1999). PrtP form an outer membrane complex associated with Msp and two lipoproteins, PrcA and PrcB (Godovikova et al, 2010; Goetting-Minesky et al, 2021; Goetting-Minesky et al, 2013). Therefore, it is conceivable to reason that the cleavage of TDE0362 by PrtP occurs on *T. denticola* cell surface after it is secreted. Deletion of PrtP blocks the production of F2 but generates four high molecular weight products (Fig. 2), suggesting that there might be other proteases implicated in the cleavage of TDE0362 in the absence of PrtP. In addition to F1 and F2, we also detected several small N-terminal cleaved products in vitro proteolytic assays as well as in the spent culture supernatants (Fig. EV3), suggesting that they are secreted too, at least in part.

Identification of Sec/SPII cleavage site suggests that TDE0362 is cleaved and translocated to the periplasmic space of *T. denticola* cells via

the Sec-dependent secretion pathway; however, the mechanism by which TDE0362 is secreted into the extracellular milieu, such as the spent culture supernatant of *T. denticola*, remains unknown. Bacterial pathogens, in particular intracellular pathogens, have evolved to export and deliver toxic proteins using different secretion apparatuses such as T3SS, T4SS, T6SS, and T9SS (Costa et al, 2015; Song et al, 2022). However, the genes encoding these apparatuses are absent in the genome of *T. denticola* (Seshadri et al, 2004). Bacterial β-barrel assembly machinery (BAM) often works in liaison with the Sec-dependent secretion pathway forming T5SS autotransporter (Rodriguez-Alonso et al, 2020). The BAM system was identified and characterized in spirochete *T. pallidum* and *Borrelia burgdorferi* (Desrosiers et al, 2011; Lenhart et al, 2012). A similar system was found in *T. denticola* (Anand et al, 2013; Seshadri et al, 2004). Thus, it is possible that *T. denticola* uses the BAM system exporting TDE0362. Alternatively, *T. denticola* may have evolved other protein secretion systems. For example, a new type of secretion apparatus was discovered in *Photorhabdus luminescens*, an insect bacterial pathogen. This secretion system consists of tripartite ABC-type toxin complexes which form a syringe-like injection apparatus (Gatsogiannis et al, 2013; Meusch et al, 2014). We interrogated the genome of *T. denticola* and identified a similar system. Thus, it is possible that *T. denticola* has evolved to assemble a similar apparatus to secrete and deliver its virulence factors such as TDE0362. We are currently investigating this secretion apparatus and its role in the secretome of *T. denticola*.

Bacterial Ig-like (Big) domains are present in both Gram-negative and Gram-positive pathogens but their functions have been characterized only in a few bacterial species (Chatterjee et al, 2021). For example, the intimin protein of enteropathogenic *E. coli* (EPEC), a virulence factor that mediates the intimate bacterial host-cell interaction and subsequent pedestal formation, contains four extracellular Big domains (Kelly et al, 1999; Weikum et al, 2020). LigA and LigB, two essential virulence factors of *Leptospira interrogans*, contain 12-13 tandem Big domains which bind to host extracellular matrices (ECM) and complement factors (Choy et al, 2011; Matsunaga et al, 2003). *Yersinia pseudotuberculosis* invasin D binds to the Fab fragment of host-derived IgG and IgA through its C-terminal adhesion module with a Big domain (Sadana et al, 2018). The β protein of Group B *Streptococcus* binds to host ECM via carcinoembryonic antigen-related cell adhesion molecules (CEACAM) receptors (van Sorge et al, 2021). N0362 possesses two Big domains sharing 26.1% and 27.8% sequence identity with the invasin D and the β protein, respectively. In this report, we provide several lines of evidence that N0362 inhibits neutrophils chemotaxis through interactions with FPR1 and CXCR1/2 receptors. We first demonstrate that N0362 binds to human neutrophils and DMSO-differentiated HL-60 (dHL-60) cells and potently inhibits neutrophils chemotaxis in a dose-dependent manner (Figs. 6 and EV4). Second, co-IP and IFA assays in context with siRNA knockdown further reveal that N0362 binds to dHL-60 cells through FPR1 and CXCR1/2 receptors (Fig. 7), which are further substantiated by ELISA and SPR analyses showing that N0362 binds to the two receptors with high affinity (Fig. 8; Appendix Fig. S9). Finally, IL-8 reporter assays indicate that N0362 blocks IL-8 stimulated calcium mobilization (Fig. 8). To the best of our knowledge, this is the first report that a protein with Big domains binds to FPR1 and CXCR1/2 and inhibits neutrophils chemotaxis. The results reported here provide new insights into understanding the role of Big domains which are present in a large

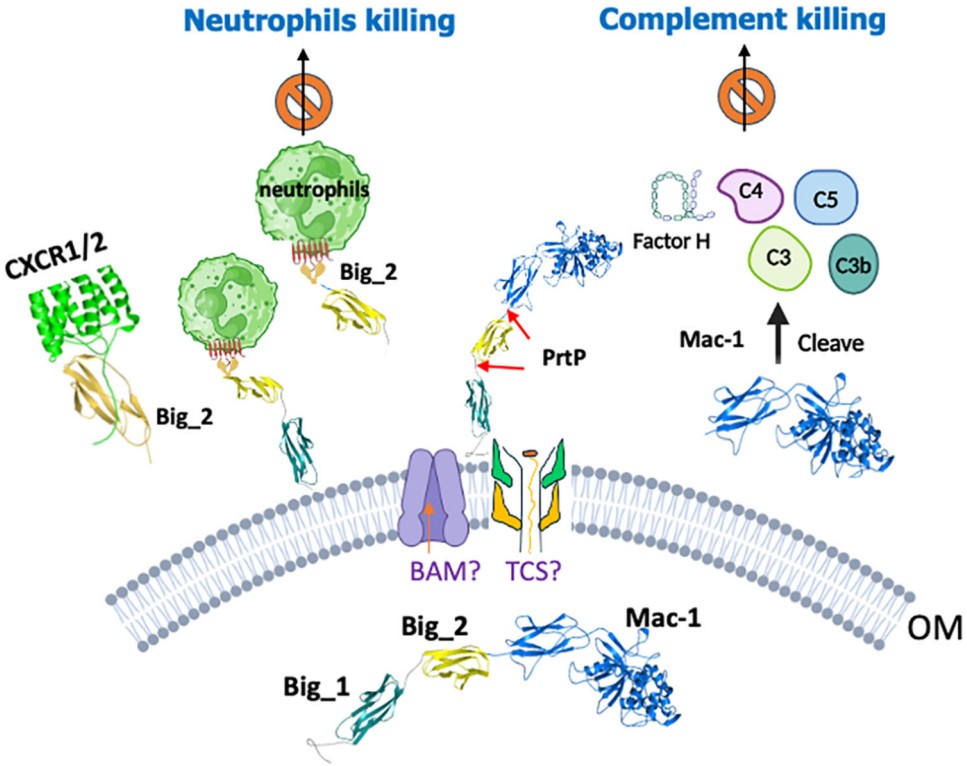

**Figure 10. A model illustrates the processing and function of TDE0362.**

TDE0362 is expressed, secreted presumably through either bacterial β-barrel assembly machinery (BAM) or tripartite ABC-type toxin complex system (TCS), and then cleaved by PrtP, generating at least two functional units: N0362 with two bacterial Ig-like domains (Big_1 and Big_2) and C0362 with a Mac-1 domain. N0362 binds to FPR1 and CXCR1/2 receptors and blocks neutrophil chemotaxis and activation. C0362 is secreted into extracellular milieu and functions as a cysteine protease that disarms the complement system via degrading key complement factors such as C3 and C4. Collectively, this dual domain protein annihilates both the complement system and neutrophils and renders bacteria protection from these two innate defense mechanisms.

array of bacterial proteins, most of which have not been functionally characterized.

The T-Mac domain of C0362 is a homolog of IdeS and gasMac-1 and functions as a cysteine protease; however, it possesses unique biochemical and structural features. First, IdeS and gasMac-1 cleave human IgG but T-Mac does not. Instead, T-Mac cleaves human complement factors C3, C3b, C4, and FH (Fig. 3). These substrate preferences may be attributed to differences in their secondary structures (Fig. 5D–F). Sudol et al recently solved the crystal structure of IdeS in complex with IgG-1 Fc and found that the PBL of IdeS serves as the primary site of interaction with IgG (Sudol et al, 2022). Our structural data reveals that the PBL of T-Mac differs from that of IdeS (Fig. 5). Further studies will be required to identify the mechanism of substrate selectivity in T-Mac. Second, we found that T-Mac was sensitive to oxygen and rapidly became inactivated by oxidation of Cys-412. Notably, this oxidation was observed in our crystal structure (Fig. 5D; Appendix Fig. S8). Although the oxidation of Cys-412 could also arise from radiation damage during data collection, there are no other obvious signs of global radiation damage visible (e.g., decarboxylation of nearby Glu-583 or oxidation of Cys-279, Cys-526, Cys-565, or Cys-578). To our knowledge, this high sensitivity to oxidation has not been reported in other bacterial cysteine proteases, particularly in IdeS and gasMac-1. This feature is in line with

the obligate anaerobic nature of *T. denticola*. While the exact mechanism associated with this oxygen sensitivity is unclear, we propose that the enhanced oxygen sensitivity may be tied to a more open active site cleft in T-Mac (Appendix Fig. S11). Lastly, we found that T-Mac cleaved human C3b between Glu-207 and Gly-208. Cleavage at Glu-207 is interesting, given the similarity between T-Mac and IdeS. The S2 subsite in papain superfamily of enzymes is recognized as the specificity determining site for cleavage and often accommodates hydrophobic amino acids (Dutta et al, 2016; Petushkova et al, 2022). Based on our AlphaFold2 model of full-length TDE0362, the S2 subsite pocket of T-Mac is indeed primarily hydrophobic, lined by Phe-549, Tyr-484, Ala-562, Ile-485, and Ile-523. However, the distal end of this pocket contains two hydrophilic residues, Glu-486 and Thr-630, which may facilitate interactions with Glu-207. A substrate side chain larger than glycine, alanine, or proline would likely clash with the loop spanning Asn-479 through Gly-483, suggesting that T-Mac likely prefers small amino acids after the cleavage position.

In addition to the aforementioned differences in its biochemical feature, C0362 also shows some structural differences compared to IdeS and gasMac-1. The first difference is that C0362 contains an extra domain (N-120, residues 211–324) which forms an Ig-like fold structure (Fig. 4). To elucidate the role of N-120, we attempted the expression and purification of the Mac-1 domain (T-Mac)

only. As C0362 is highly expressed under recombinant conditions, we were surprised that the removal of N-120 resulted in completely insoluble protein. Combining this result with the structural similarity of N-120 to the PapD-like family of chaperones suggests that this domain may be involved in stabilizing or enhancing the solubility of the Mac-1 domain. Alternatively, the conserved positioning of N-120, relative to the Mac-1 domain, and the Ig-like fold suggests the possibility that this domain may be involved in substrate recognition and binding. Additional experimentation may assist in elucidating the role of N-120 in the function of C0362. The other structural difference is that C0362 forms monomers rather than dimers as reported in IdeS and gasMac-1 (Agniswamy et al, 2006). Dimerization of IdeS is mediated by the antiparallel β-strands $\beta_2$ and $\beta_3$. Notably, in the crystal structure of IdeS complexed with human IgG-1 Fc (PDB entry 8A47), the dimerization reported by Agniswamy and colleagues is not present. In C0362, we indeed observed a propensity for dimerization during SEC, although the fraction only accounted for ~10% of the total material. However, when we analyzed neighboring units within the crystal structures of C0362 (both SeMet and wild-type), no obvious dimerization interface was observed. When we attempted to generate the IdeS/gasMac-1 dimer by aligning C0362 to the two molecules, an interface suitable for dimerization was not possible. This is unsurprising given that antiparallel β-strands are one of the secondary structural differences we noted (Fig. 5). Thus, although we observed some dimerization of C0362 during SEC, whether the minor dimerization of C0362 is physiologically relevant and the actual site of C0362 dimerization is unclear.

Using TDE0362 as a query, we searched bacterial genomes in the NCBI database and identified 156 hits with Ig-like domains from 24 different bacterial genus and 178 hits of cysteine proteases from 10 bacterial genus. Phylogenetic analysis revealed that the majority of these hits belong to the family of Treponemataceae which constitute a large clade that are rooted from the CFB (Cytophaga, Fusobacterium, and Bacteroides) group bacteria from human oral and gut microbiota (Appendix Fig. S12), suggesting that *T. denticola* may acquire TDE0362 through a horizontal gene transfer. In addition to *T. denticola*, we also found TDE0362 homologs in other oral *Treponema* species such as *T. socranski* and *T. vincentii*. Sequence alignment showed that those homologs all contain a conserved catalytic motif of cysteine protease (Appendix Fig. S13), suggesting that a similar mechanism may be adopted by other oral *Treponema* species to evade host innate immune defense. The study in this report opens an avenue to investigate the role of Ig-like domains and cysteine proteases in bacterial virulence.

Building upon the above discussion and the results shown in this report, a model is proposed to summarize the role of TDE0362 and its underlying molecular mechanism (Fig. 10). As a dual domain protein, TDE0362 is secreted and then cleaved by PrtP releasing two functional units. While the C-terminal unit (C0362) is secreted and functions as a cysteine protease that disarms the complement system via degrading key complement factors such as C3 and C4, the N-terminal functional unit (N0362) with two Big domains binds to FPR1 and CXCR1/2 receptors with high affinity and blocks neutrophil chemotaxis and activation most likely through competition inhibition, thereby annihilating both the complement system and neutrophils and rendering bacteria protection from these two innate defense mechanisms.

# Methods

## Reagents and tools table

| Reagent/resource | Reference or source | Identifier or catalog number |
|---|---|---|
| **Experimental models** | | |
| HL-60 cell line | ATCC | HL-60 CCL-240 |
| HEK293T cell line | ATCC | 293T CRL-3216 |
| SP2/0-Ag14 cell line | ATCC | SP2/0-Ag14 CRL-1581 |
| Human neutrophils | Isolated from humans | HM20015900 |
| Mouse neutrophils | Isolated from C57BL/6J | AD10001778 |
| CXCR1 receptor cell line | Millipore-Sigma | HTS001RTA |
| C57BL/6J mice | Jackson Lab | Strain#: 000664 |
| BALB/c mice | Jackson Lab | Strain #: 000651 |
| *Treponema denticola* Td35405 | ATCC | *Treponema denticola* (ex Flügge) |
| *Treponema denticola Δ0362* | This study | |
| *Treponema denticola CΔ0362* | This study | |
| NEB 5-alpha | New England Biolabs | |
| BL21-CodonPlus (DE3)-RIL | Agilent Technologies | |
| Lemo21(DE3) | New England Biolabs | C2987H |
| *Saccharomyces cerevisiae* strain W303 | Fan et al, 1996 | 230245 |
| | | C2528J |
| **Recombinant DNA** | | |
| CXCR1 siRNA | Santa Cruz Biotechnology | sc-40026 |
| FPR1 siRNA | | sc-40121 |
| scramble siRNA | | sc-37007 |
| pGEM-T Easy Vector | Promega | A137A |
| pGEX-6P-1 Vector | Cytiva | 28954648 |
| pQE80L | Qiagen | 32943 |
| pET101/D-TOPO | Invitrogen | K10101 |
| p426GDP | Mumberg et al, 1995 | – |
| p426TEF | Mumberg et al, 1995 | – |
| pYES2/NTA | Invitrogen | V82520 |
| p426GDP::FL0362 | This study | SH30228.01 |
| pYES2/NTA::FL0362 | This study | |
| pQE30::FL0362 | This study | |
| pET100/D::N0362 | This study | |
| p426TEF::N0362 | This study | |
| pcDNA6.2 C-EmGFP-GW/TOPO::N0362 | This study | |
| pQE80L::N1-N0362 | This study | |
| pQE80L::N2-N0362 | This study | |

| Reagent/resource | Reference or source | Identifier or catalog number |
|---|---|---|
| pET101/D::C0362 | This study | |
| p426GDP::C0362 | This study | |
| pYES2/NTA::C0362 | This study | |
| pcDNA6.2 C-EmGFP GW/ TOPO::C0362 | This study | |
| pcDNA6.2 C-EmGFP GW/ TOPO::C0362 | This study | |
| C412A | This study | |
| pGEX-6P-1::CXCR1 | This study | |
| pGEM-T easy::TDE0362::ermB | This study | |
| pGEM-T easy:: tap1-TDE0362 | This study | |
| **Antibodies** | | |
| CD11b | Abcam | ab8878 |
| CXCR1 | Santa Cruz Biotechnology | SC-7303 |
| CXCR2 | Santa Cruz Biotechnology | SC-7304 |
| C5aR | Abcam | ab234757 |
| FPR1 | Invitrogen | PA1-41398 |
| Alexa Fluor 647 Mouse Anti-Human CD66b | 561645 | BD Biosciences |
| Goat anti-Rabbit IgG (H+L) Cross Adsorbed | Invitrogen | A-11012 |
| Secondary Antibody, Alexa Fluor 594 | Invitrogen | A-11006 |
| Goat anti-Rat IgG (H+L) Cross-Adsorbed | This study | |
| Secondary Antibody, Alexa Fluor 488 | This study | |
| PolyAb TDE0362 | This study | |
| mAb-C3 TDE0362 | | |
| mAb-N7 TDE0362 | | |
| **Oligonucleotides and other sequence-based reagents** | | |
| Primers are listed in Table EV1 | IDT DNA | – |
| **Chemicals, enzymes, and other reagents** | | |
| Alexa Fluor™ 488 Protein Labeling Kit | Thermo Fisher Scientific | A10235 |
| IMDM | Cytiva | SH30228.01 |
| DMEM | Gibco | 11965092 |
| Normocin | InvivoGen | ant-nr-05 |
| Erythromycin | Sigma-Aldrich | E5389 |
| Gentamicin | Sigma-Aldrich | G3632 |
| Chloramphenicol | Sigma-Aldrich | C0378 |
| Ampicillin | Sigma-Aldrich | A9393 |
| Penicillin/streptomycin | Gibco | 10378016 |
| Flamingo Fluorescent Protein Gel Stain | Bio-Rad | 1610490 |

| Reagent/resource | Reference or source | Identifier or catalog number |
|---|---|---|
| TRI Reagent | Sigma-Aldrich | T9424 |
| TURBO DNase | Invitrogen | AM2238 |
| SuperScript IV VILO | Invitrogen | 11756050 |
| FirstChoice RLM-RACE Kit | Invitrogen | AM1700 |
| EcoRI | New England Biolabs | R3101S |
| BamHI | New England Biolabs | R3136S |
| XhoI | New England Biolabs | R0146S |
| Ni-NTA agarose | Qiagen | 30210 |
| Bio-Rad Protein Assay Dye | Bio-Rad | 5000006 |
| Proteinase K | Sigma-Aldrich | 1.24568 |
| PMSF | Sigma-Aldrich | 52332 |
| Lipofectamine 3000 Transfection Reagent | Invitrogen | L3000015 |
| Lipofectamine LTX with PLUS Reagent | Invitrogen | 15338100 |
| Percoll | Sigma-Aldrich | P4937 |
| ProLong Gold Antifade Mountant with DNA Stain DAPI | Invitrogen | P36931 |
| Glutathione Agarose | Thermo Scientific | 16100 |
| Warthin-Starry stain kit | Abcam | ab150688 |
| Human serum | BIOIVT | HUMANSRM-0101711 |
| IgG | Sigma-Aldrich | 56834 |
| IgA | Sigma-Aldrich | I4036 |
| C3 | Complement Technology | A113 |
| C3b | Complement Technology | A114 |
| C4 | Complement Technology | A105 |
| C5 | Complement Technology | A120 |
| Factor H | Complement Technology | A137 |
| C4bp | Complement Technology | A109 |
| FPR1 from wheat germ | Abnova | H00002357-P01 |
| CXCRI from wheat germ | Abnova | H00003577-P01 |
| CXCR1 from *E.coli* | This study | – |
| N0362 | This study | – |
| FL0362 | This study | – |
| IL-8 | Novus Biological | NBC1-18459 |
| **Software** | | |
| GraphPad | GraphPad | |
| Image Lab | Bio-Rad | |
| Zen LE | Zeiss | |
| TraceDrawer 1.8 | TraceDrawer | |
| Clustal Omega | EMBL | |
| CCP4i2 | Collaborative Computational Project No. 4 | |

| Reagent/resource | Reference or source | Identifier or catalog number |
|---|---|---|
| Coot | CCP4i2 | v. 0.9.8.92 |
| Phaser | CCP4i2 | v. 2.8.3 |
| Aimless | CCP4i2 | v. 0.7.7 |
| DIALS | CCP4i2 | v. 2.2.10 |
| Xia2 | CCP4i2 | v. 0.6.475 |
| Gesamt | CCP4i2 | v. 1. 1.6 |
| Molprobity | PHENIX | |
| PHENIX | | v. 1.2.1_4487 |
| Blue-Ice | Advanced Photon Source (APS) Argonne National Laboratory | |
| PyMOL | | v. 2.5.5 |
| ColabFold | Google Colab | |
| **Other** | | |
| Petroff-Hausser Counting Chambers | Hausser Scientific | 3900 |
| Spectra/Por 3 Dialysis Tubing 3.5 | Spectrum | 132724 |
| Superdex 200 Increase 10/300 GL | Cytiva | 28990944 |
| High binding microplate | Greiner Bio-One | 655061 |
| Reichert SR7500DC instruments | Reichert | – |
| Zeiss LSM700 laser scanning confocal microscope | Zeiss | – |
| Poly-L-Lysine Coated German Glass Cover Slips | Electron Microscopy Sciences | 72292-01 |
| BD FACSCanto II | BD Biosciences | – |
| BD FACSymphony A1 Cell Analyzer | BD Biosciences | – |

## Methods and protocols

### Ethics statement

All animal experimentation was conducted following the NIH guidelines for housing and care of laboratory animals and performed in accordance with the Virginia Commonwealth University (VCU) institutional regulations after review and approval by the Institutional Animal Care and Use Committees [IACUC approval number: AD10001778]. Human neutrophils, serum, and blood samples were collected from healthy donors in accordance with the institutional review board (IRB) from the University of Buffalo (approval number: 030-529353) and VCU (approval number: HM20015900).

### Bacterial strains, culture conditions, and oligonucleotide primers

*T. denticola* ATCC 35405 (wild type, WT) was used in this study (Seshadri et al, 2004). Bacterial cells were grown in tryptone-yeast

extract-gelatin-volatile fatty acids-serum (TYGVS) medium at 37 °C in an anaerobic chamber in the presence of 90% nitrogen, 5% carbon dioxide, and 5% hydrogen (Ohta et al, 1986). *T. denticola* isogenic mutants were grown with erythromycin (50 μg/ml) and/or gentamicin (20 μg/ml). The growth rates of *T. denticola* in TYGVS medium or a serum-free TYGV medium supplemented with L-cysteine (1 g/liter) were monitored via direct counting using a Petroff-Hausser counting chamber (Hausser Scientific, Horsham, PA), as previously described (Kurniyati et al, 2013). *Escherichia coli* DH5α strain (New England Biolabs, Ipswich, MA) was used for DNA cloning; BL21-CodonPlus (DE3)-RIL (Agilent, Santa Clara, CA) and Lemo21(DE3) (New England Biolabs) were used for preparing recombinant proteins. *E. coli* strains were grown in lysogeny broth (LB) supplemented with appropriate concentrations of antibiotics for selective pressure as needed: erythromycin (400 μg/ml), kanamycin (50 μg/ml), chloramphenicol (50 μg/ml), or ampicillin (100 μg/ml). The oligonucleotide primers for PCR and RT-PCR used in this study are listed in Table EV1. These primers were synthesized in IDT (Integrated DNA Technologies, Coralville, IA).

### Human cell lines and culture conditions

All the cell lines were maintained in appropriate media at 37 °C, 5% $CO_2$ in a standard tissue culture incubator. HL-60 CCL-240 cell line was purchased from the American Type Culture Association (ATCC, Manassas, VA). HL-60 cells are promyeloblasts isolated from the peripheral blood of a female with acute promyelocytic leukemia and commonly used to study neutrophil functions (Collins et al, 1977). HL-60 cells were cultured in Iscove's Modified Dulbecco's Medium (IMDM; Cytiva, Marlborough, MA) supplemented with 20% (v/v) fetal bovine serum (FBS) and 100 μg/ml normocin (InvivoGen, San Diego, CA). HL-60 cells were differentiated into neutrophil-like cells with 1.3% v/v DMSO for 5 days (Millius and Weiner, 2010). Cell differentiation was assessed microscopically or by using immunoblotting analysis for CD11b, a biomarker for neutrophils (Carrigan et al, 2005). HEK293T cells were maintained in DMEM containing 10%(v/v) FBS, 2 mM glutamine, and 100 g/ml penicillin/streptomycin (Gibco, Waltham, MA) at 37 °C in a 5% $CO_2$ incubator.

### Proteomic analysis of *T. denticola* culture supernatants

*T. denticola* culture supernatants were prepared as previously described with modifications (Kurniyati et al, 2013; Link et al, 2009). Briefly, 200 ml of *T. denticola* cells were cultured in a serum-free TYGV medium supplemented with L-cysteine (1 g/liter) until reaching the mid-log phase ($5 \times 10^8$ cells/ml). The cultures were centrifuged at 3000×*g* for 15 min at 4 °C to remove bacterial cells; spent culture supernatants (SCSs) were filtered through 0.2 μm-pore-size filters to further remove remaining bacterial cell debris. The filtered SCSs were then centrifuged using ultracentrifugation at 28,000×*g* for 3 h at 4 °C to remove the membrane vesicles. The obtained supernatants were dialyzed in 50 mM Tris-HCl buffer (pH 7.8) at 4 °C overnight using 3.5 kDa molecular weight cut-off Spectra/Por® dialysis membranes (Spectrum Laboratories, Rancho Dominguez, CA) and further concentrated to 10 ml. The concentrated SCSs were then precipitated using trichloroacetic acid (TCA). The resulting samples were

centrifuged at 20,000×g for 30 min at 4 °C and then washed three times with ice-cold acetone. The final pellets were air-dried and resuspended in the appropriate sample buffers and then subjected to SDS-PAGE and two-dimensional (2D) gel electrophoresis, followed by Flamingo fluorescence gel staining analysis (Bio-Rad Laboratories, Hercules, CA) and matrix-assisted laser desorption/ionization-time of flight mass spectrometry (MALDI-TOF-MS) analysis at the University of Pittsburgh Health Sciences Mass Spectrometry Core.

## RNA preparations, Reverse transcription-PCR (RT-PCR), and 5′-rapid amplification of cDNA end (RACE)

RNA isolation was performed as previously described (Kurniyati et al, 2022b). Briefly, *T. denticola* cells were harvested at the mid-logarithmic phase (~$5 \times 10^8$ cells/ ml). Total RNA was extracted using TRI reagent (Sigma-Aldrich, St. Louis, MO), following the manufacturer's instructions. The resultant samples were treated with Turbo DNase I (Thermo Fisher Scientific, Waltham, MA) at 37 °C for 2 h to eliminate genomic DNA contamination. The resultant RNA samples were re-extracted using acid phenol-chloroform, then precipitated in isopropanol, and finally washed once with 70% ethanol. The RNA pellets were resuspended in RNase-free water. cDNA was generated from the purified RNA (1 μg) using SuperScript IV VILO cDNA synthesis kit (Thermo Fisher Scientific). 5′RACE analysis was performed using the First Choice RLM-RACE kit (Ambion, Austin, TX), according to the manufacturer's protocol. The primers for RT-PCR and 5′RACE are listed in Table EV1.

## β-galactosidase activity assay

A fragment spanning from nucleotides −162 to +41 of *TDE0369*, the first gene in the operon (Appendix Fig. S1), was PCR amplified with primer pair $P_{35}/P_{36}$, generating a fragment with engineered EcoRI and BamHI cut sites at the 5′ and 3′ ends, respectively. The obtained fragment was in-frame fused to the promoterless *lacZ* gene in the pRS414 plasmid (a gift from R. Breaker, Yale University), creating $pRS_{TDE0369}$. The resultant plasmid was transformed into *E. coli* DH5α. β-galactosidase activity was measured and expressed as average Miller units of triplicate samples from three independent experiments as previously described (Kurniyati et al, 2019).

## Construction of a *TDE0362* deletion mutant and its isogenic complemented strain

The TDE0362::ermB plasmid (Appendix Fig. S1) was constructed to replace the entire open reading frame of *TDE0362* with a previously documented erythromycin B resistant cassette (*ermB*) (Goetting-Minesky and Fenno, 2010). To construct TDE0362::ermB, the *TDE0362* upstream region, *ermB*, and the *TDE0362* downstream region were PCR amplified with primer pair $P_{37}/P_{38}$, $P_{39}/P_{40}$, and $P_{41}/P_{42}$, respectively. The upstream region and *ermB* were PCR ligated with primers $P_{37}/P_{40}$. The resulted fragment was then fused together with the downstream region with primers $P_{37}/P_{42}$. The resulting PCR fragments were cloned into pGEM-T easy vector (Promega), generating TDE0362::ermB. To delete *TDE0362*, TDE0362::ermB was transformed into *T. denticola* WT competent cells via heat shock,

as previously described (Kurniyati and Li, 2021). The deletion was confirmed by PCR and immunoblotting analyses. The resultant mutant was designated as Δ*0362*. To construct TDE0362$^{com}$, the *TDE0362* upstream region and a previously documented gentamicin resistant cassette (*aacC1*) (Bian et al, 2012) were PCR amplified with primer pair $P_{37}/P_{43}$ and $P_{44}/P_{45}$, respectively, and then fused together with primer pair $P_{37}/P_{45}$, generating *upstream-aacC1*. The *tap1* promoter (Limberger et al, 1999) and the full length *TDE0362* gene along with its downstream region were amplified with $P_{46}/P_{47}$ and $P_{48}/P_{42}$, respectively, and then fused together with primers $P_{46}/P_{42}$, generating *tap1-TDE0362*. The obtained DNA fragments, *upstream-aacC1* and *tap1-TDE0362* were cloned into pGEM-T easy vector and fused by engineered cut sites XhoI as illustrated in Appendix Fig. S2, generating the vector TDE0362$^{com}$. To restore the expression of TDE0362, TDE0362$^{com}$ was transformed into the Δ*0362* strain via heat shock. The resultant complemented clones were confirmed by PCR and immunoblotting analyses. One complemented clone (*CΔ0362*) was selected for further characterizations. The primers for constructing these two plasmids are listed in Table EV1.

## Electrophoresis and immunoblotting analyses

Sodium dodecyl sulfate-polyacrylamide gel electrophoresis (SDS-PAGE) and immunoblotting analyses were carried out as previously described (Kurniyati et al, 2013). For immunoblotting analysis, *T. denticola* cells were harvested in the mid-logarithmic (log) phase (~$5 \times 10^8$ cells/ml). Equal amounts of whole-cell lysates (~5 μg) were separated on SDS-PAGE and then transferred to polyvinylidene difluoride (PVDF) membranes. Immunoblots were probed with specific polyclonal or monoclonal antibodies against TDE0362 which were generated in this study. The following antibodies were purchased, including anti-His-tag (Invitrogen, Waltham, MA), anti-FLAG-tag (Sigma-Aldrich), anti-CXCR1 (Santa Cruz Biotechnology, Dallas, TX), anti-CXCR2 (Santa Cruz Biotechnology), anti-FPR1 (Invitrogen), anti-C5aR (Abcam, Cambridge, United Kingdom), and anti-CD11b (Abcam). Immunoblots were developed using a horseradish peroxidase-conjugated secondary antibody with enhanced chemiluminescence (ECL) assays; signals were quantified using the Molecular Imager ChemiDoc system with the Image Lab software (Bio-Rad Laboratories, Hercules, CA), as previously described (Kurniyati et al, 2019).

## Preparation of TDE0362 recombinant proteins

Six TDE0362 recombinant proteins with different domains were prepared (see Reagent Table for details), including the full length protein (FL0362) without the signal peptide (30–647 aa), N0362 (23–204 aa) with both Big_1 and Big_2 domains, N1 (23–114 aa) with only the Big_1 domain, N2 (115–204 aa) with only the Big_2 domain, and C0362 (205–647 aa) with the Mac-like domain. The DNA fragments encoding these proteins were PCR amplified using *Pfx* DNA polymerase (Life Technologies, Grand Island, NY) with primer pair $P_5/P_6$ for N1, $P_7/P_8$ for N2, $P_9/P_{10}$ for N0362, $P_{11}/P_{12}$ for C0362, and $P_1/P_2$ for FL0362. The resultant PCR products of N1, N2, and F0362 were cloned into either pQE80L or pQE30 expression vectors (Qiagen, Valencia, CA), which encodes a 6xHis-tag at the N-terminus. The PCR product of N0362 was cloned into the pET100/D-TOPO expression vector (Life Technologies), which encodes a 6xHis-tag at the N-terminus. The resultant PCR product

of C0362 was cloned into the pET101/D-TOPO expression vector (Life Technologies), which encodes a 6xHis-tag at the C-terminus. The resulting plasmids were transformed into Lemo21(DE3) (New England Biolabs) for N1 and N2, BL21-CodonPlus (DE3)-RIL (Agilent) for N0362 and C0362, and M15 (Qiagen) for F0362. Protein expressions were conducted by inoculating 1 liter of LB with 25 mL of overnight starter *E. coli* cultures, which were incubated at 37 °C until an $OD_{600}$ of 0.4-0.8, whereupon induction was initiated by the addition of isopropyl β-D-thiogalactopyranoside (IPTG) to a final concentration of 1 mM and the temperature was lowered to 16 °C. Following 16–18 h of incubation, the cells were harvested by centrifugation and stored at −80 °C. The recombinant proteins were first purified using Ni-NTA agarose (Qiagen) under native conditions, dialyzed, and further purified by using size exclusion chromatography (SEC) with Superdex 200 Increase 10/300 GL columns (GE Healthcare Life Sciences, Marlborough, MA). The purified proteins were then dialyzed in a buffer containing 20 mM Tris-HCl buffer, pH 8.0 at 4 °C overnight using 3.0 kDa molecular weight cut-off Spectra/ Por® dialysis bags (Spectrum Laboratories). The concentrations of purified proteins were determined using a Bio-Rad Protein Assay Kit (Bio-Rad). Of note, for C0362 protein purifications, dithiothreitol (DTT) was added into the purification and dialysis buffers at a final concentration of 2 mM. The primers used here are listed in Table EV1.

### Production of TDE0362 polyclonal and monoclonal antibodies

The above purified F0362 protein was used to generate polyclonal antibodies in rats on a fee-for-service basis in General Bioscience (Brisbane, California) as previously described (Kurniyati et al, 2022a). The purified N0362 and C0362 proteins were used to produce monoclonal antibodies at the Center for Protein Therapeutics at University of Buffalo following a standard protocol of monoclonal antibody production, as previously described (Hansen and Balthasar, 2002). In brief, BALB/c mice were injected subcutaneously with 50 μg of recombinant proteins in 100 μl of saline buffer every 4 weeks. Four days after the second immunization, mouse splenocytes were isolated and fused with SP2/0-Ag14 (CRL-1581; ATCC) cells to obtain hybridomas. Binding of antibodies to N0362 and C0362 was initially assessed by screening hybridoma supernatants using ELISA followed by immunoblots. Hybridomas that produced antibodies that specifically reacted with the proteins were cloned by limiting dilutions and grown in serum-free medium (Thermo Fisher Scientific). Monoclonal antibodies were subsequently purified using a protein G affinity column (GE Healthcare Life Sciences).

### Co-immunoprecipitation (co-IP) for *T. denticola* cells

This study was carried out as previously described (Kurniyati et al, 2023). Briefly, 200 ml of the late-log phase *T. denticola* wild-type and *Δ0362* (negative control) cultures ($10^9$ cells/ml) were centrifuged at 3000×g for 20 min at 4 °C. The obtained cell pellets were washed twice with phosphate-buffered saline (PBS, pH 7.4) and then resuspended in 2 ml of TSEA buffer (50 mM Tris-HCl, 150 mM NaCl, 5 mM EDTA, 0.05% sodium azide, pH 7.5) containing 1% NP-40 and 50 μg PMSF. The resulting cell suspensions were sonicated on ice followed by

centrifugation (14,800×g for 30 min at 4 °C) to collect the cell supernatants. For co-IP, 900 μL of the cell supernatant was incubated with 100 μL of TDE0362 polyclonal antibody (polyAb) generated above for 16 h at 4 °C in the presence of 1% bovine serum albumin (BSA). After the incubation, 100 μl of protein G agarose (Sigma-Aldrich) was added, and the mixture was further incubated for 16 h at 4 °C. The resulting samples were centrifuged (1,600×g at 4 °C) and then washed three times with PBS buffer containing 0.05% Tween 20. The final pellets were suspended in Laemmli sample buffer, boiled for 5 min, and then briefly centrifuged prior to collecting the supernatants, which were then subjected to SDS-PAGE followed by liquid chromatography with tandem mass spectrometry (LC-MS/MS) analysis at the Mass Spectrometry Center University of Maryland School of Medicine and the Proteomics and Metabolomics Facility of Cornell University.

### Mapping TDE0362 cleavage sites

To determine the processing of TDE0362, FL0362 (30–647 aa) recombinant proteins were treated with the supernatants of *T. denticola* cell lysates. To prepare cell lysate supernatants, 10 ml of the mid-logarithmic-phase cultures (~$5 \times 10^8$ cells/ml) of the wild-type and dentilisin deficient mutant (*ΔCCE*) (Goetting-Minesky et al, 2021) strains were harvested at 4500×g for 15 min at 4 °C. The resulted cell pellets were washed once with phosphate-buffered saline (PBS, pH 7.4), resuspended in 1 ml of PBS buffer, lysed by sonication for 30 s on ice, and cleared by centrifugation at 15,000×g for 30 min at 4 °C. The resulted cell supernatants (5 μl, 10 μl or 20 μl) were co-incubated with 3.5 μM FL0362 in PBS buffer for overnight at 37 °C. The resulted samples were subjected to SDS-PAGE and cleaved products were excised for LC-MS/MS at the Proteomic Shared Resource (PSR), Virginia Commonwealth University.

### Determining the surface localization of TDE0362 using immunofluorescence assays (IFA)

This experiment was performed as previously described (Kurniyati et al, 2013). In brief, the mid-log phase *T. denticola* cells (~$5 \times 10^8$ cells/ml) were harvested by centrifugation at 3,000×g for 10 min. The harvested cells were washed once in PBS, pH 7.4, and then resuspended in PBS containing 400 μg of proteinase K (Sigma-Aldrich) and incubated at 22 °C for 1 h with gentle agitation every 5 min. The treatments were stopped by adding a final concentration of 2 mM phenylmethylsulphonyl fluoride (Sigma-Aldrich), followed by the incubation at 22 °C for 10 min and centrifugation at 3000×g for 10 min. The obtained cell pellets were washed once in PBS, resuspended in PBS, and subjected to IFA as previously described (Kurniyati et al, 2013). The resultant cells were placed on poly-L-lysine-coated cover slips allowing to fully air dry. The obtained coverslips were first incubated in a blocking solution (2% BSA in PBS, pH 7.4) for 1 h, followed by the incubation in the blocking solution containing 1:100 diluted polyAb for 1 h at room temperature. Finally, the cover slips were washed four times with PBS containing 0.05% Tween 20 (PBST), incubated with either goat anti-rat or goat anti-mouse IgG (H + L) highly cross-adsorbed secondary antibody, Alexa Fluor Plus 488 (Invitrogen) for 1 h at room temperature, washed with PBST, and mounted on ProLong™ Gold Antifade Mountant with DNA Stain DAPI (Invitrogen).

Images were collected using a Zeiss Axioimager Z1 Axiophot wide-field microscope and analyzed and quantitated using Zen Pro software (Zeiss, Germany).

## Yeast manipulation

*Saccharomyces cerevisiae* strain W303 (Fan et al, 1996) was grown at 30 °C in YPD medium or in synthetic media with appropriate amino acid dropout supplemented with either 2% glucose or galactose as the sole carbon source. Yeast transformation was performed using the standard lithium acetate protocol (Gietz and Schiestl, 2007). For yeast lethality assay, the full-length TDE0362 (from aa 1 to 647) and the C-terminal of TDE0362 (from aa 331 to 647) were cloned into p426GPD plasmid carrying a constitutively expressed GPD promoter with primer pair $P_{13}/P_{14}$ and $P_{53}/P_{14}$, respectively, or into pYES2/NTA plasmid that carries the galactose inducible Pgal1 promoter with primer pair $P_{54}/P_{55}$ and $P_{56}/P_{55}$, respectively, and the N-terminal of TDE0362 (from aa 22 to 204) was cloned into p426TEF plasmid, which carries a constitutively expressed TEF promoter with primer pair $P_3/P_4$ (Mumberg et al, 1995). The p426GDP, pYES2/NTA, and p426TEF were gifts from Z. Q. Luo from Purdue University. After 3 days incubation at 30 °C, the resultant transformed clones were confirmed by immunoblotting analysis. To prepare cell lysates for protein analysis, yeast cells from 5 ml overnight cultures were first lysed with a cracking buffer (40 mM Tris-Cl, pH6.8, 5% SDS, 0.1 mM EDTA, 8 M urea, bromothymol Blue 0.4 mg/ ml) with glass beads. Samples were resolved by SDS-PAGE after adding Laemmli buffer. For yeast lethality assay, overnight cultures of yeast transformants were serially diluted and spotted on appropriate amino acid dropout supplemented with either 2% glucose or galactose plates (Guo et al, 2014). The primers for constructing the plasmids are listed in Table EV1.

## Transient expression of TDE0362 in HEK293T cells

The TDE0362 gene sequences were codon optimized by GenScript Biotech (Piscataway, NJ). The DNA fragments encoding the full-length TDE0362 (1–647 aa), N0362 (1–204 aa), and C0362 (331–647 aa) were PCR amplified using SapphireAmp fast DNA polymerase (Takara Bio USA, San Jose, CA) with primer pair $P_{73}$/$P_{76}$, $P_{73}/P_{74}$, and $P_{75}/P_{76}$, respectively, and cloned into pcDNA6.2 C-EmGFP-GW TOPO expression vector (Invitrogen). The primers used here are listed in Table EV1. For transfection, a total of $10^4$ HEK293T cells were seeded into 24-well cell culture plates with Poly-L-lysine coated German glass coverslips (Electron Microscopy Sciences, Hatfield, PA) and cultured at 37 °C and 5% $CO_2$ for 24–48 h until the cells reached approximately 80% confluency. Transfection assay was performed using Lipofectamine 3000 Transfection Reagent (Thermo Fisher Scientific) according to the manufacturer's instructions for 72 h at 37 °C and 5% $CO_2$. After the incubation, the cells were collected and subjected to immuno-fluorescence imaging.

## Measuring the proteolytic activity of TDE0362

This study was performed by co-incubating different substrates, including C3 (675.68 nM), C3b (852.27 nM), C4 (365.85 nM), C4bp (138.89 nM), C5 (789.48 nM), Factor H (967.74 nM) (Complement

Technology, Tyler, TX), IgG (1.16 µM), and IgA (925.25 nM) (Sigma Aldrich), with either recombinant C0362 (1.90 µM) protein or its inactive form C412A point mutant (1.90 µM) in a reaction buffer (0.2 M Tris-HCl, pH 7.4, containing 0.1 M NaCl, 5 mM $CaCl_2$ and 2 mM DTT) for 16 h at 37°C anaerobically. After the incubation, Laemmli sample buffer was added to the samples, boiled for 5 min, and then subjected to SDS-PAGE analysis followed by Coomassie blue staining.

## Serum killing assays of *T. denticola*

This study was conducted as described previously with some modifications (Kochi et al, 1991; Kurniyati et al, 2013). Briefly, 75 µl of the mid-log phase *T. denticola* cultures (containing $7.5 \times 10^6$ spirochete cells) were first mixed with 25 µl of either normal human serum (NHS) or heat-inactivated serum (HIS, 56 °C for 30 min) and then incubated at 37 °C in the anaerobic chamber for 90 min. After incubation, the number of living spirochete cells was enumerated using a Petroff-Hausser counting chamber as described above. The living cells were determined by motility and cell surface integrity during the observation time (at least 1 min). The average survival rates (the number of living cells in NHS/the number of living cells in HIS) were calculated and the results are represented as the mean of survival rates ± standard error of the mean (SEM). Statistical analysis was determined using one-way ANOVA followed by Tukey's multiple comparison at $P < 0.05$.

## *E. coli* serum killing protection assays

For this study, the late-log phase ($\sim 1 \times 10^9$ cells/ml) of *T. denticola* cultures were diluted to $OD_{650}$ of 0.4 using HBSS buffer with $Ca^{2+}$ and $Mg^{2+}$ followed by centrifugations (12,000×g for 5 min at 4 °C). The resulted cell pellets were lysed by sonication and then centrifuged to collect *T. denticola* cell supernatants (*Td*-CS). Prior to serum killing assay, 6 µl of normal human serum (NHS) or heat-inactivated serum (HIS- 56 °C for 30 min) was treated with 56 µL of the *Td*-CS for 5 h at 37 °C anaerobically. After the treatment, 140 µL of overnight *E. coli* DH5α culture ($1 \times 10^3$ cells/ml) was added to the mixtures, further incubated for 15 min at 37 °C, and EDTA was added to a final concentration of 10 mM to stop the serum killing. The final samples were plated on to LB agar plates and incubated for 24 h at 37 °C. After the incubation, the *E. coli* colonies were enumerated to calculate the average survival rates (the number of colonies in NHS in relative to the number of colonies in HIS). The results are represented as the mean of survival rates ± standard error of the mean (SEM). Statistical analysis was determined using one-way ANOVA followed by Tukey's multiple comparison at $P < 0.05$.

## Preparation of C0362 proteins for crystal structure

BL21-CodonPlus (DE3)-RIL (pET101/D:: C0362) was used to prepare recombinant C0362 (204-647 aa) for crystal structure analysis. Large-scale C0362 expression was conducted by inoculating 1 L of LB with 25 mL of overnight starter culture. The culture was incubated at 37 °C until an $OD_{600}$ of 0.4–0.8, whereupon induction was initiated by the addition of IPTG to a final concentration of 1 mM and the temperature lowered to 18 °C. Following 16–18 h of incubation, the cells were harvested by

centrifugation and stored at −80 °C until use for purifications. To purify C0362 proteins, the cell pellet was resuspended in ice-cold Buffer A (50 mM Tris, 300 mM NaCl, 0.5 mg/mL lysozyme, 2 mM dithiothreitol (DTT), 20 mM imidazole, pH 8.0) and lysed by two passages through a microfluidizer at ~25,000PSI. The cell lysate was clarified by centrifugation at 40,000×g for 20 min at 4 °C. Clarified lysate was then incubated with 5 mL of pre-equilibrated Cobalt-NTA resin (Thermo Fisher, Rockford, IL) for 30 min at 4 °C with gentle mixing. The resin was poured onto a column support and washed with 10 column volumes (CV) of Buffer B (50 mM Tris, 300 mM NaCl, 2 mM DTT, 40 mM imidazole, pH 8.0). C0362 was eluted using 5 CV of Buffer C (Buffer B containing 300 mM imidazole). Elution fractions were then pooled and concentrated to 2 mL with 30 kDa cutoff Amicon Ultra centrifugal filter unit (Millipore-Sigma). Concentrated C0362 was applied to a HiLoad 16/600 Superdex 200 pg column (Cytiva) equilibrated in 20 mM Tris, pH 7.5, 150 mM NaCl, 2 mM DTT. Peak fractions containing C0362 were pooled and concentrated to ~50 mg/mL using a 30 kDa cutoff Amicon Ultra centrifugal unit, flash frozen in liquid nitrogen, and stored at −80 °C.

## Preparation of selenomethionine (SeMet) C0362 proteins

BL21-CodonPlus (DE3)-RIL (pET101/D:: C0362) was used to prepare SeMet C0362 for crystal structure analysis. The expression of SeMet C0362 was performed in a modified M9 minimal medium, which had the following composition (per liter): 6.0 g Na$_2$HPO$_4$, 3.0 g KH$_2$PO$_4$, 0.5 g NaCl, and 1.0 g NH$_4$Cl (M9 minimal medium). The medium was autoclaved. The following ingredients were mixed, sterilized by filtration, and added aseptically to the autoclaved medium containing (final concentration, per liter): 0.2 g MgSO$_4$, 10 mg CaCl$_2$, 9.0 g glucose, and 50 mg thiamine. To express SeMet C0362, 50 ml of overnight started culture was pelleted down by centrifugation, washed once in modified M9 minimal medium, and inoculated into 1 L of modified M9 minimal medium. The culture was incubated at 37 °C until an OD$_{600}$ of 0.6. The amino acid cocktail was mixed, sterilized by filtration, and added aseptically to the culture containing (final concentration, per liter): 100 mg lysine HCl, 100 mg threonine, 100 mg phenylalanine, 50 mg leucine, 50 mg isoleucine, and 50 mg valine. The culture was incubated at 37 °C for 15 min. Induction was initiated by the addition of IPTG to a final concentration of 1 mM and 50 mg of selenomethionine. The culture temperature was lowered to 25 °C. Following 48 h of incubation, cells were harvested by centrifugation and stored at −80 °C. The purification was performed using Ni-NTA agarose (Qiagen) as described previously for the wild-type C0362 purification.

## Crystallization and data collection

Initial crystallization leads were identified employing high-throughput crystallization screening coupled to a 1536 cocktail standard crystal screen (Luft et al, 2011). One lead was subsequently optimized in the micro-batch-under-oil format at 23 °C using a drop volume ratio approach (Luft et al, 2007). Rod-shaped crystals of SeMet substituted protein were obtained after 1–2 weeks by combining 2 μL of protein solution with 4 μL of a cocktail solution utilizing 20–28% polyethylene glycol (PEG) 20k, 100 mM MES, pH 6.0, 100 mM ammonium chloride. Protein crystals were grown utilizing the sitting-drop vapor

diffusion method by combining 2 μL of protein solution at 10 mg/mL with 4 μL of a cocktail solution containing 20% PEG 20k, 100 mM Tris, pH 8.0, 100 mM potassium bromide. For cryopreservation of SeMet protein crystals, 100 μL of the cocktail solution, supplemented with 15% (v/v) ethylene glycol, was added directly to the drop. Crystals were then harvested, followed by direct cooling in a gaseous nitrogen stream cooled to 100 K. Wild-type crystals were directly harvested and cooled in liquid nitrogen without additional cryoprotectant. Diffraction data from wild-type crystals were collected on beamline 23-ID-B at the Advanced Photon Source (APS) at Argonne National Laboratory utilizing a wavelength of 1.033 Å. Diffraction data from SeMet substituted crystals were collected on IMCA-CAT beamline 17-ID at the APS utilizing a wavelength of 0.9794 Å. Xia2/DIALS (Beilsten-Edmands et al, 2020; Winter, 2010; Winter et al, 2018) (v0.6.475/2.2.10) was used to integrate both datasets. Scaling and merging of the datasets was conducted with *Aimless* (Evans and Murshudov, 2013) and *Pointless* (Evans, 2011) in the CCP4 suite (Winn et al, 2011) (v7.1.016). The SeMet dataset contained 4 monomers in the asymmetric unit in space group P2$_1$, while the wild-type dataset contained a single monomer in the asymmetric unit in space group P2$_1$2$_1$2. Data collection statistics are reported in Appendix Table S1.

## Structure solution and refinement

Initial efforts to solve the structure using molecular replacement (MR) methods and the crystal structure of gasMac-1 (Agniswamy et al, 2006) as a search model failed to produce a solution. Single wavelength anomalous diffraction (SAD) methods were then utilized in conjunction with *HySS* (Grosse-Kunstleve and Adams, 2003; McCoy et al, 2004) and *Phaser* (McCoy et al, 2007) in the *Phenix* (Liebschner et al, 2019) suite of programs to generate initial phases. *HySS* resolved 23 SeMet sites with a CC of 0.42. Next, *Phaser* and *RESOLVE* (Terwilliger, 2000) were employed to assist in resolution of the hand ambiguity. Iterative cycles of *Autobuild* (Terwilliger, 2004; Terwilliger, 2002; Terwilliger et al, 2008; Zwart et al, 2005), which uses *Phenix.refine* (Afonine et al, 2012) for refinement during the building process, were then run. At this stage, ~70% of the residues were built for the four monomers in the asymmetric unit, with the model having an R-work and R-free values of 0.30 and 0.33, respectively, and a CC of 0.70. To confirm the correct hand selection and the automated building with *AutoBuild*, the anomalous map was inspected and confirmed anomalous signal at the regions where SeMet was built. Iterative cycles of manual rebuilding and refinement of the model in *Coot* (Emsley et al, 2010) and *Phenix.refine* were then conducted to fit the remaining residues. Initial stages of refinement employed the use of torsion-angle Non-Crystallographic Symmetry (NCS) restraints (Terwilliger, 2013), which were released at the final stages of refinement. *Phaser* was then utilized to determine initial phases for the wild-type structure using one monomer of the SeMet substituted structure as a search model. Model building and refinement for the wild-type structure was carried out utilizing *Coot* and *Phenix.refine*. In the final rounds of refinement for both models, ordered solvent molecules and ligands were added and Translation-Libration-Screw (TLS) refinement (Winn et al, 2001) was employed. The final model of C0362 is comprised of residues 211–647, with no corresponding electron density observed for residues Thr-205 through Ile-210 at the N-terminus, Asn-350 through Ile-352, and Ser-553 through Pro-558.

## Structural modeling

Model validation was carried out using *MOLPROBITY* (Davis et al, 2007; Williams et al, 2018). Structural alignments were performed using *GESAMT* (Krissinel, 2012) (v1.16) in the CCP4 suite of programs. The coordinates and structure factors for the wild-type structure have been deposited in the Protein Data Bank as entry 9ARC. The structure of full-length TDE0362 (from aa 23 to 647) was generated using the AlphaFold2 Colab (Mirdita et al, 2022; Varadi et al, 2021).

## Small angle X-ray scattering

Full-length TDE0362 was expressed and purified as described for C0362, except utilizing a construct encoding residues 41–647 of TDE0362, with an N-terminal 6xHis-tag. Additionally, DTT was excluded from purification and SEC buffers, and 5% glycerol was included in all purification buffers, with 1% glycerol supplemented into the size exclusion chromatography (SEC) buffer. Following SEC purification, protein was concentrated to 10 mg/mL, flash frozen in liquid nitrogen, and stored at −80 °C until use. Protein was submitted at 10 mg/mL for mail-in SEC-Small angle X-ray Scattering (SEC-SAXS) (Yang et al, 2021) on the LIX (DiFabio et al, 2016; Yang et al, 2020) (16-ID) beamline at the National Synchrotron Light Source II at Brookhaven National Laboratory. Buffer used for SEC-SAXS experiments was 20 mM Tris, pH 7.5, 150 mM NaCl, 1% glycerol. SEC-SAXS series analysis was conducted with the LC analysis built into BioXTAS *RAW* (Hopkins et al, 2017). Buffer subtraction was conducted by using regions before and after the elution peaks from the chromatogram. The experimental SEC-SAXS profile showed two peaks, for which we averaged frames 116–150 to generate a single SAXS profile for the first peak, and for the second peak frames 163–182 were averaged. To assess fit of the AlphaFold2 model to the experimental SAXS data, we used *CRYSOL* (Svergun et al, 1995). We modified the standard *CRYSOL* settings to allow constant subtraction, harmonic order set to 50, and scattering angle range of 0.021–1.0 Å$^{-1}$. We used *GNOM* (Svergun, 1992) to perform an indirect Fourier transform (IFT) (Appendix Fig. S7) and determine the radius of gyration ($R_g$), molecular weight, and real space pair-distance distribution function (P(r)). The resultant IFT was then used with denss.all.py to conduct 20 individual *DENSS* (Global Burden of Cardiovascular Diseases et al, 2018) (v1.7.1) simulations with a Shrinkwrap threshold of 0.15, 128 sampling points per dimension, and a $D_{max}$ of 150.0 Å for 15,000 enforced steps for each simulation. The final *DENSS* electron density map was generated by aligning and averaging the individual *DENSS* reconstructions. We conducted refinement of the averaged electron density map to the data using *denss.refine.py* and fit the AlphaFold2 model to the refined map using *dens.align.py* and fine-tuned the fit in *ChimeraX* (Pettersen et al, 2021) (v1.6.1). Bead modeling was conducted using *DAMMIF* (Franke and Svergun, 2009), in SLOW mode, and *GNOM* (Svergun, 1992), with a $D_{max}$ of 150Å to fit the data and generate the necessary input file, in the *RAW* suite of programs. The reconstruction resolution was determined using *SASRES* (Tuukkanen et al, 2016). The 20 models generated by *DAMMIF* were averaged with *DAMAVER* (Volkov and Svergun, 2003) and refined with *DAMMIN* (Svergun, 1999). The final bead model was aligned to the AlphaFold2 full-length TDE0362 model using *CIFSUP* (Franke and Svergun, 2009) in *RAW* and fine-tuning of the fit in *ChimeraX* (Pettersen et al, 2021) (v1.6.1).

## Human and mouse neutrophil isolations

Human neutrophil cells were isolated from peripheral human blood samples using 1-Step Polymorphs solution (Accurate Chemical & Scientific, Carle Place, NY) as previously described (Anselmi et al, 2023). The final cell pellets were resuspended in Hanks-Balanced Salt Solution (HBSS) without $Ca^{2+}$ and $Mg^{2+}$. The isolated human neutrophils were confirmed by Alexa Fluor 647 mouse anti-human CD66b (BD Biosciences, San Jose, CA) using flow cytometry. Mouse neutrophil isolation was conducted as previously described (Jones et al, 2017). Briefly, C57BL/6J wild-type mice (male, 6 weeks old) were purchased from Jackson Laboratory (Bar Harbor, Maine). Femurs and tibias were removed and cells were isolated from bone marrows by fractionation into discontinuous Percoll (Sigma) gradients (80%, 65%, 55%). Mature neutrophils were isolated from the 80%/65% interface.

## Immunofluorescence staining

Immunofluorescence staining was performed as previously described (Kurniyati et al, 2013). A total of $1 \times 10^6$ cells human neutrophils were incubated with 20 μg Alexa Fluor 488-labeled N0362 proteins in a final volume of 500 μl on ice for 30 min. After the incubation, the cells were harvested, fixed with 2% paraformaldehyde in Hanks-Balanced Salt Solution (HBSS) for 30 min at room temperature, washed with cold HBSS twice, and resuspended in HBSS. The cells suspension was stained with propidium iodide to a final concentration of 30 μM and immobilized on Poly-L-lysine coated German glass coverslips (Electron Microscopy Sciences, Hatfield, PA). The coverslip was then mounted using ProLong Gold Antifade Mountant (Invitrogen) and visualized using a Zeiss microscope. For HL-60, DMSO-differentiated HL-60 (dHL-60), a total of $2.5 \times 10^4$ cells were seeded into 12-well cell culture plates and cultured at 37 °C and 5% $CO_2$ for 24 h. After the incubation, the cells were treated with 20 μg Alexa Fluor 488-labeled N0362 proteins for 2 h at room temperature and fixed with 3.7% paraformaldehyde in PBS (pH 7.4) for 30 min at room temperature. After fixation, the cells were permeabilized with 0.1% Triton X-100 for 10 min and washed twice with cold PBS. The coverslips were first incubated in a blocking solution (5% BSA in PBS, pH 7.4) for 1 h at room temperature, followed by incubation with anti-FPR1 or anti-CXCR1 antibody at 4 °C for 24 h. After the incubation, the coverslips were washed three times with cold PBS, incubated with either goat anti-mouse or goal anti-rabbit IgG (H + L) cross-adsorbed secondary antibody, Alexa Fluor 594 cross-adsorbed secondary antibody, Alexa Fluor 594 (Invitrogen) for 1 h at room temperature, and washed three times with cold PBS. The coverslips were then mounted using ProLong Gold Antifade Mountant with DNA Stain DAPI (Invitrogen). Images were collected using Zeiss Axioimager Z1 Axiophot wide-field microscope and analyzed and quantitated using Zen Pro software (Zeiss)

## siRNA transfection

dHL-60 cells ($1 \times 10^6$) were seeded into 24-well cell culture plates and transfected with 0.5 μg of either CXCR1 siRNA (sc-40026), FPR1 siRNA (sc-40121) or scramble siRNA (sc-37007, Santa Cruz Biotechnology) using Lipofectamine LTX with PLUS reagent (Invitrogen) for 96 h at 37 °C and 5% $CO_2$. After the incubation,

the cells were fixed with 3.7% paraformaldehyde in PBS for 30 min at room temperature and were then subjected to immunoblotting and immunofluorescence imaging analysis.

## Neutrophil chemotaxis assays

Analysis of neutrophil chemotaxis using transwell assay was performed as previously described (Jones et al, 2017). Briefly, media with or without N-formyl-methionyl-leucyl-phenylalanine (fMLP; 1 μM) was placed in the bottom of the plates. Human or mouse neutrophils ($0.5 \times 10^6$) were incubated with or without recombinant N0362 proteins for 30 min at room temperature. The neutrophil suspensions were added to the top of the transwell and incubated at 37 °C for 1 h. The top of the membrane was gently wiped clean and the entire membrane was fixed in 4% paraformaldehyde overnight at 4 °C. Membranes were then washed and stained with crystal violet. After subsequent washing with dH$_2$O until all excess dye had been removed, cells fixed to the membrane representing the migrated cell population were counted using an inverted microscope. The cells in 5 different areas of each membrane were counted, with duplicate transwells per condition, to yield the total migrated cells. All data were normalized to the control with fMLP alone (positive control). The data were analyzed by one-way ANOVA followed by Tukey's multiple comparison at $P < 0.05$.

## Neutrophil killing assays

This experiment was performed as described previously (Yaseen et al, 2017). Briefly, dHL-60 cells were plated at $1 \times 10^6$ cells total in 24-well non-treated cell culture plates and pre-stimulated with phorbol 12-myristate 13-acetate (PMA; 50 nM) at 37 °C and 5% CO$_2$ for 30 min. Unstimulated dHL-60 cells and samples without dHL-60 were used as a control. Meanwhile, T. denticola cells (wild-type, $\Delta 0362$, and $C\Delta 0362$) were harvested at the mid-log phase ($\sim 5 \times 10^8$ cells/ml) and washed by centrifugation at 3000×g for 10 min. The harvested cells were washed once in HBSS. Then, dHL60 cells were co-incubated with T. denticola cells at a multiplicity of infection (MOI) of 1:2 in a final volume of 250 μl at 37 °C in 5% CO$_2$ for 1 h. Following the incubation, the plates were brought into the anaerobic chamber and 250 μl water was added to the reaction to lyse the dHL-60 cells for 30 min at 37 °C. Then, 1 ml of TYGVS growth medium was added to the reaction, stained with propidium iodide to a final concentration of 30 μM, and incubated for 15 min at room temperature. After the incubation, the number of living bacterial cells was enumerated using a Petroff-Hausser counting chamber (Hausser Scientific). For the protection assay using the $\Delta 0362$ mutant, prior to activation with PMA, dHL-60 cells were treated with 10 μM of N0362 proteins at 37 °C and 5% CO$_2$ for 30 min and then co-incubated with $\Delta 0362$ cells at MOI of 1:2. The average survival rates were calculated as: the number of living cells with dHL-60 cells/the number of living cells without dHL-60 cells. The results are represented as the mean of survival rates ± standard error of the mean (SEM). Statistical analysis was determined using one-way ANOVA followed by Tukey's multiple comparison at $P < 0.05$.

## Flow cytometry

For this assay, N0362 proteins were labeled using Alexa Fluor 488 Protein Labeling Kit (Invitrogen) following the manufacturer's protocol. For the binding assay, a total of $1 \times 10^6$ human neutrophils or HL-60 cells were incubated with different concentration of labeled proteins in a final volume of 500 μl on ice for 30 min. After the incubation, the cells were harvested, washed in HBSS buffer by centrifugation at 200×g for 10 min and resuspended in 500 μl of HBSS buffer. Fluorescence intensities were measured using BD FACSCanto II flow cytometry system (BD Biosciences). The gating cells strategy was based on FSC-A/SSC-A profile, followed by selecting single cells population (FSC-A/FCH-H), and subsequently detecting fluorescently labeled cells (Alexa Fluor 488-A/PE-A). Statistical analysis was determined using one-way ANOVA followed by Tukey's multiple comparison at $P < 0.05$.

## Co-IP using HL-60 cells

This study was carried out as previously described (Kurniyati et al, 2023). In brief, dHL-60 cell cultures were harvested by centrifugation at 200×g for 10 min at 4 °C, washed with PBS, pH 7.4, resuspended in Pierce IP Lysis Buffer (Thermo Fisher Scientific), and incubated for 15 min on ice. After the incubation, the cell lysates were centrifuged at 12,000×g for 10 min at 4 °C. For co-IP, the cleared cell lysate was incubated with 10 μg His-tagged N0362 protein for 3 h at 4 °C. Then, 5 μl Ni-NTA resin slurry was added to the mixture and further incubated at 4 °C for overnight. Following the incubation, the mixture was washed five times with PBS buffer containing 0.05% Tween 20. The final pellets were suspended in Laemmli sample buffer, boiled for 5 min, and then briefly centrifuged to collect the supernatants, which were then subjected to SDS-PAGE followed by immunoblotting analysis.

## Enzyme-linked immunosorbent assay (ELISA)

This experiment was carried out to determine if N0362 binds to FPR1 and CXCR1 receptors as previously described (Choy et al, 2011). Purified N0362 and GST proteins (negative control) were each immobilized in high binding microtiter wells (Greiner Bio-One, Monroe, NC) by overnight incubation of 1 μg of proteins in 100 μl PBS, pH 7.4, at 4 °C. After washing with PBS, non-specific binding sites were blocked with 2% BSA in PBS for 1 h at room temperature, followed by incubation with recombinant GST-FPR1 and GST-CXCR1 proteins (Abnova, Taiwan) for 2 h at room temperature. After washing with PBS, bound FPR1 and CXCR1 were detected with a polyclonal GST antibody, followed by incubation with a horseradish peroxidase-conjugated sheep anti-goat IgG (H + L) (Invitrogen). The binding was measured by the absorbance at 450 nm of the enzymatic conversion of 3,3',5,5'-tetramethylbenzidine (TMB; Thermo Fisher Scientific). The data are presented as the mean of binding affinity relative to GST ± standard error of the mean (SEM).

## Expression and purification of CXCR1 proteins

CXCR1 (Uniport entry P25024) recombinant protein was prepared for surface plasmon resonance (SPR) analysis. The gene sequences were codon optimized and synthesized by GenScript Biotech. The DNA fragment encoding the full length of CXCR1 protein was PCR amplified with primer pair P$_{79}$/P$_{80}$ using PrimeSTAR GXL DNA polymerase (Takara Bio USA) and cloned into pGEX-6P-1 expression vector with a N-terminus GST tag (Millipore-Sigma). The resulted

expression vector was transformed into *E. coli* BL21 strain. The expression of CXCR1 was conducted by inoculating 1 L of LB with 25 mL of overnight BL21 starter culture. The culture was incubated at 37 °C until an $OD_{600}$ of 0.4–0.8, whereupon induction was initiated by addition of 1 mM IPTG and the temperature was lowered to 16 °C. Following 16–18 h of incubation, *E. coli* cells were harvested by centrifugation and stored at −80 °C until use for purification. The recombinant protein was purified using Glutathione Agarose (Thermo Fisher) under native conditions according to the manufacturer's protocol. To solubilize the protein, 5% sarkosyl was added to the lysis buffer. The purified protein was then subjected to SEC using Superdex 200 Increase 10/300 GL column (GE Healthcare Life Sciences, Marlborough, MA). The concentration of purified proteins was determined using a Bio-Rad Protein Assay Kit (Bio-Rad). The primers used here are listed in Table EV1.

## Surface plasmon resonance (SPR) analysis

This study was conducted using Reichert SR7500DC instruments (Reichert, Depew, NY) using CM5 sensor chips. Experiments were performed at 25 °C unless otherwise specified. The running buffer containing dPBS and 0.01% Tween 20 was 0.22-μm sterile filtered and degassed for at least 3 h. All other reagents and protein solutions were degassed for 20 min. CM5 sensors were first pre-conditioned with the running buffer at a flow rate of 10 μL/min for overnight, followed by activation using freshly prepared solution containing 40 mg EDC and 10 mg NHS at a flow rate of 10 μL/min for 7 min. Following the activation, 60 μg/ml of CXCR1in Borate buffer pH 8.5 was applied at a flow rate of 8 μl/min for 10 min. This immobilization step was repeated for a total of four times, followed by blocking using Ethylenediamine (1 M) and Ethanolamine (1 M) at a flow rate of 10 μl/min for 8 min. For the binding experiment, N0362 proteins (ranging from 0.1 μM to 3 μM) diluted in the running buffer were applied to the sensor surface at a flow rate of 25 μl/min for 3 min. Following the association, running buffer alone was applied to the sensor at a flow rate of 25 μL/min for 10 min to allow N0362 proteins to dissociate from immobilized CXCR1. At the end of each injection, regeneration buffer (10 mM glycine, pH 1.7) was applied at a flow rate of 25 μL/min for 1:30 min to remove any undissociated N0362 prior to the next injection. The data analysis was analyzed using TraceDrawer software (version 1.8.1; Ridgeview Instruments, Sweden). For kinetic analysis, the data was globally fit to one to two binding model; for EC50 analysis, the data was fit using affinity model.

## Chemokine receptor assay

This experiment was performed using ready-to-assay CXCR1 chemokine receptor cells (Millipore-Sigma). The frozen cells from the manufacturer were recovered in the media at 37 °C and 5% $CO_2$ for 24 h. And then, 100 μl of the cell suspension was seeded on 96-well black plate and incubated with Fura-2AM solution (Abcam) at 37 °C and 5% $CO_2$ for 1 h. Then, N0362 proteins (5 μM) were added and further incubated at 37 °C and 5% $CO_2$ for 10 min. After the incubation, IL-8 (10 nM, Novus Biologicals, Centennial, CO) was added and the fluorescence intensity was monitored at Ex/Em = 340/510 nm and Ex/Em = 380/510 nm. The data was recorded as the mean of relative fold change based on the fluorescence ratio of 340/380 nm ± standard error of the mean

(SEM). The sample treated with only IL-8 was set as the baseline (onefold). Statistical analysis was determined using unpaired *t* test at $P < 0.05$.

## Assessing the pathogenic role of TDE0362 using a murine skin abscess model

A previously documented mouse skin abscess model (Bian et al, 2013) was used to assess the virulence of *T. denticola*. For this study, 6- to 8-week-old BALB/c mice (Jackson Laboratory, Bar Harbor, ME) were used. Each mouse (5 mice per bacterial strain) received a single subcutaneous injection of 200 μl of bacterial suspension (~$10^9$ cells) on its posterior dorsolateral surface. After the injection, infected mice were monitored for symptoms of infection on a daily basis for a total of 10 days. The diameters of the observed abscesses were measured with a caliper gauge. Each abscess was measured twice from different angles, and average sizes were calculated (area = $\pi r^2$, where r is the radius). The data were analyzed by one-way ANOVA followed by Tukey's multiple comparison at $P < 0.01$. Skin tissues around the abscesses were excised, fixed in 4% paraformaldehyde solution, and then subjected to Hematoxylin and Eosin (H&E) staining and Warthin Starry staining. H&E staining was performed at VCU Massey Cancer Center. Warthin Starry staining was performed using the Warthin-Starry stain kit (Abcam), following the manufacturer's instructions.

## Data availability

The structural coordinates data from this publication have been deposited to the wwPDB database [https://deposit.wwpdb.org/deposition/] and assigned the identifier [9ARC].

The source data of this paper are collected in the following database record: biostudies:S-SCDT-10_1038-S44318-024-00342-8.

## Peer review information

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

## Acknowledgements

This project is supported by DE023080 to C Li and MG Malkowski; DE030667 to C Li; USPHS award GM095459 to ND Clark; and DE027073 to MB Visser. Services in support of the research project were provided by the VCU Massey Cancer Center Flow Cytometry Shared Resource supported, in part, with funding from NIH-NCI Cancer Center Support Grant P30 CA016059. The LiX beamline is part of the Center for BioMolecular Structure (CBMS), which is primarily supported by the National Institutes of Health, National Institute of General Medical Sciences (NIGMS) through a P30 Grant (P30GM133893), and by the DOE Office of Biological and Environmental Research (KP1605010). LiX also receives additional support from NIH Grant S10 OD012331. As part of NSLS-II, a national user facility at Brookhaven National Laboratory, work performed at the CBMS is supported in part by the U.S. Department of Energy, Office of Science, Office of Basic Energy Sciences Program under contract number DE-SC0012704. This research used resources of the Advanced Photon Source U.S. Department of Energy (DOE) Office of Science User Facility operated for the DOE Office of Science by Argonne National Laboratory under Contract No. DE-AC02-06CH11357. Use of the IMCA-CAT beamline 17-ID (or 17-BM) at the Advanced Photon Source was supported by the companies of the Industrial Macromolecular Crystallography Association through a contract with Hauptman-Woodward Medical Research Institute. GM/CA@APS has been funded by the National Cancer Institute (ACB-12002) and the National Institute of General Medical Sciences (AGM-12006, P30GM138396). The Eiger 16 M detector at GM/CA-XSD was funded by NIH grant S10 OD012289. We thank the Proteomics and Metabolomics Facility of Cornell University for providing the mass spectrometry data and NIH SIG grant 1S10 OD017992-01 support for the Orbitrap Fusion mass spectrometer. We thank Joseph P. Balthasar for producing TDE0362 monoclonal antibodies, Dr. Zhao-qing Lu for providing plasmids for yeast toxicity assays, Dr. Megan Jones for her assistance with transwell assays, Dr. Xinyan Pei for her assistance with SPR, Drs. Thomas Grant and Sarah Chamberlain for their assistance with SEC-SAXS, and Dr. Mary Koszelak-Rosenblum for performing initial crystallization screening experiments.

## Author contributions

**Kurni Kurniyati**: Data curation; Formal analysis; Validation; Investigation; Visualization; Methodology; Writing—original draft; Writing—review and editing. **Nicholas D Clark**: Data curation; Formal analysis; Funding acquisition; Validation; Investigation; Visualization; Methodology; Writing—original draft; Writing—review and editing. **Hongxia Wang**: Data curation; Validation; Investigation; Visualization; Methodology. **Yijie Deng**: Data curation; Validation; Investigation; Visualization; Methodology; Writing—original draft. **Ching Wooen Sze**: Data curation; Validation; Methodology; Writing—original draft; Writing—review and editing. **Michelle B Visser**: Data curation; Formal analysis; Funding acquisition; Investigation; Methodology; Writing—original draft; Writing—review and editing. **Michael G Malkowski**: Conceptualization; Data curation; Formal analysis; Supervision; Funding acquisition; Validation; Investigation; Visualization; Methodology; Writing—original draft; Project administration; Writing—review and editing. **Chunhao Li**: Conceptualization; Resources; Data curation; Formal analysis; Supervision; Funding acquisition; Validation; Investigation; Visualization; Methodology; Writing—original draft; Project administration; Writing—review and editing.

Source data underlying figure panels in this paper may have individual authorship assigned. Where available, figure panel/source data authorship is listed in the following database record: biostudies:S-SCDT-10_1038-S44318-024-00342-8.

## Disclosure and competing interests statement

The authors declare no competing interests.

# Expanded View Figures

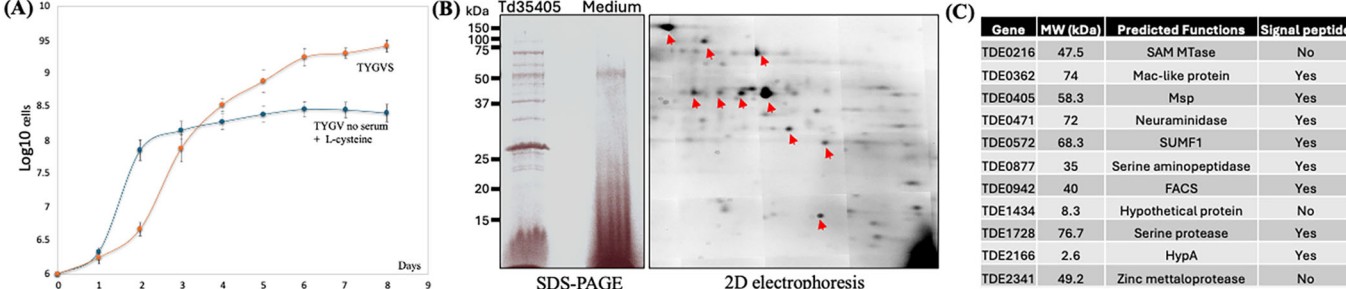

**Figure EV1.  Analysis of *T. denticola* ATCC35405 cell spent culture supernatants (SCSs).**

(**A**) Growth curves of *T. denticola* ATCC 35405 in the TYGVS medium with or without rabbit serum supplemented with L-cysteine (1 g/liter). Cell counting was repeated in triplicate. (**B**) SDS-PAGE (left panel) and 2D-gel electrophoresis (right panel) of *T. denticola* SCSs prepared from the serum-free growth medium. Samples with the medium alone served as a negative loading control. The red arrows indicated 11 demarcated spots that were excised and subjected to matrix-assisted laser desorption/ionization-time of flight mass spectrometry (MALDI-TOF MS) analysis. (**C**) A list of 11 proteins were identified in *T. denticola* SCSs by MALDI-TOF MS with high confidence. Signal peptides (SPs) were predicted using Signal IP 5.0.

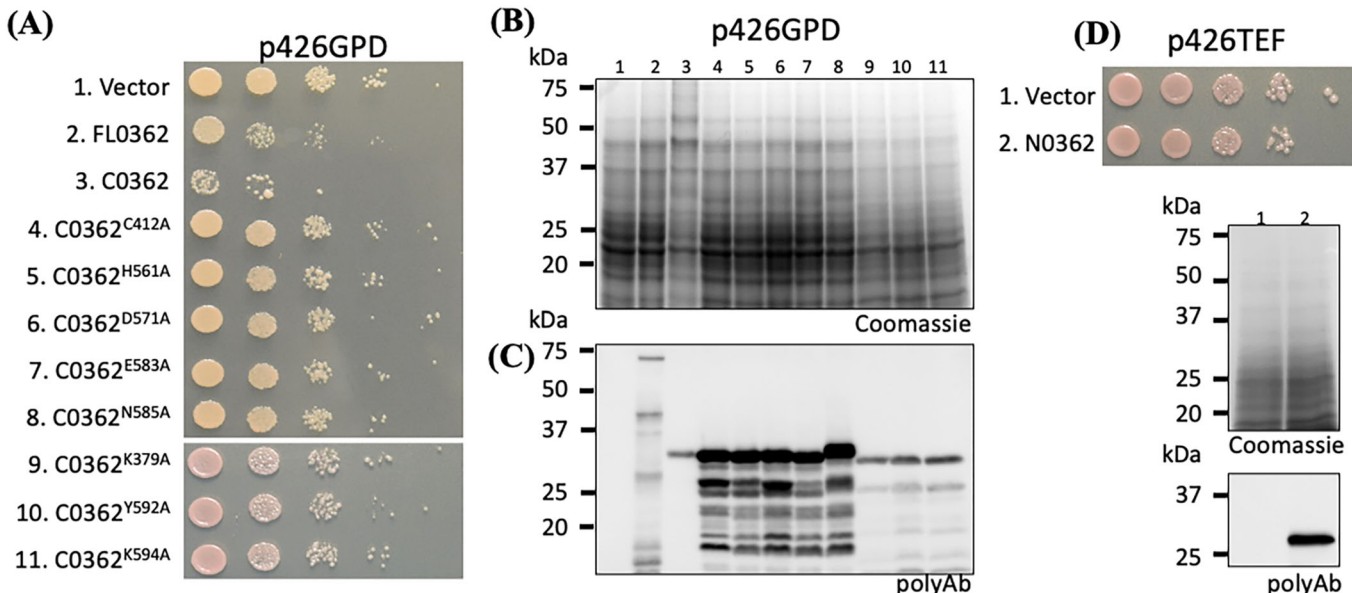

**Figure EV2. Yeast cell toxicity assays.**

(A) Yeast strains expressing the full-length TDE0362 (FL0362, 1–647 aa), C-terminal Mac-1 domain in TDE0362 (C0362, 331–647 aa), or eight point mutants as labeled, were expressed constitutively using GPD promoter (p426GPD). The strains were serial-diluted and spotted onto plates containing glucose. Plates were incubated at 30 °C for 48 h before image acquisition. (B, C) Monitoring the expression of TDE0362 proteins using SDS-PAGE followed by immunoblots using a specific polyclonal antibody against TDE0362 (polyAb). (D) N-terminal TDE0362 (N0362, 23–204 aa) was constitutively expressed in yeast cells using TEF promoter (p426TEF). The strains were serial-diluted and spotted onto plates containing glucose. Plates were incubated at 30 °C for 48 h before image acquisition. The expression of N0362 was monitored by immunoblots probed against polyAb.

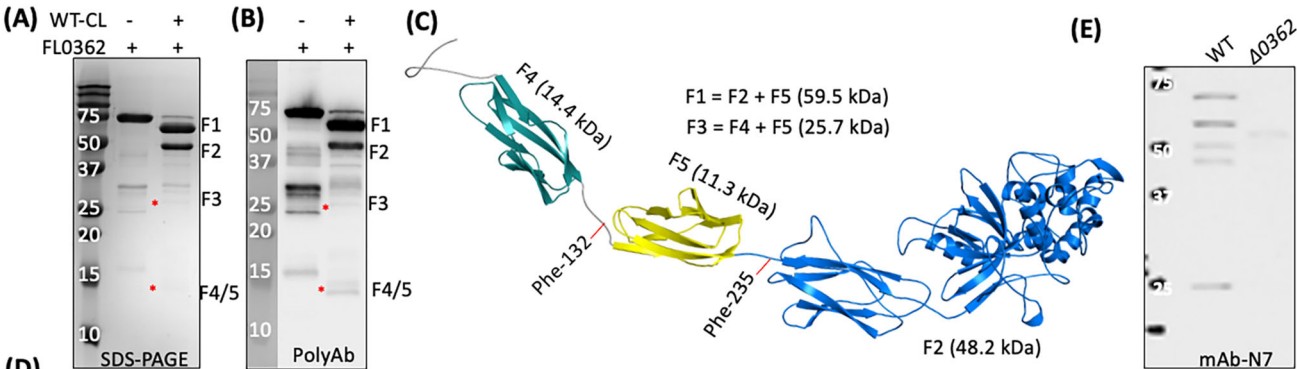

**Figure EV3. Mapping TDE0362 cleavage sites.**

For this study, the full-length TDE0362 (FL362) recombinant proteins were incubated with or without *T. denticola* WT cell lysate supernatants anaerobically for overnight; the resulted samples were subjected to SDS-PAGE (**A**) or immunoblotting probed against polyAb (**B**). The bands of F1 and F2 were excised and subjected to LC-MS/MS. Red asterisks are two minor cleaved bands. (**C**) A diagram illustrating the domain composition of TDE0362, cleavage sites, and the sizes of five cleaved products which were calculated based on their corresponding sequences. (**D**) The amino acid sequence of TDE0362. The detected cleavage products were labeled in different colors. * stands for the two Phe cleavage sites mapped. (**E**) Detection of TDE0362 in the SCSs of WT and *Δ0362* strains by immunoblots probed against mAb-N7, a monoclonal antibody against N0362. For this experiment, co-IP was first carried out to pull down TDE0362 cleaved products from the SCSs using polyAb and then probed against mAb-N7.

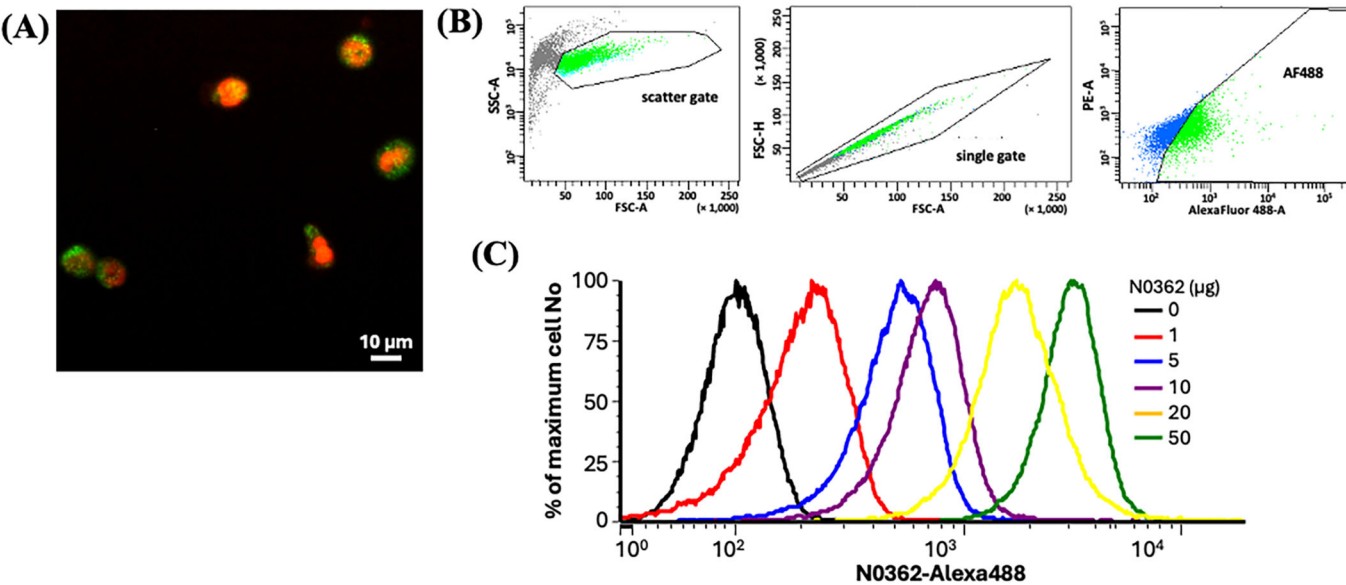

**Figure EV4. N0362 binds to human neutrophils.**

(A) Immunofluorescence staining using human neutrophil and Alexa Fluor 488 labelled N0362. The neutrophil was stained using propidium iodide. The scale bars represent 10 µm. (B, C) Flow cytometry was performed using human neutrophils and different concentrations of Alexa Fluor 488-labeled N0362, ranging from 0 to 50 µg of total proteins. (B) The gating cells strategy was based on FSC-A/SSC-A profile, followed by selecting single cells population (FSC-A/FCH-H), and subsequently detecting fluorescently labelled cells (Alexa Fluor 488-A/PE-A). (C) Histogram of flow cytometric analysis showed dose-dependent binding between neutrophil and N0362.

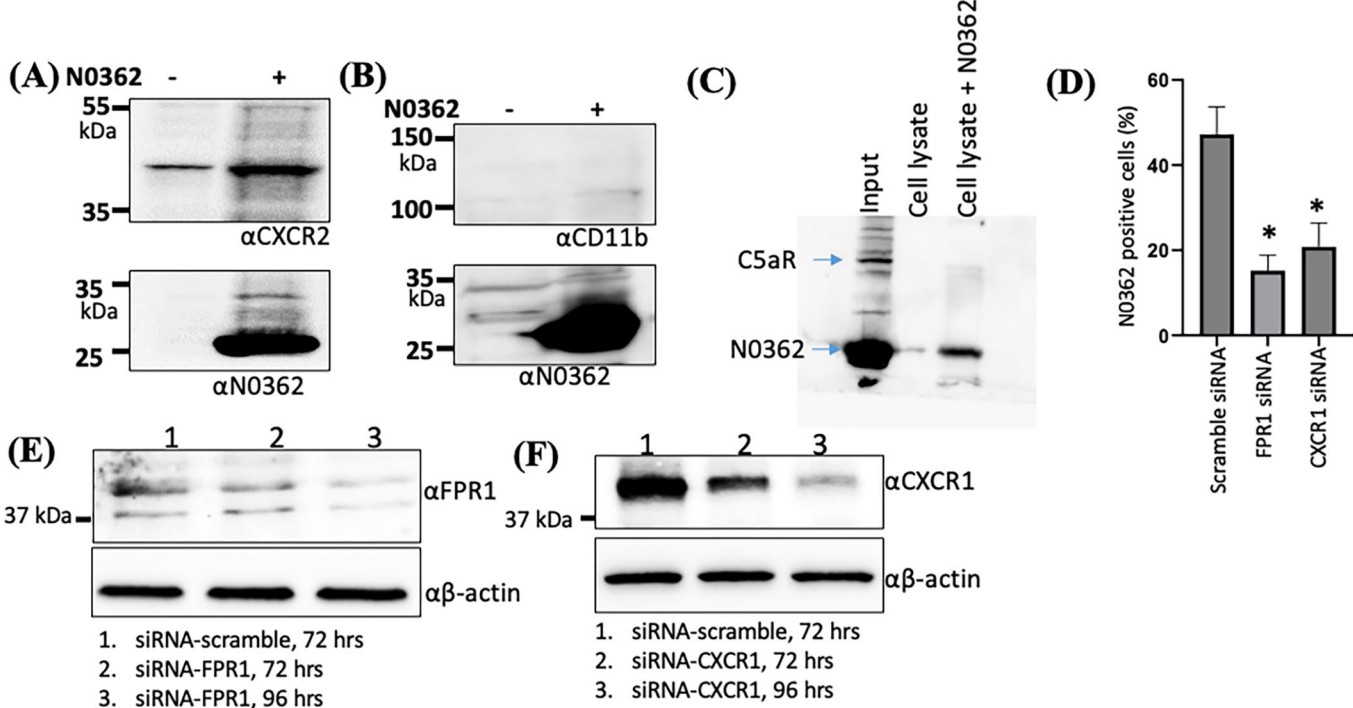

**Figure EV5.   Co-IP assays.**

(A–C) For this experiment, DMSO-differentiated HL-60 cell (dHL-60) lysates were incubated with or without His-tagged N0362 protein (10 µg) for 3 h at 4 °C, followed by precipitation using Ni-NTA resin. The resulted co-IP samples were subjected to immunoblotting analysis probed against mAb-N7, a monoclonal antibody that recognizes N0362. (A) anti-CXCR2 (αCXCR2), (B) anti-CD11b (αCD11b), and (C) anti-C5aR (αC5aR). (D) siRNA knockdown of FPR1 or CXCR1 reduces the binding of N0362 to HL-60 cells. For this experiment, activated HL-60 cells were transfected with scramble siRNA (control), FPR1 siRNA or CXCR1 siRNA for 3 days and then the cells were co-incubated with His-tagged N0362 for 2 h. Cells were then fixed and stained anti-His antibody. Fluorescent cells were counted and recorded as percentage of positive cells in relative to total cells. *$P$ value < 0.05. (E, F) Detection of FPR1 and CXCR1 by immunoblotting analysis. Lane 1: scramble siRNA; Lane siRNA-FPR1 or CXCR1 72 h after the knockdown; and Lane 3: siRNA-FPR1 or CXCR1 96 h. β-actin was used as a loading control.

