## [Peer Review File · The EMBO Journal]

A bipartite bacterial virulence factor targets the complement system and neutrophil activation

Kurni Kurniyati, Nicholas Clark, Hongxia Wang, Yijie Deng, Ching Woon Sze, Michelle Visser, Michael Malkowski, and Chunhao (Chris) Li

Corresponding author(s): Chunhao (Chris) Li (cli5@vcu.edu) , Michael Malkowski (mgm22@buffalo.edu)

Review Timeline:

Submission Date:	30th Mar 24
Editorial Decision:	14th May 24
Revision Received:	2nd Aug 24
Editorial Decision:	20th Sep 24
Revision Received:	26th Oct 24
Editorial Decision:	20th Nov 24
Revision Received:	22nd Nov 24
Accepted:	25th Nov 24

Editor: Ioannis Papaioannou

Transaction Report:

Dear Prof. Li,

Thank you for submitting your manuscript EMBOJ-2024-117447 for consideration by The EMBO Journal, and for your patience during peer review. It has now been seen by two experts in the field, and we have received the full set of their comments, which you can find below.

As you will see, both referees are quite supportive and suggest that the findings are novel and potentially interesting. However, they also identify limitations in the study and the manuscript, and raise concerns -regarding the level of support of the study's conclusions by the available data as well as technical issues such as the specificity of the used antibodies and missing controls from some experiments, among other concerns- that must be addressed for further consideration of the manuscript at The EMBO Journal.

Given the referees' comments and recommendations, I would like to invite you to submit a revised version of the manuscript along with a detailed point-by-point response addressing all referees' comments, as long as you are able and willing to embark on a major revision that will substantially strengthen the study and satisfactorily address the referees' concerns. I should add that it is EMBO Journal policy to allow only a single round of major revision, and acceptance of your manuscript will therefore depend on the completeness of your responses in this revised version. Please let me know if you have any questions or comments that you would like to discuss with me.

We generally allow three months as standard revision time (August 13, 2024). As a matter of policy, competing manuscripts published during this period will not negatively impact our assessment of the conceptual advance presented by your study. However, we request that you contact us as soon as possible upon publication of any related work, to discuss how to proceed. Should you foresee a problem in meeting this three-month deadline, please let us know in advance and we may be able to grant an extension.

Thank you for the opportunity to consider your work for publication in The EMBO Journal. I look forward to your revision.

Best regards,

Ioannis

Instructions for preparing your revised manuscript

1. When you are ready to submit the revision, please upload:

- A Word file of the manuscript text (including legends of main Figures, EV Figures and Tables). Please make sure that changes are highlighted (or "tracked") to be clearly visible.

- Individual production-quality figure files (one file per figure). When assembling your figures, please refer to our figure preparation guidelines in order to ensure proper formatting and readability in print as well as on screen:

If the data shown in a figure are obtained from n {less than or equal to} 2, please use scatter plots showing the individual data points.

- i. the name of the statistical test used to generate error bars and P values
- ii. the number (n) of independent experiments (please specify technical or biological replicates) underlying each data point (discussion of statistical methodology can be reported in the Materials and Methods section, but figure legends should contain a basic description of n , P , and the test applied)
- iii. the nature of the bars and error bars (s.d., s.e.m.).

- A point-by-point response to the referees' comments, with a detailed description of the changes made (as a word file). All

referees' concerns must be fully addressed and their suggestions taken on board. When preparing your letter of response to the referees' comments, please bear in mind that this will form part of the Review Process File and will therefore be available online to the community. Please note that you have the possibility to opt out of the transparent process at any stage prior to publication by letting the editorial office know (contact@embojournal.org); if you do opt out, the Review Process File link will point to the following statement: "No Review Process File is available with this article, as the authors have chosen not to make the review process public in this case.". For more details on our Transparent Editorial Process, please visit our website: <https://www.embopress.org/page/journal/14602075/authorguide#transparentprocess>

- Expanded View (EV) files (replacing Supplementary Information) that are collapsible/expandable online. A maximum of 5 EV Figures can be typeset. EV Figures should be cited as "Figure EV1, Figure EV2" etc. in the text, and their respective legends should be included in the manuscript file after the legends of regular figures. See detailed instructions regarding Expanded View files here:

- For the figures that you do NOT wish to display as Expanded View figures, they should be bundled together with their legends in a single PDF file called "Appendix", which should start with a short Table of Contents (including page numbers). Appendix figures should be referred to in the main text as: "Appendix Figure S1, Appendix Figure S2" etc. Please see detailed instructions here: <https://www.embopress.org/page/journal/14602075/authorguide#expandedview>

- A complete author checklist, which you can download from our author guidelines (<https://www.embopress.org/page/journal/14602075/authorguide>). Please note that the checklist will also be part of the Review Process File.

2. Please note that no statistics should be calculated and shown in Figures if $n=2$. Please also note that each p value should be reported as an exact value.

3. Before submitting your revision, primary datasets (and computer code, where appropriate) produced in this study need to be deposited in appropriate public databases (see <https://www.embopress.org/page/journal/14602075/authorguide#dataavailability>).

*** In particular, you are kindly requested to deposit all mass spectrometry and structural data produced in your study in appropriate public databases. ***

The accession numbers and databases should be listed in a formal "Data availability" section (placed after Materials and Methods) that follows the model below (see also

<https://www.embopress.org/page/journal/14602075/authorguide#dataavailability>):

Data availability

- RNA-seq data: Gene Expression Omnibus GSE46843 (<https://www.ncbi.nlm.nih.gov/geo/query/acc.cgi?acc=GSE46843>)
- [data type]: [name of the resource] [accession number/identifier/doi] ([URL or identifiers.org/DATABASE:ACCESSION])

*** All links should resolve to a page where the data can be accessed. ***

*** Please remember to provide in the Data availability section of your revised manuscript reviewer passwords if the datasets are not yet public. ***

*** The Data Availability Section is restricted to new primary data that are part of this study. In case you have no data that require deposition in a public database, please state so instead of referring to the database: "Our study includes no data deposited in public repositories." under the heading "Data availability". ***

4. Please check that the title and the abstract of the manuscript are brief, yet explicit, even to non-specialists. The length of the title should not exceed 100 characters, and the abstract should be a single paragraph not exceeding 175 words.

5. Please also note our reference format: <https://www.embopress.org/page/journal/14602075/authorguide#referencesformat>.

7. Please remember: digital image enhancement is acceptable practice, as long as it accurately represents the original data and conforms to community standards. If a figure has been subjected to significant electronic manipulation, this must be noted in the figure legend or in the "Materials and Methods" section. The editors reserve the right to request original versions of figures and

the original images that were used to assemble the figure.

8. Our journal encourages inclusion of data citations in the reference list to directly cite datasets that were obtained from public databases. Data citations in the article text are distinct from normal bibliographical citations and should directly link to the database records from which the data can be accessed. In the main text, data citations are formatted as follows: "Data ref: Smith et al, 2001" or "Data ref: NCBI Sequence Read Archive PRJNA342805, 2017". In the Reference list, data citations must be labeled with "[DATASET]". A data reference must provide the database name, accession number/identifiers, and a resolvable link to the landing page from which the data can be accessed at the end of the reference. Further instructions are available at: <https://www.embopress.org/page/journal/14602075/authorguide#referencesformat>.

9. We request authors to consider both actual and perceived competing interests. Please review our policy (<https://www.embopress.org/page/journal/14602075/authorguide#conflictsinterest>) and update your competing interests statement if necessary. Please name this section 'Disclosure and competing interests statement' and place it after the Acknowledgements section.

10. Please note that all corresponding authors are required to provide an ORCID ID upon submission of a revised manuscript (<https://orcid.org/>). Please find instructions on how to link your ORCID ID to your account in our manuscript tracking system in our Author guidelines (<https://www.embopress.org/page/journal/14602075/authorguide#authorshipguidelines>).

11. We use CRediT to specify the contributions of each author in the journal submission system. CRediT replaces the author contribution section, which should be removed from the manuscript. Please use the free text box to provide more detailed descriptions. See also guide to authors: <https://www.embopress.org/page/journal/14602075/authorguide#authorshipguidelines>.

13. We would also welcome the submission of cover suggestions or motifs to be used by our Graphics Illustrator in designing a cover.

14. Please use the link below to submit your revision:
<https://emboj.msubmit.net/cgi-bin/main.plex>

Referee #1:

The manuscript by Kurniyati et al. identifies 2 new bacterial virulence factors residing in a protein produced by the oral bacterial, *Treponema denticola*. The authors conclude that these virulence factors possess new mechanisms for evading host defense. Proteomic analysis of gels separating bacterial supernatants identified a 647 amino acid protein with homology of the C-terminus to previously identified cysteine proteases and an N-terminus containing 2 bacterial Ig-like domains. They show that the C- and N-terminal domains are cleaved with secretion of C-terminal products and bacterial surface expression of N-terminal products. The C-terminal protease (C0362) was shown to cleave complement components and protect bacteria against serum complement killing. The N-terminal fragment (N0362) was shown to inhibit neutrophil chemotaxis stimulated by fMLF or IL-8, to bind to plasma membrane receptors for those chemoattractants, and to inhibit neutrophil-mediated bacterial killing. Extensive structural studies complement those findings.

Critique:

1. The experiments performed with N0362 do not support the final conclusions about mechanism of action. In vitro killing with dHL-60 cells was performed with PMA stimulated cells in such a manner that chemotaxis would not be required. Despite an effect on chemotaxis of mouse and human neutrophils, experiments suggest the mechanism of action is at least partially independent of an effect on chemotaxis. This is consistent with the presence of N0362 on the bacterial surface, not supernatant, which would result in exposure of neutrophils only after direct contact with the bacteria. Additionally, the authors cite a reference (Sela et al, 1988) showing that *T. denticola* culture supernatants, but not direct contact with the organism, inhibit neutrophil superoxide generation. This is contrary to the authors' findings that N0362 is on the surface of the organism, and not released into the supernatant. Evaluation of neutrophil microbicidal functions by supernatant, contact with organisms, and recombinant N0362 may help determine the precise mechanism of action.

2. A second issue with the effect of N0362 is the loss of cell binding with elimination of either chemotactic receptor. The loss of binding to both receptors with the elimination of only one receptor implies that binding is not receptor specific. In light of this, the

basis for inhibition of signaling through CXCR1 needs to be evaluated more thoroughly. The extensive discussion of the structural basis for binding to each receptor is irrelevant unless this issue is resolved.

3. The extensive description of structure of the two major fragments seems too detailed. I would suggest limiting that description, at least in the main text, to structural details that define similarities and differences of TDE0362 to other proteases or virulence factors.

4. The first paragraph of the discussion presenting the "lines of evidence that TDE0362 is a unique virulence factor" is confusing and should list the evidence in a more concise, specifically enumerated manner.

5. The inflammatory infiltrate around abscesses in figure 9 is not apparent. A more detailed description and quantitation of the specific cells present would ensure the description is adequate.

Minor comments:

1. The authors use an older term for the formulated peptide, fMLF.

2. In the introduction, the description of neutrophil microbicidal mechanisms on page 3 should include generation of reactive oxygen species.

3. Some terms are introduced as abbreviations without adequate description (eg. N0362 on page 6. This requires breaking from the manuscript to determine the meaning of the abbreviation.

4. More thorough editing of appropriate verb tenses and singular/pleural nouns would improve readability.

Referee #2:

Kurniyati and coworkers report on the discovery of a novel *Treponema denticola* virulence factor, TDE0362, that is implicated in suppressing host defenses. The authors show that various proteolytic fragments of the protein are shed into the culture supernatants and inhibit complement activation by proteolytic inactivation of C3, C3b, C4 and C5 by the TDE0362 C-terminal Mac-1 domain; and inhibit neutrophil activation via the binding of N-terminal TDE0362 fragments to chemokine receptors CXCR1/2 and FPR1.

These findings are novel and mostly well validated and documented, resulting in a study that will be of high general interest. Before publication, there are a few points that require the authors attention, as outlined below.

Main points

1) The Western Blots and specificity of polyAb, mAb-C3 and mAb-N7 are unclear and need a careful characterization and validation. This results in several confusing and even contradictory results:

- in Figure 2A and 2B, why does Co-IP result in a different pattern of breakdown fragments than what is detected in the WT cell lysate shown in panel A?

- in Figure 2C polyAb and mAb-C3 analysis of WT supernatants show a different banding pattern. How can this be explained? Why would mAb-C3 only faintly detect the product around ~60 kDa? Is this partial obscuring of the C-terminus in the full-length protein? Do the authors see this using their recombinant proteins where they can control for concentration?

And why would polyAb not detect the band around 40-45 kDa?

On Page 8, line 7, the authors conclude the upper band detected by polyAb and mAb-C3 to be C2. Why not C1? It appears to run higher than C2 does in panel 2B.

- Figure S6B. What is being detected by PolyAb in the 0362 lane, presumably the TDE0362 knockout (the legend does not mention this)? It would seem that with the exception of the bands at 26-28 and around 12 kDa, the other bands reflect off-target binding by the PolyAb? This seems in contradiction to Fig. 2A,B,C though. Can the authors clarify what is going on?

- Figure S7A. The legend needs better description. The labelling of the lanes is not described. What are "Td, FL+ and C362+" ? The legend mentions recombinant FL362 and C362-1 were incubated for 22h with or without lysate supernatant. The PAGE appears to show only 0h and 22h with ?

When using lysate supernatant, why does this not show endogenous TDE0362 fragments? Presumably the knockout strain was used, but this is not mentioned.

The legend needs to stand on itself and mention these lysate supernatants, as mentioned in the text. Why is the banding pattern different to Fig. 2C?

These inconsistencies are problematic and require a careful analysis and clarification.

2) Page 8, lns 8-16. The immunofluorescence assays show that at least a fraction of TDE0362 is cell-associated, it does not formally prove it is surface localized. This would require proper controls for OM permeabilization. In Fig S8, the authors find a lack of cell staining using mAb-C3, and interpret this as "TDE0362 C-terminal cleaved products are not surface exposed". Please explain or reword, mAb-C3 will not detect C-terminally cleaved products, how could it inform on their location? This experiment would suggest that no intact TDE0362 is found cell associated (possible surface localized if correct controls are run). Also the hypothesis alluded to in the Discussion (Page 20, Ln18-19) that C3 and C4 are the two major forms that are 'exported' is premature. Rather than having a dedicated secretion apparatus, I suspect that TDE0362 is most likely to reach the cell surface as full-length, N-terminally anchored lipoprotein, where it is then proteolyzed into the fragments observed in the culture supernatant. Lipoproteins have been found to reach the cell surface using use broad-substrate pathways such as the BAM - barrel assembly machinery (doi: 10.1038/s41589-020-0575-0).

3) In the complement killing assays shown in Fig. 3g, H, the culture supernatants of the complementation strain CΔ0362 only partially complements the knockout mutant. Do the authors have an explanation for this? Is less protein produced or secreted in the complementation strain?

4) Page 10, ln 18-21. The authors mention 3 potassium ions are bound to N120. How did the authors assign the electron densities as corresponding to potassium?

5) Fig. 6E and 6G show %binding of dHL₆₀ cells and N0362, as determined by flow cytometry. How is %binding defined here, what does 100% binding correspond to? Legend, main text or Material methods are not clear on how this metric was derived and normalized.

6) Figure 9A shows the mean {plus minus}SEM abscess size in square mm for 5 mice. The variation is surprisingly small (just a couple square mm). How does one measure an abscess surface area with such precision? Throughout the manuscript, add individual data points to (or rather than) the bar graphs.

7) In page 17-18 the authors model the interaction between N0362 and FPR1 or CXCR1/2 using AlphaFold2. This paragraph is largely speculative and lacks the rigor of the rest of the manuscript. I do not consider this paragraph a requirement for publication, but if the authors insist on keeping the modelling in, this should include the details of the modelling effort and its validation based on per residue confidence scores, and ideally experimentally validation by mutation and/or MS crosslinking. The discussion of the 'identified binding mode' of N0362 and CXCR1/2 in comparison with IdeS or gasMac-1 is purely speculative and should be toned down or better substantiated by validation of the modelling effort. In this same paragraph, the authors write argue for a 'potential' structural homology of Cathepsin CTSG and N0362, the latter based on the AlphaFold2 model. The figures does not speak for a high level of structural similarity. What are the Z-score and RMSD? Ig domains are quite common in secreted bacterial proteins, and will all have a level of structural similarity based on the basic topology of the fold. The figure and this argument do not add so much to the hypothesis of 'molecular mimicry'. The experimental data showing that N0362 binds FPR1 are solid, however, mimicry or not.

8) Is Figure 10 meant to imply there is a channel in the *T. denticola* outer membrane? In general, Figure 10 does not add much as a main text Figure in its current form. It looks like a graphical abstract and adds unintended imprecision and speculation.

Minor edits

- Abstract and throughout text, "gram-negative and gram-positive" should be Gram-negative and Gram-positive.

- Fig S1A shows cell counts for denticola growth curves based on triplicates

- Fig. S2B the cDNA PCR amplicons run systematically higher than the equivalent DNA amplicons. Do the authors have an explanation for this?

- Page 7, ln 19 "Figure 2C" should be "Figure 2D".

- Figure 8 G-H. In the legends, should "D shows" and "E shows" be "G shows" and "H shows" ?

We truly appreciate the two reviewers for their thoughtful and constructive comments. To address their concerns, we revisited our results, carried out some new experiments to clarify those confusions, and revised the manuscript based on our new results. Below are our point-by-point responses to these comments.

Referee#1:

1. *The experiments performed with N0362 do not support the final conclusions about mechanism of action. In vitro killing with dHL-60 cells was performed with PMA stimulated cells in such a manner that chemotaxis would not be required. Despite an effect on chemotaxis of mouse and human neutrophils, experiments suggest the mechanism of action is at least partially independent of an effect on chemotaxis. This is consistent with the presence of N0362 on the bacterial surface, not supernatant, which would result in exposure of neutrophils only after direct contact with the bacteria. Additionally, the authors cite a reference (Sela et al, 1988) showing that T. denticola culture supernatants, but not direct contact with the organism, inhibit neutrophil superoxide generation. This is contrary to the authors' findings that N0362 is on the surface of the organism, and not released into the supernatant. Evaluation of neutrophil microbicidal functions by supernatant, contact with organisms, and recombinant N0362 may help determine the precise mechanism of action.*

Response: Sorry for the confusion. Our results don't support that N0362 exclusively resides at the cell surface. Instead, using mAb-N7, a monoclonal antibody against N0362, we detected several cleaved products in the spent culture supernatant of *T. denticola* (Figure S6E), indicating that N0362 is partially secreted or dissociated from the cell surface. We tried to perform neutrophil killing assays using the spent culture supernatant; however, it lysed HL-60 cells most likely due to presence of other toxic molecules, e.g., H₂S and proteases produced by *T. denticola*.

2. *A second issue with the effect of N0362 is the loss of cell binding with elimination of either chemotactic receptor. The loss of binding to both receptors with the elimination of only one receptor implies that binding is not receptor specific. In light of this, the basis for inhibition of signaling through CXCR1 needs to be evaluated more thoroughly. The extensive discussion of the structural basis for binding to each receptor is irrelevant unless this issue is resolved.*

Response: siRNA knockdown either FPR1 or CXCR1 significantly reduced, not completely abolished, the binding of N0362 to activated HL60 cells (See Figure S13D). Additionally, we performed co-IP, ELISA, and flow cytometry to confirm that the binding of N0362 to these two receptors is specific. Moreover, we measured the binding affinity of N0362 to CXCR1 using SPR. These experiments are robust. We are thus confident to conclude that the binding between N0362 and FPR1/CXCR1 is specific.

3. *The extensive description of structure of the two major fragments seems too detailed. I would suggest limiting that description, at least in the main text, to structural details that define similarities and differences of TDE0362 to other proteases or virulence factors.*

Response: As suggested, we have trimmed the structural details in the main text and focused on only those details necessary to understand the subsequent analysis and comparisons.

4. *The first paragraph of the discussion presenting the "lines of evidence that TDE0362 is a unique virulence factor" is confusing and should list the evidence in a more concise, specifically enumerated manner.*

Response: As suggested, this paragraph was completely revised.

5. *The inflammatory infiltrate around abscesses in figure 9 is not apparent. A more detailed description and quantitation of the specific cells present would ensure the description is adequate.*

Response: While we did not directly quantify specific immune cells induced by *T. denticola* in our model, immune cell infiltrates are seen (Figure 9D, green arrows) which are likely neutrophils based on previous studies in mice skin abscesses or gingival tissue infection that have shown that the majority of those infiltrates are neutrophils (Kesavalu et al, 1997; Verma et al, 2010). Significantly less immune cell infiltrates were seen in abscesses induced by the deletion mutant of TDE0362 (Figure 9C).

Minor comments:

1. The authors use an older term for the formulated peptide, *fMLF*.

Response: Different terms, such as *fMLP*, *fMLF*, and *fMIFL*, have been used for the formulated peptide. To avoid confusion, we used *fMLP*.

2. In the introduction, the description of neutrophil microbicidal mechanisms on page 3 should include generation of reactive oxygen species.

Response: Agreed and revised as suggested.

3. Some terms are introduced as abbreviations without adequate description (e.g., *N0362* on page 6. This requires breaking from the manuscript to determine the meaning of the abbreviation.

Response: Agreed and revised as suggested.

4. More thorough editing of appropriate verb tenses and singular/pleural nouns would improve readability.

Response: As suggested, we did thorough proofreading to fix those grammar issues.

Referee #2:

1) The Western Blots and specificity of polyAb, *mAb-C3* and *mAb-N7* are unclear and need a careful characterization and validation. This results in several confusing and even contradictory results: - in Figure 2A and 2B, why does Co-IP result in a different pattern of breakdown fragments than what is detected in the WT cell lysate shown in panel A?----- The legend needs to stand on itself and mention these lysate supernatants, as mentioned in the text. Why is the banding pattern different to Fig. 2C? These inconsistencies are problematic and require a careful analysis and clarification.

Response: Thank this reviewer for these thoughtful comments. To address these concerns, we revisited our results and found out that these confusing and even contradictory results are mainly caused by the following issues: (1) The processing of TDE0362 is complex and dynamic, e.g., its cleavage patterns vary at different growth phases (Figure S15); (2) Different antibodies were used and some of these experiments were carried out by different personnel; and (3) Recombinant TDE0362 proteins have autolysis activity. To solve these issues, we repeated these experiments and our new results showed that TDE0362 is secreted, cleaved by PrtP (also known as dentilisin), and its cleavage sites were further mapped. We believe that these new results (Figure 2 and Figure S6) have addressed these issues.

2) Page 8, line 8-16. The immunofluorescence assays show that at least a fraction of TDE0362 is cell-associated, it does not formally prove it is surface localized. This would require proper controls for OM permeabilization. In Fig S8, the authors find a lack of cell staining using *mAb-C3*, and interpret this as "TDE0362 C-terminal cleaved products are not surface exposed". Please explain or reword, *mAb-C3* will not detect C-terminally cleaved products, how could it inform on their location? This experiment would suggest that no intact TDE0362 is found cell associated (possible surface localized if correct controls are run). Also the hypothesis alluded to in the Discussion (Page 20, Ln18-19) that C3 and C4 are the two major forms that are 'exported' is premature. Rather than having a dedicated secretion apparatus, I suspect that TDE0362 is most likely to reach the cell surface as full-length, N-terminally anchored lipoprotein, where it is then proteolyzed into the fragments observed in the culture supernatant. Lipoproteins have been found to reach the cell surface using use broad-substrate pathways such as the BAM β -barrel assembly machinery (doi: 10.1038/s41589-020-0575-0).

Response: These comments are constructive. Our new results support the idea that after removal of the N-terminal signal peptide TDE0362 is exported and then proteolyzed into 4-5 different forms by PrtP, some of which are released into extracellular milieu and other are surface exposed. Our new data also suggested that TDE0362 is lipidated (Figure 2). Similar to other spirochetes, *T. denticola* encodes a BAM system. In addition, we recently identified a new protein secretion apparatus in *T. denticola*. These two possible secretion mechanisms are discussed and included in Figure 10. We are currently investigating if TDE0362 is exported through one of these two secretion apparatuses.

3) In the complement killing assays shown in Fig. 3G, H, the culture supernatants of the complementation strain *CA0362* only partially complements the knockout mutant. Do the authors have an explanation for this? Is less protein produced or secreted in the complementation strain?

Response: We did western blots using *T. denticola* whole cell lysates, but not the culture supernatants, and found that the level of TDE0362 in *CA0362* was higher than that of the wild type (Figure S4E-F). This experiment was repeated several times and a similar result was obtained. *CA0362* was created by replacing the *ermB* cassette in the $\Delta 0362$ mutant with a gentamycin resistance maker (*aacC1*) along with the full-length TDE0362 that is driven by the *tap1* promoter. We had attempted to address this issue by constructing a *trans*-complemented strain. Unfortunately, this attempt was unsuccessful.

4) Page 10, ln 18-21. The authors mention 3 potassium ions are bound to N120. How did the authors assign the electron densities as corresponding to potassium?

Response: Initially water molecules were modeled in these areas; however, significant residual positive difference density (mFo-DFc) was present at the three sites. We sequentially modeled each of these sites as ions from purification and crystallization conditions. Only potassium ions were able to satisfactorily resolve the residual positive difference density. Additional validation was conducted with the “CheckMyMetal” server, which agrees with assignment of 2/3 as potassium, the third is inconclusive. However, these sites are not hypothesized to be functionally relevant (i.e., crystallization artifacts) and we have thus removed the reference to these ions from the manuscript and corresponding figure (Figure 4).

5) Fig. 6E and 6G show % binding of dHL-60 cells and N0362, as determined by flow cytometry. How is % binding defined here, what does 100% binding correspond to? Legend, main text or Material methods are not clear on how this metric was derived and normalized.

Response: For this experiment, dHL-60 cells were first gated based on FSC-A/SSC-A profile, followed by the selecting single cells population (FSC-A/FCH-H), and subsequently detecting fluorescently labelled cells (Alexa Fluor 488-A/PE-A) in this population. And then, the program automatically calculated how many percent of the single cells population having Alexa Fluor 488 signal. Please see more details in Figure S12.

6) Figure 9A shows the mean \pm SEM abscess size in square mm for 5 mice. The variation is surprisingly small (just a couple square mm). How does one measure an abscess surface area with such precision? Throughout the manuscript, add individual data points to (or rather then) the bar graphs.

Response: As suggested, we replotted this figure with individual data points to more clearly demonstrate the distribution.

7) In page 17-18 the authors model the interaction between N0362 and FPR1 or CXCR1/2 using Alphafold2. This paragraph is largely speculative and lacks the rigor of the rest of the manuscript. I do not consider this paragraph a requirement for publication, but if the authors insist on keeping the modelling in, this should include the details of the modelling effort and its validation based on per residue confidence scores, and ideally experimentally validation by mutation and/or MS crosslinking. The discussion of the 'identified binding mode' of N0362 and CXCR1/2 in comparison with IdeS or gasMac-1 is purely speculative and should be toned down or better substantiated by validation of the modelling effort.

In this same paragraph, the authors write argue for a 'potential' structural homology of Cathepsin CTSG and N0362, the latter based on the Alphafold2 model. The figures does not speak for a high level of structural similarity. What are the Z-score and RMSD? Ig domains are quite common in secreted bacterial proteins, and will all have a level of structural similarity based on the basic topology of the fold. The figure and this argument do not add so much to the hypothesis of 'molecular mimicry'. The experimental data showing that N0362 binds FPR1 are solid, however, mimicry or not.

Response: Based on the feedback provided, it is clear that the methods and results of this section were not clear. We have thus revised the writing to describe more clearly what was conducted and the results and conclusions of the modeling:

Principally, AlphaFold was only utilized to generate the individual models (N0362 and CXCR1), not the complexes. We initially attempted such complexes using AlphaFold-multimer but the resultant models were non-sensical (e.g., binding to the intracellular G-protein-binding region or transmembrane regions). To mitigate the confusion, the recently available structure of CXCR1 bound to IL-8 has been substituted for the AlphaFold model of CXCR1. As written in the text, we conducted the search by visual inspection and comparison of N0362 and IL-8. IL-8 is principally recognized by and bound to CXCR1/2 using the N-terminus. While the similarity is indeed speculative, targeting of the primary mechanism for IL-8 binding is, in our opinion, highly likely, as only a small portion of the CXCR1/2 receptor sits above the membrane and would be accessible to N0362.

We have made modifications to the discussion text regarding the 'identified binding mode' of N0362 and CXCR1/2 compared to IdeS/gasMac-1 to be in line with the speculative nature of the modeling. We agree that the structural similarity between Cathepsin CTSG and N0362 is overall low and our initial text was misleading as to our true conclusion. We meant to say that the overall composition of β -strands in both structures suggests a conceivable mechanism for mimicry, we have updated the text to reflect such.

8) *Is Figure 10 meant to imply there is a channel in the T. denticola outer membrane? In general, Figure 10 does not add much as a main text Figure in its current form. It looks like a graphical abstract and adds unintended imprecision and speculation.*

Response: Figure 10 was revised based on our new results. We decided to keep this figure in the text because it summarizes the main discovery in this paper which will be greatly appreciated by readers.

Minor edits

1. *Abstract and throughout text, "gram-negative and gram-positive" should be Gram-negative and Gram-positive.*

Response: Revised as suggested.

2. *Fig S1A shows cell counts for denticola growth curves based on triplicates.*

Response: As suggested, we replotted the result based on triplicates.

3. *Fig. S2B the cDNA PCR amplicons run systematically higher than the equivalent DNA amplicons. Do the authors have an explanation for this?*

Response: We repeated this experiment. The new result showed that the cDNA PCR amplicons run equally as the equivalent DNA amplicons.

4. *Page 7, ln 19 "Figure 2C" should be "Figure 2D".*

Response: The error was corrected.

5. *Figure 8 G-H. In the legends, should "D shows" and "E shows" be "G shows" and "H shows" ?*

Response: The legend was revised.

References

Kesavalu L, Walker SG, Holt SC, Crawley RR, Ebersole JL (1997) Virulence characteristics of oral treponemes in a murine model. *Infect Immun* 65: 5096-5102

Verma RK, Rajapakse S, Meka A, Hamrick C, Pola S, Bhattacharyya I, Nair M, Wallet SM, Aukhil I, Kesavalu L (2010) Porphyromonas gingivalis and Treponema denticola Mixed Microbial Infection in a Rat Model of Periodontal Disease. *Interdiscip Perspect Infect Dis* 2010: 605125

Dear Chris,

Thank you again for submitting your revised manuscript EMBOJ-2024-117447R for consideration by The EMBO Journal. I sincerely apologize for the rather delayed decision on this occasion, which is due to the unavailability of the referees at the time your revision was submitted to our journal. Thank you for your understanding and patience.

The revised version of your manuscript has now been seen by the two original referees, and we have received their comments, which you can find below. As you will see, both referees acknowledge that many of the previously raised concerns have now been addressed, but they also point out a number of issues that have not yet been resolved, and they are not fully convinced about some of the experiments and conclusions of the study.

Please note that it is the EMBO Journal policy to normally allow a single only round of experimental revision. Although in light of the input we received from both referees we cannot proceed with acceptance of the manuscript as it stands, considering the potentially interesting results and the amount of work you have already performed to improve your study, we would exceptionally consider another revised version as long as you are willing to sufficiently address and resolve all remaining referee concerns. Please also submit a detailed point-by-point response to all referees' comments, describing all changes to the manuscript.

I would also like to note that there are several formatting and other minor issues with your manuscript. You are kindly requested to read our updated instructions below carefully and adhere to them in your revised manuscript; this will allow faster processing by our team and minimize further delays upon its resubmission.

Please let me know if you have any questions or comments that you would like to discuss with me. I look forward to your revision.

Best regards,

Ioannis

Instructions for preparing your revised manuscript

1. When you are ready to submit the revision, please upload:

- A Word file of the manuscript text (including legends of main Figures, EV Figures and Tables). Please make sure that changes are highlighted (or "tracked") to be clearly visible.

- Individual production-quality figure files (one file per figure). When assembling your figures, please refer to our figure preparation guidelines in order to ensure proper formatting and readability in print as well as on screen:

If the data shown in a figure are obtained from n {less than or equal to} 2, please use scatter plots showing the individual data points.

- i. the name of the statistical test used to generate error bars and P values
- ii. the number (n) of independent experiments (please specify technical or biological replicates) underlying each data point (discussion of statistical methodology can be reported in the Materials and Methods section, but figure legends should contain a basic description of n , P , and the test applied)
- iii. the nature of the bars and error bars (s.d., s.e.m.).

- A point-by-point response to the referees' comments, with a detailed description of the changes made (as a word file). All referees' concerns must be fully addressed and their suggestions taken on board. When preparing your letter of response to the referees' comments, please bear in mind that this will form part of the Review Process File and will therefore be available online to the community. Please note that you have the possibility to opt out of the transparent process at any stage prior to publication

by letting the editorial office know (contact@embojournal.org); if you do opt out, the Review Process File link will point to the following statement: "No Peer Review File is available with this article, as the authors have chosen not to make the review process public in this case.". For more details on our Transparent Editorial Process, please visit our website: <https://www.embopress.org/page/journal/14602075/authorguide#transparentprocess>

- Expanded View (EV) files (replacing Supplementary Information) that are collapsible/expandable online. A maximum of 5 EV Figures can be typeset. EV Figures should be cited as "Figure EV1, Figure EV2" etc. in the text, and their respective legends should be included in the manuscript file after the legends of regular figures. See detailed instructions regarding Expanded View files here:

- For the figures that you do NOT wish to display as Expanded View figures, they should be bundled together with their legends in a single PDF file called "Appendix", which should start with a short Table of Contents (including page numbers). Appendix figures should be referred to in the main text as: "Appendix Figure S1, Appendix Figure S2" etc. Please see detailed instructions here: <https://www.embopress.org/page/journal/14602075/authorguide#expandedview>

- A complete author checklist, which you can download from our author guidelines (<https://www.embopress.org/page/journal/14602075/authorguide>). Please note that the checklist will also be part of the Review Process File.

2. Please note that no statistics should be calculated and shown in Figures if $n=2$. Please also note that each p value should be reported as an exact value.

3. Before submitting your revision, primary datasets (and computer code, where appropriate) produced in this study need to be deposited in appropriate public databases (see <https://www.embopress.org/page/journal/14602075/authorguide#dataavailability>). Their accession numbers, databases, and the specific URLs (links) should be listed in a formal "Data availability" section (placed after Methods).

*** The Data Availability Section is restricted to new primary data that are part of this study. In case you have no data that require deposition in a public database, please state so instead of referring to the database: "Our study includes no data deposited in public repositories." under the heading "Data availability". ***

*** All links should resolve to a page where the data can be accessed. ***

*** Please remember to provide in the Data availability section of your revised manuscript reviewer passwords if the datasets are not yet public. ***

*** Please use detailed data citations for already available datasets that were re-analyzed in your study - for more information on the format, see point #9 below. ***

4. Please check that the title and the abstract of the manuscript are brief, yet explicit, even to non-specialists. The length of the title should not exceed 100 characters, and the abstract should be a single paragraph not exceeding 175 words.

5. All materials and methods need to be described in the manuscript using our "Structured Methods" format, which is now required for all research articles. According to this format, the Methods section includes a single "Reagents and Tools Table" - listing key reagents, experimental models, software and relevant equipment including their sources and relevant identifiers- followed by a "Methods and Protocols" section describing the methods. Please download and fill our Reagents and Tools Table template (.docx), which you can find in our author guide:

<https://www.embopress.org/page/journal/14602075/authorguide#structuredmethods>. When submitting your revised manuscript, please do not include the Reagents and Tools Table in the Methods section of the manuscript but upload it as a separate file choosing the file type "Reagent Table".

6. Please also note our reference format: <https://www.embopress.org/page/journal/14602075/authorguide#referencesformat>.

8. Please remember: digital image enhancement is acceptable practice, as long as it accurately represents the original data and conforms to community standards. If a figure has been subjected to significant electronic manipulation, this must be noted in the figure legend or in the "Materials and Methods" section. The editors reserve the right to request original versions of figures and the original images that were used to assemble the figure.

9. Our journal encourages inclusion of data citations in the reference list to directly cite datasets that were obtained from public

databases. Data citations in the article text are distinct from normal bibliographical citations and should directly link to the database records from which the data can be accessed. In the main text, data citations are formatted as follows: "Data ref: Smith et al, 2001" or "Data ref: NCBI Sequence Read Archive PRJNA342805, 2017". In the Reference list, data citations must be labeled with "[DATASET]". A data reference must provide the database name, accession number/identifiers, and a resolvable link to the landing page from which the data can be accessed at the end of the reference. Further instructions are available at: <https://www.embopress.org/page/journal/14602075/authorguide#referencesformat>.

10. We request authors to consider both actual and perceived competing interests. Please review our policy (<https://www.embopress.org/page/journal/14602075/authorguide#conflictsinterest>) and update your competing interests statement if necessary. Please name this section 'Disclosure and competing interests statement' and place it after the Acknowledgements section.

11. Please note that all corresponding authors are required to provide an ORCID ID upon submission of a revised manuscript (<https://orcid.org/>). Please find instructions on how to link your ORCID ID to your account in our manuscript tracking system in our Author guidelines (<https://www.embopress.org/page/journal/14602075/authorguide#authorshipguidelines>).

12. We use CRediT to specify the contributions of each author in the journal submission system. CRediT replaces the author contribution section, which should be removed from the manuscript. Please use the free text box to provide more detailed descriptions. See also guide to authors: <https://www.embopress.org/page/journal/14602075/authorguide#authorshipguidelines>.

14. We would also welcome the submission of cover suggestions or motifs to be used by our Graphics Illustrator in designing a cover.

15. Please use the link below to submit your revision:
<https://emboj.msubmit.net/cgi-bin/main.plex>

Referee #1:

While the revision submitted by Kurniyati et al. has responded to many of the reviewers' comments and is improved, I continue to have reservations about some of the authors' conclusions and some experimental design.

1. Knock down of either FPR or CXCR1 shown in figures S13 and 7 results in marked reduction in binding to dHL-60 cells. This suggests that N0362 binds simultaneously to those 2 receptors. However, the degree of reduction in receptor expression by siRNA is not provided. Therefore, it is unclear if both receptors are required for binding or if the binding is independent. The authors need to determine receptor expression and discuss the specificity of receptor binding.
2. Figure S12C does not show saturation of N0362 binding to human neutrophils, which would be expected if binding is receptor specific with the K_d shown in figure 8B. Is this a concentration issue and has binding saturation been shown?
3. The ability of N0362 to bind to FPR and CXCR1, but not C5AR, provides an important control for receptor specificity and mechanism of action. Comparison of N0362 inhibition of ligand-mediated calcium transients and chemotaxis among fMLF, IL-8, and C5a would allow the authors to conclude that N0362 mechanism of action results from receptor-specific antagonism. Without showing the specificity of the mechanism of action to FPR and CXCR1, the extensive speculation of possible structural mechanisms of N0362 binding is unnecessary.
4. The authors emphasize the role of N0362 inhibition of chemotaxis as the basis of prevention of *T. denticola* killing by neutrophils, as summarized in figure 10. However, ligand-stimulation of FPR and CXCR1/2 induce a number of other functional responses in neutrophils, in addition to chemotaxis. The methodology employed to measure dHL-60 cell killing is not dependent on either chemotaxis or on the expression of FPR or CXCR1. Mixing the spirochetes and dHL-60 together in culture plates reduces the need for cell recruitment. Stimulation of HL-60 cells with PMA results in receptor-independent activation of granule release and respiratory burst activity. My interpretation of the killing assays is that N0362 has an effect on dHL-60 activation by PMA that would be independent of chemoattractant receptors.

Minor comments;

1. Although it is possible to determine the number of replicates in experimental results shown as bar graphs, it would be most helpful to include that number in each figure legend.
2. Although many investigators continue to use the outdated abbreviation fMLP, I consider it better to use the correct

abbreviation for the amino acids in the tripeptide, fMLF.

3. On page 14 the authors state that binding of N0362 to dHL-60 cells detected by fluorescence was significantly reduced. However, I cannot find a statistical analysis to support that statement.

Referee #2:

Although the manuscript by Kurniyati et al. has improved by inclusion of the reviewers' comments and request, the remain issues that need to be resolved.

1) The authors revised the Western blot results. However, there remain a number of issues that are not explained.

In particular:

Fig. S4 - panel F. What is shown in this panel? How do the authors explain the bands below 75 kDa in delta0362 and deltaC0362? How does this differ from the FL0362 in Fig. 2C and why is the band present in the knockout mutants?

This ~75 kDa band is also seen in WT (ML) and deltaC0362 in Fig. S15.

2) The manual modelling of the N0362 - CXCR1/2 complex should be removed from the result section and Figures (Figure 8 D-G). What is the value of showing this? The model is purely speculative without any form of experimental or computational data to support or validate the proposed model. The result section should only include validated results.

A large part of the discussion also concerns the mode of interaction, and the authors also keep referring to the model as "identified binding model". The authors did not 'identify' any binding model. Without any form of validation or data, there is little value or significance in comparing binding modes of different proteins in the discussion.

Finally, I repeat my request that the inclusion of AlphaFold models (Fig. 1b) should always be accompanied by the proper statistics that show the local confidence of the model. This can be shown in a supplementary Figure.

3) Fig. 9A. I remain surprised by the small variance in the measured skin abscess sizes. The spread seems to be just a couple mm². In the labs' prior study that is referred to (Bian et al. 2013), the spread in skin abscess diameter of WT mice and *T. denticola* seems closer to ~30 mm². Can the authors explain this difference and the remarkably low variance? Can the authors provide representative images of the skin abscesses? At least for the review purpose.

Referee #1:

1. Knock down of either FPR1 or CXCR1 shown in figures S13 and 7 results in marked reduction in binding to dHL-60 cells. This suggests that N0362 binds simultaneously to those 2 receptors. However, the degree of reduction in receptor expression by siRNA is not provided. Therefore, it is unclear if both receptors are required for binding or if the binding is independent. The authors need to determine receptor expression and discuss the specificity of receptor binding.

Response: After the knock down experiment, we did carry out immunoblots to measure the expression of FPR1 and CXCR1 in dHL-60 cells. The result showed that the expression of FPR1 and CXCR1 was significantly reduced. This result was added to Fig. S13.

2. Figure S12C does not show saturation of N0362 binding to human neutrophils, which would be expected if binding is receptor specific with the K_d shown in Figure 8B. Is this a concentration issue and has binding saturation been shown?

Response: The objective of this experiment is to determine if N0362 binds to human neutrophils in a dose-dependent manner, not its binding affinity. If higher concentrations of N0362 were included, we anticipate that the binding of N0362 to human neutrophils would have reached binding saturation. That said, this is a concentration issue. For the similar experiment using dHL-60 cells as shown in Fig.6E-F, we indeed observed a trend of binding saturation when we replotted the result as lines instead of bar graphs (see below figure), e.g., % binding of N0362 and dHL-60 cells at 20 μg is very close to that at 30 μg .

3. The ability of N0362 to bind to FPR1 and CXCR1, but not C5a receptor, provides an important control for receptor specificity and mechanism of action. Comparison of N0362 inhibition of ligand-mediated calcium transients and chemotaxis among fMLF, IL-8, and C5a would allow the authors to conclude that N0362 mechanism of action results from receptor-specific antagonism. Without showing the specificity of the mechanism of action to FPR1 and CXCR1, the extensive speculation of possible structural mechanisms of N0362 binding is unnecessary.

Response: For the study shown Fig.8C, Chem-1, a reporter cell line for IL-8, was used. However, this cell line only expresses human CXCR1. Thus, we cannot perform the same experiment using fMLF and C5a. Regarding the possible structural mechanism of N0362, Reviewer 2 raised a similar concern. We agreed with this concern and thus deleted this part of the result from Fig. 8.

4. The authors emphasize the role of N0362 inhibition of chemotaxis as the basis of prevention of *T. denticola* killing by neutrophils, as summarized in figure 10. However, ligand-stimulation of FPR1 and CXCR1/2 induce a number of other functional responses in neutrophils, in addition to chemotaxis. The

methodology employed to measure dHL-60 cell killing is not dependent on either chemotaxis or on the expression of FPR or CXCR1. Mixing the spirochetes and dHL-60 together in culture plates reduces the need for cell recruitment. Stimulation of HL-60 cells with PMA results in receptor-independent activation of granule release and respiratory burst activity. My interpretation of the killing assays is that N0362 has an effect on dHL-60 activation by PMA that would be independent of chemoattractant receptors.

Response: We agreed that this is a collective effect of neutrophils chemotaxis and activation and accordingly revised Figure 10.

Minor comments

1. Although it is possible to determine the number of replicates in experimental results shown as bar graphs, it would be most helpful to include that number in each figure legend.

Response: Agreed and revised as suggested.

2. Although many investigators continue to use the outdated abbreviation fMLP, I consider it better to use the correct abbreviation for the amino acids in the tripeptide, fMLF.

Response: Revised as suggested.

3. On page 14 the authors state that binding of N0362 to dHL-60 cells detected by fluorescence was significantly reduced. However, I cannot find a statistical analysis to support that statement.

Response: A statistical analysis was done in Figure S13D.

Referee #2:

1) The authors revised the Western blot results. However, there remain a number of issues that are not explained.

In particular:

Fig. S4 - panel F. What is shown in this panel? How do the authors explain the bands below 75 kDa in delta0362 and deltaC0362? How does this differ from the FL0362 in Fig. 2C and why is the band present in the knockout mutants? This ~75 kDa band is also seen in WT (ML) and deltaC0362 in Fig. S15.

Response: Fig.S4F was probed against a polyclonal antibody of TDE0362 (polyAb) and processed with the SuperSignal™ Western Blot Enhancer kit in order to detect the cleaved products of TDE0362 as many as possible, particularly those small N-terminal cleaved products, e.g., F4-F5 shown in Fig. S6. Therefore, many non-specific bands were detected including ~75 kDa band. When normal ECL substrate was used, only two major bands were detected in WT and the complemented strain but not in the deletion mutant of TDE0362 (see new Fig. S4F). The molecular weights of these two bands are corresponding to F2 (~48 kDa) and F3 (~25 kDa) as shown in Fig. S6. To avoid the confusion, we replaced Fig.S4F with the blot processed with normal ECL substrate. Similar to Fig.S4F, Fig.S15 was performed to detect those cleaved products. As shown in Fig. S15B, a band nearly 25 kDa (F3) and one band below 20 kDa (F4/F5) were detected in WT but not in the mutant.

2) The manual modelling of the N0362 - CXCR1/2 complex should be removed from the result section and Figures (Figure 8 D-G). What is the value of showing this? The model is purely speculative without any form of experimental or computational data to support or validate the proposed model. The result section should only include validated results. A large part of the discussion also concerns the mode of interaction, and the authors also keep referring to the model as "identified binding model". The authors did not 'identify' any binding model. Without any form of validation or data, there is little value or significance in comparing binding modes of different proteins in the discussion.

Response: To address this concern, we removed Figure 8D-G and revised the discussion accordingly.

Finally, I repeat my request that the inclusion of AlphaFold models (Fig. 1b) should always be

accompanied by the proper statistics that show the local confidence of the model. This can be shown in a supplementary Figure.

Response: Agreed and revised as suggested.

3) Fig. 9A. I remain surprised by the small variance in the measured skin abscess sizes. The spread seems to be just a couple mm². In the labs' prior study that is referred to (Bian et al. 2013), the spread in skin abscess diameter of WT mice and *T. denticola* seems closer to ~30 mm². Can the authors explain this difference and the remarkably low variance? Can the authors provide representative images of the skin abscesses? At least for the review purpose.

Response: This is a well-established animal model that has been used by several groups. For this study, mice were injected with a define amount of bacterial cells and thus results are often reliable with low variance . The raw measurements of skin abscesses induced by three *T. denticola* strains as shown in Fig. 9A were presented in Data Sources. Actually, the smallest measurement for the mutant is approximate 11 mm², not a couple mm². Per reviewer's request, a representative images of mice with skin abscesses is shown below. Abscesses are clearly visible on mice skin.

Dear Chris,

Thank you again for submitting your revised manuscript (EMBOJ-2024-117447R1) to The EMBO Journal and for your patience during re-review. Your revised manuscript has now been seen by the two original referees (their comments are included below), and I am happy to say that they are both satisfied with the revision and mention that all previously raised concerns have been successfully addressed. I am thus happy to inform you that your manuscript has in principle been accepted for publication in The EMBO Journal. Congratulations on an excellent work and thank you for your comprehensive responses to the referees.

There are a few remaining editorial and formatting requests that we need from you to address in a final version of your manuscript before we can proceed with its publication:

- Please remove the figures from the main manuscript file, and upload them to our manuscript tracking system as individual high-resolution Figure files. Their legends should remain in the manuscript file, below the References.
- The full funding information should be both included in the Acknowledgements section of your manuscript and consistently entered in the manuscript tracking system (eJP) during re-submission. Please note that each funder (and grant) acknowledged in the manuscript must be provided as a separate funder item in eJP (not just pasted in the Comments box).
- Please revise your Author Checklist: only the sections of the manuscript where the relevant information can be found should be indicated in the last column of the checklist; the information itself should all be moved to the manuscript.
- Please provide a list of up to 5 relevant keywords after the Abstract of your revised manuscript.
- Please add a conflict-of-interest statement with the heading "Disclosure and competing interests statement" after the Methods section of your revised manuscript. More information can be found in our author guidelines: <https://www.embopress.org/page/journal/14602075/authorguide#conflictsofinterest>.
- As per our journal's policy, "data not shown" (on page 21) is not permitted. All data referred to in the paper should be displayed in the main or Expanded View figures, or in the Appendix. Please add these data or change the text accordingly if these data are not central to the study and its conclusions, or properly cite the respective published sources if these data can be found elsewhere.
- All materials and methods need to be described in the manuscript using our "Structured Methods" format, which is now required for all research articles. According to this format, the Methods section includes a single "Reagents and Tools Table" - listing key reagents, experimental models, software and relevant equipment including their sources and relevant identifiers- followed by a "Methods and Protocols" section describing the methods. Please download and fill our Reagents and Tools Table template (.docx), which you can find in our author guide: <https://www.embopress.org/page/journal/14602075/authorguide#structuredmethods>. When submitting your revised manuscript, please do not include the Reagents and Tools Table in the Methods section of the manuscript but upload it instead as a separate file choosing the file type "Reagent Table".
- We noticed that your manuscript does not contain any Expanded View (EV) Figures. If you wish, you could select and promote up to 5 supplementary figures for inclusion in the article as Expanded View Figures in order to improve their accessibility, visibility and utility. These figures will be displayed in the main HTML version of the paper in a collapsible format. The remaining figures that will not be promoted to the Expanded View should remain in the Appendix. More information can be found in our guide: <https://www.embopress.org/page/journal/14602075/authorguide#expandedview>.
- Please include the information about primers, proteins and plasmids used in your study (currently found in your Supplementary Tables S1 and S3) in your new Reagents and Tools Table.
- Please change "Supplementary Data" to "Appendix" on the first page of your Appendix pdf file.
- Please also include page numbers in the Table of Contents on the first page of your Appendix. The correct nomenclature for the Appendix items is "Appendix Figure S#" or "Appendix Table S#". Please update them throughout your Appendix.
- Callouts for supplementary figures and tables should be updated to "Appendix Figure S#" and "Appendix Table S#" throughout the main manuscript.
- The following statement should be removed from the manuscript: "Supplementary information includes three tables and 18 figures."
- The source data files need to be organized in a single .zip folder per Figure, named "SD Figure #.zip". For example, all Source

Data for the panels of Figure 1 should be included in (and uploaded as) a "SD Figure 1.zip" folder. If you also have source data to upload for Expanded View (EV) and/or Appendix Figures, please bundle them together in a single .zip folder.

- Please note that EMBO press papers are accompanied online by:

A) a short (2 sentences) summary of the findings and their significance,

B) 2-5 short bullet points highlighting the key results, and

C) a synopsis image in .jpg or .png format that is exactly 550 pixels wide and 300-600 pixels high (the height is variable). Please note that the text needs to be legible at the final size.

Please upload this information along with your revised manuscript (the text for A and B should be provided in a separate Word file).

- During our routine pre-acceptance checks, our data editors have raised the following queries regarding figures, data, and legends. Please make sure that all requests below are completely addressed in the Figure legends of the final version of your manuscript:

1. Please provide the exact p values in the legends of Figures 3g; 6a-d; 9a.
2. Please provide information related to "n" in the legends of Figures 3g-h; 6a-e, g; 8c.
3. Please define the error bars in the legends of Figures 6e, g.
4. Please define the scale bars for Figures 7e-f.

- Please also note re-order the manuscript sections as follows: Title page - Abstract & Keywords - Introduction - Results - Discussion - Methods - Data Availability - Acknowledgments - Disclosure Statement and Competing Interests - References - Figure Legends - (Main Tables with legends, if there are any) - Expanded View Figure Legends (if there are any).

Please also note that as part of the EMBO publications' Transparent Editorial Process, The EMBO Journal publishes online a Peer Review File along with each accepted manuscript. This File will be published in conjunction with your paper and will include the referee reports, your point-by-point response and all pertinent correspondence relating to the manuscript. You can opt out of this by letting the editorial office know (contact@embojournal.org). If you do opt out, the Peer Review File link will point to the following statement: "No Peer Review File is available with this article, as the authors have chosen not to make the review process public in this case."

We look forward to seeing a final version of your manuscript as soon as possible. Please let us know if you have any questions and use this link to submit your revision: <https://emboj.msubmit.net/cgi-bin/main.plex>.

Best wishes,

Ioannis

Referee #1:

The authors have addressed the reviews' comments.

Referee #2:

The authors successfully addressed any remaining concerns I had. This is a very nice study that will be of high general interest.

All editorial and formatting issues were resolved by the authors.

Dear Chris,

Congratulations on an excellent manuscript! I am very pleased to inform you that it has been accepted for publication in The EMBO Journal. Thank you for your comprehensive responses to the initially raised referees' concerns, and for addressing all editorial and formatting requests.

If you have any questions, please do not hesitate to contact the Editorial Office. Thank you for your contribution to The EMBO Journal. Working with you has been a pleasure!

Best wishes,

Ioannis
